

# Benchmarking the predictive capability of hydrological models for river flow and flood peak predictions across a large-sample of catchments in Great Britain

Rosanna A. Lane[1], Gemma Coxon[1], Jim E. Freer[1,3], Thorsten Wagener[2,3], Penny J. Johnes[1,3], John P. Bloomfield[4], Sheila Greene[5], Christopher J. A. Macleod[6], Sim M. Reaney[7]

[1]School of Geographical Sciences, University of Bristol, Bristol, BS8 2NQ, United Kingdom
[2]Faculty of Engineering, University of Bristol, Bristol, BS8 2NQ, United Kingdom
[3]Cabot Institute, University of Bristol, Bristol, BS8 2NQ, United Kingdom
[4]British Geological Survey, Maclean Building, Wallingford, OX10 8BB, United Kingdom
[5]Trinity College Dublin, Dublin, Ireland
[6]The James Hutton Institute, Craigiebuckler, Aberdeen, AB15 8QH, United Kingdom
[7]Department of Geography, Durham University, Durham, DH1 3LE, United Kingdom

*Correspondence to*: Rosanna A. Lane (R.A.Lane@bristol.ac.uk)

**Abstract.** Benchmarking model performance across large samples of catchments is useful to guide future model development. Given uncertainties in the observational data we use to drive and evaluate hydrological models, and uncertainties in the structure and parameterisation of models we use to produce hydrological simulations and predictions, it is essential that model evaluation is undertaken within an uncertainty analysis framework.

Here, we benchmark the capability of multiple, lumped hydrological models across Great Britain, by focusing on daily flow and peak flow simulation. Four hydrological model structures from the Framework for Understanding Structural Errors (FUSE) were applied to over 1100 catchments. Model performance was then evaluated using a standard performance metric for daily flows, and more novel performance metrics for peak flows considering parameter uncertainty.

Our results show that simple, lumped hydrological models were able to produce adequate simulations across most of Great Britain, with median Nash-Sutcliffe efficiency scores of 0.72-0.78 across all catchments. All four models showed a similar spatial pattern of performance, producing better simulations in the wetter catchments to the west, and poor model performance in Scotland and southeast England. Poor model performance was often linked to the catchment water balance, with models unable to capture the catchment hydrology where the water balance did not close. Overall, performance was similar between model structures, but different models performed better for different catchment characteristics and for assessing daily or peak flows, demonstrating the value of using an ensemble of model structures.

This research demonstrates what conceptual lumped models can achieve as a performance benchmark, as well as providing interesting insights into where and why these simple models may fail. The large number of river catchments included in this study makes it an appropriate benchmark for any future developments of a national model of Great Britain.





# 1 Introduction

Lumped and semi-distributed hydrological models, applied singularly or within nested sub-catchment networks, are used for a wide range of applications. These include water resources planning, flood/drought impact assessment, comparative analyses of catchment and model behaviour, regionalisation studies, simulations at ungauged locations, process based analyses, and
climate or land-use change impact studies (see for example Coxon et al., 2014; Formetta et al., 2017; Melsen et al., 2018; Parajka et al., 2007; Perrin et al., 2008; Poncelet et al., 2017; Rojas-Serna et al., 2016; Salavati et al., 2015; van Werkhoven et al., 2008). However, model skill varies between catchments due to differing catchment characteristics such as climate, land use and topography. Benchmarking the capability of hydrological models, by understanding where models perform well/poorly and the reasons for these variations in model performance, can help us better interpret modelling results across large samples
of catchments (Newman et al., 2017) and lead to more targeted model improvements through synthesising those interpretations. In the literature there has been a call for more large-sample hydrological studies, also known as comparative hydrology, testing hydrological models on many catchments of varying characteristics (Gupta et al., 2014; Sivapalan, 2009; Wagener et al., 2010). Gupta et al., (2014) discuss the need to pursue large sample hydrological studies, highlighting science benefits such an improved understanding of hydrological processes, robustness of generalizations and the development of catchment
classification schemes to inform understanding of model behaviour thus improving modelling of ungauged basins. Large-sample studies can teach us a lot about hydrological models; the appropriateness of model structures for different types of catchment characteristics (i.e. Van Esse et al., 2013), emergent properties and spatial patterns, key processes that we should be improving and identification of areas where models are unable to produce satisfactory results (e.g. Newman et al., 2015; Pechlivanidis and Arheimer, 2015).

## 1.1 National Scale Hydrological Modelling

Large-scale challenges such as climate change and population growth will impact river flow regimes across country wide hydro-climatic gradients affecting some regions more than others. Solutions need to be found to better predict water supplies, water scarcity and improve modelling studies used in the evaluation of national water policies such as the European Union's Water Framework Directive (European Parliament, 2000). National-scale hydrological modelling studies using a consistent
methodology across large areas are increasingly applied (Van Esse et al., 2013b; Højberg et al., 2013a, 2013b; McMillan et al., 2016; Veijalainen et al., 2010; Velazquez et al., 2010) facilitated by increasing computing power and the availability of open source large datasets such as the CAMELS or MOPEX datasets in the USA (Addor et al., 2017; Duan et al., 2006). For these large-scale challenges, national scale modelling approaches using a consistent methodology allow for comparison between places and identification of areas which may be most impacted.

However, the range of catchment characteristics and hydrological processes across national scales pose a great challenge to the implementation and evaluation of a national-scale model, as well as the need to provide predictions at ungauged catchments



(Lee et al., 2006). A national model ideally needs to represent varied catchment characteristics such as climate, topography and hydrogeology. Furthermore, whilst many hydrological studies focus only on natural catchments, a national scale approach must include catchments with human impacted modified flow regimes, for example heavily urbanised areas, rivers downstream from reservoirs and areas where abstractions and discharges are taking place to understand their impacts, albeit these can be

difficult to disentangle (Salavati et al., 2015). Incorporating this influence of human activity has been identified as a major challenge to advancing societally relevant hydrological analyses (Montanari et al., 2013).

## 1.2 Benchmarking hydrological models

Model skill varies between places, and it is therefore important for a modeller to understand the relative model skill for their study region, and how that relates to their core objectives. A single model structure will vary in its ability to produce good

flow time-series across different environments and time-periods (McMillan et al., 2016), expressed sometimes as model agility (Newman et al., 2017). It is important for a modeller to know the performance capabilities of their model when deciding whether to place confidence in any predictive skill.

The creation of a national benchmark series of performance of simple, lumped models can be useful for a variety of reasons. Firstly, a benchmark series of lumped model performance is a useful baseline upon which more complex or highly distributed

modelling attempts can be evaluated (Newman et al., 2015). This would ensure that future model developments are improving upon our current capability therefore justifying additional model complexity. Secondly, evaluating more complex hydrological models relative to benchmark performance of simple models ensures that the relative difficulty of simulating different catchments is implicitly considered (Seibert et al., 2018). Using benchmarks in this way can improve our evaluation of hydrological models, helping to identify whether a model is performing well in a catchment relative to how it should be

expected to perform for the particulars of that catchment. Thirdly, national benchmarks are useful for users of models as they can highlight areas where models have more or less skill, and where model results should be treated with caution. Large-scale benchmarking of hydrological models also has value as a large-sample hydrology study, enabling generalisations about how catchment characteristics or other factors may influence model performance (Gupta et al., 2014).

## 1.3 Assessing Uncertainty

Hydrological model output is always uncertain, due to uncertainties in the observational data used to drive and evaluate the models, boundary conditions, uncertainties in selection of model parameters and in the choice of a model structure (Beven and Freer, 2001). There is a large and rapidly growing body of literature on uncertainty estimation in hydrological modelling, with many techniques emerging to assess the impact of different sources of uncertainty on model output, as summarised in Beven (2009). Despite this, uncertainty estimation is not yet routine practice in comparative or large-sample hydrology and few

nationwide hydrological modelling studies have included uncertainty estimation, tending to look more at regionalization of parameters, multi-objective calibration techniques, or the use of flow signatures in model evaluation (i.e. Donnelly et al., 2016; Kollat et al., 2012; Oudin et al., 2008; Parajka et al., 2007b).



Hydrological modelling studies often assume that the rainfall and evapotranspiration data used to drive hydrological models, and the discharge data used to evaluate models is correct. However, errors in observational data can be high, especially for extreme events (Coxon et al., 2015; Mcmillan et al., 2012; Westerberg et al., 2016). In a recent review of observational uncertainties for hydrology, McMillan et al. (2012) found that measures of discharge typically have uncertainties in the range

2-19%, but low flows and out of bank flows have much higher uncertainties of 50-100% and 40% respectively. It is therefore important to consider these data uncertainties in the evaluation of hydrological models and any diagnostic evaluations (Coxon et al., 2014).

Parameter uncertainty can be evaluated through calibrating models within an uncertainty evaluation framework. There are many different uncertainty analysis procedures in hydrology such as the Parameter Solution (ParaSol) method (van Griensven

and Meixner, 2006), the Integrated Bayesian Uncertainty Estimator (IBUNE) (Ajami et al., 2007), the Sequential Uncertainty Fitting algorithm (SUFI-2) (Abbaspour et al., 2007), and the Generalized Likelihood Uncertainty Estimation (GLUE) framework (Beven and Binley, 1992). In this study, we choose to apply the GLUE framework which is based on the equifinality concept, that there are many different model structures and parameter sets for a given model structure which result in acceptable model simulations of observed river flow (Beven and Freer, 2001). This methodology has been widely applied to explore

parameter uncertainty within hydrological modelling (Freer et al., 1996; Gao et al., 2015; Jin et al., 2010; Shen et al., 2012) and includes approaches to directly deal with observational uncertainties in the quantification of model performance (Freer et al., 2004; Krueger et al., 2010; Liu et al., 2009).

Whilst many studies have explored parameter uncertainty, it is less common to evaluate the additional impact of model structural uncertainty on hydrological model output (Butts et al., 2004). Model structures can differ in their choice of processes

to include, process parameterisations, model spatial and temporal resolution and model complexity (i.e. the number of storage components). Studies attempting to address model structural uncertainty often apply multiple hydrological model structures and compare the differences in output (i.e. Ambroise et al., 1996; Vansteenkiste et al., 2014; Velázquez et al., 2013), and in climate impact studies (i.e. Bosshard et al., 2013; Karlsson et al., 2016; Samuel et al., 2012). These studies have found that the choice of hydrological model structure can strongly affect the model output, and therefore hydrological model structural

uncertainty is an important component of the overall uncertainty in hydrological modelling and cannot be ignored.

The use of flexible model frameworks has emerged as a useful tool for exploring the impact of model structural uncertainty in a controlled way, and for identifying the different aspects of a model structure which are most influential to the model output. These flexible modelling frameworks allow a modeller to build many different model structures using combinations of generic model components (Fenicia et al., 2011). For example, the Modular Modelling System (MMS) of Leavesley et al., (1996)

allows the modeller to combine different sub-models. The SUPERFLEX modelling framework presented by Fenicia et al., (2011) is based on generic building blocks such as reservoirs, junctions, and constitutive functions, and allows the modeller to generate new model configurations using these building blocks. There is also the Framework for Understanding Structural Errors (FUSE), developed by Clark et al., (2008), which combines process representations from four commonly used hydrological models to create over 79 unique model structures.





### 1.4 Study Scope and Objectives

The main objective of this study is to comprehensively benchmark performance of an ensemble of lumped hydrological model structures across Great Britain, focusing on daily flow and peak flow simulation. This will be the first evaluation of hydrological model ability across a large sample of British catchments whilst considering model structural and parameter uncertainty. This will be useful both as a benchmark of model performance against which other models can be evaluated and improved upon in Great Britain, and as a large-sample study which can provide general insights into the influence of catchment characteristics and selected model structure and parameterisation on model performance.

The specific research questions we investigate are:

1. How well do simple, lumped hydrological model structures perform across Great Britain, when assessed over annual and seasonal time scales via standard performance metrics?
2. Are there advantages in using an ensemble of model structures over any single model, and so are there any emergent patterns/characteristics in which a given structure and/or behavioural parameter set outperforms others?
3. What is the influence of certain catchment characteristics on model performance?
4. What is the predictive capability of these identified as behavioural models for then predicting annual maximum flows when applied in a parameter uncertainty framework?

To address these questions, we have applied the four core conceptual hydrological models from the FUSE hydrological framework to 1128 British catchments, within an uncertainty analysis framework. Model performance and predictive capability have been evaluated at each catchment, providing a national overview of hydrological modelling capability for simpler lumped conceptualisations over Great Britain.

## 2 Data and Catchment Selection

### 2.1 Catchment Data

This study was national in scope, using a large data set of 1128 Environment Agency catchments distributed across Great Britain (GB). The catchments cover all regions and include a wide variety of catchment characteristics including topography, geology and climate.

Rainfall is highest in the West and North of GB and lowest in the East and South varying from a minimum of 500mm to a maximum of 4496mm per year (see Fig. 1). There is also seasonal variation with the highest monthly rainfall totals generally occurring during the winter months and the lowest totals occurring in the summer months. This pattern is enhanced by seasonal variations in temperature with evaporation losses concentrated in the summer months from April – September. Besides climatic conditions, river flow patterns are also heavily influenced by groundwater contributions. Figure 1 shows the major aquifers in GB. In catchments overlying the Chalk outcrop in the South-East, flow is groundwater-dominated with a predominantly seasonal hydrograph that responds less quickly to rainfall events. Land use and human modifications to river flows also



significantly impact river flows, with river flows heavily modified in the South-East of and Midland regions of England due to high population densities.

Catchments were selected from the National River Flow Archive (Centre for Ecology and Hydrology, 2016) based upon the quality and availability of rainfall, potential evapotranspiration (PET) and river discharge data over the period 1988-2008. The

full NRFA dataset contains records for 1463 catchments across GB. Of these, 1128 had sufficient information available to include in this analysis. Catchments were only included if more than 2 years of discharge data were available within the time period.

## 2.2 Observational Data

Twenty-one years of daily rainfall and PET data covering the period 01/01/1988 to 31/12/2008 were used as hydrological

model input. The rainfall product used was the Centre for Ecology and Hydrology Gridded Estimates of Areal Rainfall, CEH-GEAR (Tanguy et al., 2014). This is a 1km$^2$ gridded product giving daily estimates of rainfall for Great Britain (Keller et al., 2015). It is based upon the national database of rain gauge observations collated by the UK Meteorological Office, with the natural neighbour interpolation methodology used to convert the point data to a gridded product (Keller et al., 2015). Catchment areal precipitation were then determined by averaging the values of all the grid squares that lied within the catchment

boundaries to produce a daily rainfall time series for each of the 1128 catchments.

The Climate Hydrology and Ecology research Support System Potential Evapotranspiration (CHESS-PE) dataset was used to estimate daily PET for each catchment. The CHESS-PE dataset is a 1km$^2$ gridded product for Great Britain, providing daily PET time-series (Robinson et al., 2015a). PET estimates were produced using the Penman-Monteith equation, calculated using meteorological variables from the CHESS-met dataset (Robinson et al., 2015b). Daily PET time series were produced for each

catchment by averaging values of all grid squares that lied within the catchment boundaries.

Observed discharge data was used to evaluate model performance. Gauged daily flow data from the National River Flow Archive (NRFA) was used for all catchments where available (Centre for Ecology and Hydrology, 2016).

## 3 Methodology

### 3.1 Hydrological Modelling

The Framework for Understanding Structural Errors (FUSE) modelling framework was used to provide four alternative hydrological model structures. This framework was selected as it enables comparison between hydrological models with varying structural components (Clark et al., 2008) and the computational efficiency of these relatively simple hydrological models enabled modelling to be carried out across a large number of catchments within an uncertainty analysis framework.

The four parent models from the FUSE framework were selected. These models are based on four widely used hydrological

models; TOPMODEL (Beven and Kirkby, 1979), the Variable Infiltration Capacity (ARNO/VIC) model (Liang et al., 1994; Todini, 1996), the Precipitation-Runoff Modelling System (PRMS) (Leavesley et al., 1983) and the Sacramento model



(Burnash et al., 1974). The models are all lumped, conceptual models of similar complexity and all run at a daily timestep within the FUSE framework. However, the structures of these models differ through the architecture of the upper and lower soil layers and parameterizations for simulation of evaporation, surface runoff, percolation from the upper to lower layer, interflow and baseflow (Clark et al., 2008), as shown in Fig. 2. This leads us to believe that the model structures are dynamically

different, yet as all are based on widely used hydrological models they are equally plausible (Clark et al., 2008).

The models were run within a Monte-Carlo simulation framework. There are 23 adjustable parameters within the FUSE framework, as shown in Table 2. Each of these was assigned upper and lower bounds based upon feasible parameter ranges and behavioural ranges identified in previous research (Clark et al., 2008; Coxon et al., 2014). Monte-Carlo sampling was then used to generate 10,000 parameter sets within these given bounds. Therefore, for each of the 1128 catchments, the 4

hydrological model structures were each run using the 10,000 possible parameter sets over the 21 year period 1988-2008. This resulted in >45 million simulations being carried out.

## 3.2 Evaluation of Model Performance

The objective of this study was to evaluate the model's ability to reproduce observed catchment behaviour with a focus on assessing the strengths and weaknesses of each model in different catchments. Given the large number of catchments evaluated,

it was not possible to evaluate model performance against multiple different objective functions with this paper, here we aim to benchmark behaviour to a couple of metrics. Consequently, we chose to evaluate the performance of the hydrological models through the widely used Nash-Sutcliffe Efficiency Index (Nash and Sutcliffe, 1970), however results are stored for a number of additional metrics not reported here. The efficiency index was calculated for each individual simulation using:

$$E = 1 - \frac{\sum(O_i - S_i)^2}{\sum(O_i - \bar{O})^2} \qquad (1)$$

where $O_i$ refers to the observed discharge at each timestep, $S_i$ refers to the simulated discharge at each timestep and $O$ is the mean of the observed discharge values. This results in values of $E$ between 1 (perfect fit) and $-\infty$, where a value of zero means that the model simulation has the same skill as using the mean of the observed discharges.

To gain insights into model agility and time varying model performance during different times of the year, we also assess differences in seasonal performance by splitting the observed and simulated discharge into March-May (Spring), June-August

(Summer), September-November (Autumn) and December-February (Winter). Seasonal Nash-Sutcliffe Efficiency values were then re-calculated for all the catchments, using only data extracted for that season. This allowed us to see if there were any seasonal patterns in model performance, for example during periods of higher or lower general flow conditions.

## 3.3 Evaluation of Model Predictive Capability

In order to evaluate model predictive capability, the widely applied Generalised Likelihood Uncertainty Estimation (GLUE)

framework was used (Beven and Freer, 2001; Romanowicz and Beven, 2006). For every catchment and model structure, an





Efficiency score was calculated for each of the 10,000 Monte Carlo (MC) sampled parameter sets. Parameter sets with an efficiency score exceeding 0.5 were regarded as behavioural, therefore all other sampled parameter sets were rejected and so given a score of zero. Conditional probabilities were assigned to each behavioural parameter set based on their behavioural Efficiency score, and these were normalised to sum to 1. For each daily timestep, a 5[th], 50[th] and 95[th] simulated discharge bound

was produced from these conditional probabilities, for each catchment and model structure individually as described in Beven and Freer (2001).

Predictive capability for an additional performance metric regarding annual maximum flows was then calculated from these behavioural simulations to test the model's ability to predict peak flood flows over the 21 year period. Annual maximum flows were extracted from both the observed discharge time-series and the 5[th], 50[th] and 95[th] percentile simulated behavioural

discharge uncertainty bounds. Two metrics were then used to assess the predictive capability of the models to this objective. The first metric aimed to assess the model's ability to closely replicate the observed annual maximum flows, whilst considering the plausible range of observational uncertainties that may be associated with the observed discharge value. Observed uncertainty bounds of ±13% were applied to all observed AMAX discharges. This observed error value was selected following previous research on quantifying discharge uncertainty at UK gauging stations for high flows (Coxon et al., 2015; Mcmillan

et al., 2012). The equations used to calculate the model skill relative to these observational uncertainty certainty bounds are

$$E_y = \frac{|O_y - S_y|}{O_y \times 0.13} \tag{2}$$

$$E_{mean} = \frac{\sum_{y=1}^{n} E_y}{n} \tag{3}$$

Where $E_y$ refers to skill for a particular year, $y$, $E_{mean}$ refers to skill across all years, $O$ refers to observed AMAX discharge for a particular year and $S$ refers to the 50[th] percentile simulated AMAX discharge. This results in a score of 0 if

the 50th percentile simulated AMAX is equal to observed AMAX discharge, a score of 1 if the simulated AMAX is at the limit of the observed error bounds and a score of 2 if it is twice the limit and so on in a similar approach to Liu et al., (2009) as a limits of acceptability performance score. A score was calculated for each of the 20 simulation years, excluding the first year as it was within the model warm-up period, as shown in Eq. (2). A mean score was then calculated across all years for each catchment and model, as shown in Eq. (3).

The second metric assessed how well the simulated AMAX uncertainty bounds (5th to 95th) overlapped observed AMAX uncertainty bounds to assess model skill given the range of predictive uncertainty. The range of overlap between the observed discharge uncertainty bounds and simulated bounds was first calculated for each year. This was normalised by the maximum range of the observed and simulated AMAX uncertainty bounds. The resulting value can be interpreted as the fraction of overlap versus the total uncertainty, whereby a value of 0 means the simulated AMAX bounds for a particular

year do not overlap the observations at all, and a value of 1 means the simulated bounds perfectly overlap the observational



uncertainties. Therefore, simulation bounds which overlap the observed AMAX uncertainty range due to having a very large uncertainty spread are penalised for this additional uncertainty width compared to the observed normalised uncertainty.

## 4 Results

### 4.1 National-scale Model Performance

Our first objective was to assess how well simple, lumped hydrological model structures perform across Great Britain, assessed over annual time scales via standard performance metrics. Maps showing the performance of each model structure, chosen here by using the maximum modelled NSE from the MC parameter samples, for catchments across Great Britain are given in Fig. 3.

Our results show that there is a large range in model performance across Great Britain, with catchment maximum NSE scores
ranging from 0.97 to <0. Generally, there is an east/west divide in model performance, with models typically performing better in wetter western catchments compared to drier catchments in the east. Clusters of poorly performing catchments can be seen in the east of England around London and in central Scotland, where all models are failing to produce satisfactory simulations. There are also areas where all models are performing well, such as south Wales, southwest England and southwest Scotland. The performance of the four model structures was similar, with the median maximum NSE from all the catchments being 0.72
for TOPMODEL, 0.75 for ARNO, 0.73 for PRMS and 0.78 for SAC. A similar spatial pattern of performance was also seen across all four model structures, with certain catchments resulting in poor or good simulations for all four model structures. Across all catchments, the average range in maximum NSE between the four models is only 0.08. This is far smaller than the range of maximum NSE gained by each model across all catchments.

The similar performance of the models could be partially due to the models all being run at the same spatial and temporal
resolution, having a similar model architecture splitting the catchment into upper and lower stores, and including the same process representations (such as lack of a snowmelt module). However, there are important differences between the models. The architecture of the upper and lower model layers differs, as can be seen in Fig. 2. TOPMODEL and ARNO/VIC have more parsimonious structures with only one store in each layer, while PRMS has a more complex upper layer which is split into multiple stores, and SACRAMENTO splits both upper and lower layers into multiple stores. The modelling equations
governing water movement between stores also differ, as explained in Clark et al., (2008). The number of model parameters is also an important difference between the models, as shown in Table 2, with TOPMODEL and ARNO/VIC having the least model parameters, with ten model parameters each, and the SACRAMENTO model having the most parameters with twelve.

### 4.2 Seasonal Model Performance

As part of our first objective, we also assessed how well models performed across GB when evaluated over seasonal time
scales, with results given in Fig. 4. These maps show the best sampled seasonal NSE score for each catchment taken from any of the FUSE model variants. There is a clear seasonal pattern to model performance, with models generally producing better





simulations during wetter winter periods. The models cannot produce adequate simulations for many catchments over the summer months of June to August, especially in the Southeast of England. However, for some catchments, especially catchments in the west, good simulations are produced all year round.

There is a seasonal impact on model performance across the areas previously identified as regions where models are failing. In central Scotland, model performance is generally worst during the winter and spring months of December to May, with a few catchments also being poorly simulated in summer. In south eastern England, model performance is particularly poor during the summer months of June-August. Reasons for this are discussed in later sections.

## 4.3 Model Structure Impact on Performance

An interesting question is whether a certain model structure is favoured for certain types of climatology or generalised catchment behaviour. Therefore, the relative performance of the four model structures ranked by both baseflow index (BFI) and annual catchment rainfall totals, is presented in Fig. 5. The Sacramento model tends to be the dominant model structure across most catchments, producing the largest number of behavioural simulations. However, catchment specific BFI and annual average rainfall both have an impact on which model structure tends to produce the most behavioural simulations as well as the total number of behavioural simulations.

Catchments with increasing BFI from 0 to 0.87 show an increasing trend of the SACRAMENTO model structure becoming dominant albeit with considerable variability (see Fig. 5a). TOPMODEL and PRMS performance relative to the other models decreased for catchments with increasing BFI, TOPMODEL especially is known to have a conceptual structure that better relates to a variable source area concept that does not relate as well to more groundwater dominated catchments. However, for slower responding and more groundwater dominated catchments with a BFI of greater than 0.9, the ARNO/VIC model was the only structure able to represent the hydrological dynamics well. ARNO-VIC is the only model that has a very strong non-linear relationship in its upper storage zone that links the deficit ratio of this store to saturated area extent and thus rainfall-driven surface runoff amounts. For very low values of the ARNO-VIC 'b' exponent (AXV_BEXP) as seen for high BFI vales in Fig. 6 for behavioural model distributions means that only at very high, near full upper storage levels is any larger extent of saturated areas predicted. This formulation clearly helps these more groundwater dominated catchments where both higher infiltration and percolation dynamics may be expected by constraining fast rainfall driven runoff process except to only more extreme storm event behaviour. It is also the reason why the sensitivity to BFI of this parameter is stronger in Fig. 6 than the other 'surface runoff' formulations that link storages to saturated area extent.

For catchments with annual rainfall totals below 2000mm (see Fig. 5b), there is no clear relationship between annual rainfall and relative performance of each model structure besides the SACROMENTO model tending to dominate. However, for catchments with average annual rainfall totals of above 2000mm, then TOPMODEL and ARNO/VIC became more dominant whilst the relative performance of the SACROMENTO model decreased. In effect the final trend is that for very wet catchment types (by rainfall totals) no model dominates, there is no 'gain' in the nuances of the non-linear model formulation and all structures can produce behavioural simulations from some part of their parameter space through a variety of flow pathway



mechanism from different storages. This again is clear in Fig. 6, where for at least 3 of the parameters shared between structures and controlling different parts of the hydrograph show little sensitivity across the parameter ranges sampled. The core exception to that is the TIMEDELAY parameter that controls the Gamma distribution routing formulation and shifts to less routing delay that is common to all model structures and so no one structure has an advantage. Similarly, TIMEDELAY is

also sensitive to high BFI catchments by increasing to longer routing times.

## 4.4 Influence of Hydrological Regime on Model Performance

The influence of hydrological regime was then assessed to see if there were specific types of catchments that the models were unable to represent given the spatial differences in model performance already observed. Catchment hydrological regime was defined using two metrics, the overall runoff coefficient (ratio of annual discharge to annual rainfall), and the catchment

wetness index (ratio of precipitation to potential evapotranspiration), results are provided in Fig. 7.

Fig. 7 shows that model performance relates to the catchment water balance. For catchments when the water balance tends to close, indicated as the area between the dashed lines in Fig. 7, the models are generally able to produce reasonable simulations. For these catchments, precipitation, evaporation and discharge are balanced, and runoff can be explained using the precipitation and evaporation data. When this relationship breaks down, we have situations where catchment runoff exceeds total rainfall

i.e. there is more water than we would expect, or catchments where runoff is low relative to precipitation, and this deficit cannot be explained solely by evapotranspiration i.e. the catchment is losing water. These catchments fall above the top dashed line in Fig. 7, or below the bottom dashed line, respectively. The models cannot simulate these catchments, as they cannot account for large water additions or losses, and so become stressed. This problem is most extreme for the driest catchments, where models may be converting less potential evaporation to actual evaporation as the conditions are drier, and so we have

an even larger water deficit which the model structures cannot simulate.

## 4.5 Benchmarking Predictive Capability for Annual Maximum Peak Flows

Model predictive capability for simulating annual maximum (AMAX) flows from behavioural models defined from the NSE measure is shown in Fig. 8. The top row of plots assesses the ability of models to produce AMAX discharge estimates which are as close as possible to observations. Here, a value of 0 means simulated AMAX discharge is equal to observed, up to 1

means simulated AMAX discharge is within the bounds of the observational uncertainties applied and larger values such as 2 indicate that simulated discharge is double the limit of observational uncertainties away from the observed discharge (negative values mean that the model simulations are lower than the observed). Median $E_{amax}$ values from Eq. (2) are around -2.4 to -3.2 across all four models, with PRMS producing slightly better predictions in general than the other models. This shows that the models are underestimating peak annual discharges across the majority of GB catchments even though behavioural model

have been selected using NSE which favours models that perform well at higher flows.

Fig. 9 shows the percentage overlap between the simulated 5th and 95th AMAX bounds and the observed AMAX uncertainty bounds. Here, the boxplot on the left shows the variation of results across all catchments and models for each year, whilst the





boxplot on the right summarizes results across all catchments and years for each model. The median value across all catchments is 0.16, meaning that there is a 16% overlap between the observed and simulated AMAX bounds averaged across all 20 years. There are large variations in model ability to simulate observed annual maximum flows between years, when looking at median predictions. For example, 1990 and 2008, which were wetter than average years across most of GB, model ability to represent

annual maximum discharge is poor. However, in 1996, which was a particularly dry year following the 1995 drought (Marsh et al., 2007), the models do a much better job of representing the annual maximum discharge. This may be in part due to the model tendency to underestimate discharge as seen in Fig. 8. However, variations between years are less apparent when looking at 25th and 75th percentiles in Fig. 8. This could suggest that there are some catchments where predictions are more consistent between years, or that the large climatic variation across GB may conceal some of the effects of inter-year differences.

## 10   5 Discussion

### 5.1 Benchmarking Performance of Multiple Lumped Hydrological Models Across GB

This study provides a useful benchmark of the performance and associated uncertainties of four commonly used lumped model structures across GB, for future model developments and model types to be compared against. The large number of catchments included in this study makes this assessment a fair benchmark for any future national modelling studies, as well as smaller

scale modelling efforts. Overall, it was found that simple, lumped models can perform well across most catchments in Great Britain. Using an ensemble of all four hydrological models, 93% of catchments studied produced a simulation with a Nash Sutcliffe value exceeding 0.5, and 75% of catchments exceeded a Nash-Sutcliffe value of 0.7. Individually, TOPMODEL, ARNO, PRMS and Sacramento produced simulations exceeding 0.5 NSE for 87%, 90%, 81% and 88% of catchments respectively, showing that overall performance of the four different models was similar but no single model structure was as

good as the ensemble. A similar spatial pattern of performance was seen for the four models. This indicates that some catchments are easy to model and can be represented by several different model structures and parameter sets, whereas other catchments cannot be modelled easily.

Model performance varied seasonally for some catchments, with a marked reduction in model capability in southeast England during summer, and Scotland during winter and spring. However, there were also many catchments, particularly those wetter

catchments in the west, where models were able to produce equally good simulations year-round.

There were some clusters of catchments, notably catchments in central Scotland and those on permeable bedrock in southeast England, where all models failed to produce good simulations. The Scottish catchments are mountainous catchments, at a considerably higher elevation than the rest of GB, and experience colder temperatures with daily maximum temperatures in January consistently below zero (Met Office, 2014). Scottish rivers are often impacted by anthropogenic flow controls, such

as dams and inter-basin transfers of stream flow. As model failures in Scotland were particularly pronounced during winter and spring, this suggests that models were unable to capture the different seasonal climatic conditions of these catchments, such as snow accumulation and melt or the impact of frozen ground. The FUSE models in this study do not incorporate





snowmelt processes, and the model failures show that inadequacies in the model structures could not be fully mitigated for by sampling wide parameter ranges. This shows that future modelling efforts for GB may need to include a snowmelt regime and in general some of the anthropogenic flow controls to capture Scottish hydrology.

The catchments in southeast England receive relatively little rainfall compared to the rest of GB and are overlaying a chalk
aquifer as can be seen in Fig. 1. Previous studies have found that hydrological models tend to perform better in wetter catchments (Liden and Harlin, 2000; McMillan et al., 2016), which could be part of the reason model performance is so poor for these catchments. The presence of the chalk aquifer could also stress the models, as there is nothing in the model structure to account for groundwater and particularly groundwater flows between catchment boundaries. Equally a considerable proportion of river discharges throughout the Thames are abstracted, this clearly impacts lower flow conditions in the drier
seasonal periods. This is further discussed in section 5.3.

**5.2 Insights from Applying an Ensemble of Model Structures Across a Large Sample of Catchments**

We found that the ensemble of model structures produced better results overall than any single model, showing that there is value in considering model structural uncertainty. Certain model structures were more likely to be considered behavioural, or resulted in the best sampled simulations, compared to others dependent on the catchment characteristics coupled to climatic
conditions. This is clearly seen in Fig. 5,7,8 and 9. The ensemble of model structures was able to take advantage of this, as can be seen by the different proportion of the four model structures comprising the behavioural ensemble for different climate and catchment characteristics (Fig. 5). This supports previous research highlighting the importance of considering alternative model structures and using model structure ensembles (Butts et al., 2004; Clark et al., 2008; Perrin et al., 2001). Exploring the relative performance of the different model structures within an uncertainty framework has enabled us to
identify scenarios where one model can become dominant or where all model structures are equally likely to generate behavioural simulations. Understanding why different model structures generally perform well for certain catchments can help us to identify parts of a model structure that may be particularly effective and can lead to model improvements. We found that for catchments with average annual rainfall values of around 2000mm/year or lower, the SACRAMENTO model structure is more dominant. As we move towards catchments with higher annual rainfall, the relative importance of the
different structures shift until all structures are approximately equal for the catchments with the highest annual rainfalls. This shows that for very wet catchments, the model structure is less important as all models can produce behavioural simulations through some part of the parameter space, as seen by the relatively high number of behavioural simulations for wetter catchments (Fig. 5b). This agrees with previous studies, where models have been found to perform better for wetter catchments, which are likely to have more connected saturated areas, as there is a more direct link between rainfall and
runoff (McMillan et al., 2016).

Another result that was particularly striking, was only the ARNO/VIC model was able to produce behavioural simulations for catchments with the very highest BFI's. This could be explained by the strong non-linear relationship in the upper storage zone of the ARNO/VIC model, which separates it from the other model structures. This enables the ARNO/VIC model to





constrain the fast rainfall-runoff processes, which would only occur for extreme events in these groundwater dominated catchments and so allow for a complex mixture of highly non-linear saturated fast responses coupled with more general baseflow dynamics to be captured effectively.

### 5.3 Influence of Catchment Characteristics and Climate on Model Performance

One of the key advantages of large-sample studies is that by applying models to many catchments, we can see general trends and identify important catchment characteristics or climates that are not represented well by our choice of model structures. We found that looking at the catchment water balance, considering the relationship between catchment precipitation, evaporation and observed flows, helped to identify common features of catchments where all models were failing. Firstly, all models failed in dry catchments where observed runoff exceeds total rainfall. Secondly, there was a pattern of extreme model

failures (no model could produce simulations with a Nash-Sutcliffe value exceeding zero) in dry catchments with very low observed flows relative to the observed rainfall, such that the difference between runoff and rainfall could not be explained by evapotranspiration alone. Thirdly, we found that models produced poor results in catchments of varying degrees of wetness index when runoff was much lower than rainfall, and this deficit was larger than the observed PET. Potential reasons for this will be discussed below.

Firstly, all model structures produced poor simulations in dry catchments where total runoff exceeded total rainfall. Potential reasons for catchments having higher runoff than precipitation include human modifications to the natural flow regime such as dams, effluent returns or inter-catchment water transfers, groundwater flow between catchments or it is also possible that there are systematic errors in the observational data and this information is dis-informative (Beven and Westerberg, 2011; Kauffeldt et al., 2013). Most of these catchments were located within chalk aquifers in southeast England, and therefore are in

a heavily urbanised area where groundwater flows between catchments could be expected. The simple, lumped models used here were only given inputs of observed precipitation and PET, therefore they are unable to account for the additional observed runoff and so are 'stressed' even in terms of simulating mean annual runoff, irrespective of more detailed hydrograph behaviour.

Secondly, the worst performing model simulations were generally found to be for dry catchments where observed runoff was

very low compared to total rainfall, and this runoff deficit could not be accounted for by evapotranspiration losses alone. For most of these catchments, the ensemble of model structures was unable to produce any simulations with Nash-Sutcliffe efficiency values exceeding zero. Similarly to above, this scenario could occur in catchments influenced by human modifications such as inter-catchment water transfers or abstractions, or groundwater flows between catchments. Most of these catchments were also located in southeast England, where groundwater flows between catchments as well as human

modifications are more prevalent. This would severely stress the model structures, as evaporation alone would not be able to account for the loss of water from the catchment, and processes such as human-modifications and inter-catchment groundwater flows are not accounted for within the model structures. Previous studies have also found poorer model performance in drier catchments (McMillan et al., 2016).



Finally, we also found models performed poorly in catchments where runoff was substantially lower than rainfall, irrespective of wetness index, and this runoff deficit could not be explained by the evapotranspiration data. These catchments were geographically spread, but many were found to be human-impacted by reservoirs, hydro-electric schemes or abstractions, again a lack of sensible water balance results in a clear link to non-behavioural performance.

Whilst it has been noted that there is a general pattern of poor performance for catchments in southeast England, it is hard to disentangle the reasons that this may be the case. Both the underlying chalk geology causing water transfer between catchments, and heavily human modified flow regimes could explain model failures which are greatest during the summer. Interestingly, McMillan et al., (2016) found that whilst aquifer fraction was expected to have a strong link to model performance, no relationship was found for the TOPNET model applied in New Zealand.

These results highlight the difficulty in national and large-scale modelling studies, which for GB must incorporate human modified hydrological regimes, complex groundwater processes, a range of different climates and the potential of dis-informative data, or at least a lack of process understanding to adjust model conceptualisations. Whilst simple, lumped hydrological models can produce adequate simulations for most catchments, the model structures are put under too much stress when trying to simulate catchments where the water balance does not close or is increasingly departing more normal conditions.

The models fail or produce poor simulations when large volumes of water enter or leave the catchment due to human activities or groundwater processes, indicating the importance of considering these influences in any national study. What is striking here in these results, is that general hydrological processes, defined by water availability and BFI metrics to infer the extent of slower flow pathways, are important in defining the quality of simulated output and differences in model structures and parameter ranges, even though nationally many catchments are impacted by additional anthropogenic activities such as

abstractions and multiple flow structures.

### 5.4 Predictive Capability of Models for Predicting Annual Maximum Flows

Predictions of annual maximum discharge using behavioural models based on Nash-Sutcliffe Efficiency posed a larger challenge for the models, even when allowing for an estimate of observational uncertainty from results generalised in Coxon et al., (2015). It was found that all model structures systematically underpredicted annual maximum flows across most

catchments. These results are in line with previous large-scale modelling efforts. McMillan et al., (2016) report that their TOPNET model applied across New Zealand showed a smoothing of the modelled hydrograph relative to the observations, which resulted in overestimation of low flows and underestimation of annual maximum flows. Newman et al., (2015) found the same effect in their study covering 617 catchments across the US.

It was found that there were some variations in the ability of models to simulate AMAX flows between years, and this often

related to the wetness of a particular year. Models tended to perform worse in wetter years, and better in drier years. This could be linked to the fact that all models tended to underestimate annual maximum flows, and therefore are closer to observations in years with lower annual maximum flows.



## 5.5 Uncertainty Evaluation in Hydrological Modelling

This study evaluated both model parameter and model structural uncertainty. The results showed that there is considerable value in using multiple model structures. No one model structure was appropriate for all catchments, seasons and when evaluating different metrics from the hydrographs. We found the Sacramento model resulted in the best NSE values overall, the ARNO/VIC model proved best for high baseflow catchments yet the PRMS model was the best at capturing AMAX peak flows. This highlights the benefits of using an ensemble of hydrological models or a flexible model structure framework such as FUSE when modelling many varied catchments. Furthermore, it was found that for some catchments only a selection of the model structures were able to produce good simulations, such as the baseflow dominated catchments which only ARNO could simulate well and became dominant for a particular advantage in the model structure. For these catchments, selection of the appropriate model structure is important to produce good simulations and unsuitability of the model structure cannot be corrected for through parameter calibration. Consequently, future hydrological modelling over a national scale and/or over a large sample of catchments need to ensure appropriate model structures are selected for these catchments and consider the possibility that multiple model structures to represent hydrological processes in varied catchments.

The results also highlighted the importance of considering parameter uncertainty. It was shown that there were often many different parameter sets which could produce good simulation results for the same model structure. For some catchments, particularly the wetter catchments in the west, all model structures were able to produce good simulations through sampling the parameter space. We also shown how behavioural parameter distributions change with regards to BFI, which shows expected shifts in some of the common behavioural parameters/concepts for different conditions, showing the model behaviour and parameter formulations are in general making rationale sense (i.e. Higher BFI equals higher time delays).

While this study incorporated uncertainties in model structures and parameters, future work will also focus on incorporating uncertainties in the data used to drive hydrological models and more sophisticated representation of discharge uncertainties. This is important because errors in observational data will introduce errors to runoff predictions when fed through rainfall-runoff models (Andréassian et al., 2001; Fekete et al., 2004; Yatheendradas et al., 2008), and in conjunction with uncertainties in the observational data used to evaluate hydrological models will also affect our ability to calibrate and evaluate hydrological models (Blazkova and Beven, 2009; Coxon et al., 2014; McMillan et al., 2010; Westerberg and Birkel, 2015). This is particularly important when modelling across large samples of catchments as comparisons between catchments may be incorrect or biased in the face of erroneous data.

## 6 Summary and Conclusions

In this study, we have benchmarked the performance of an ensemble of lumped, conceptual models across over 1100 catchments in Great Britain.

Overall, we found the four models performed well over most of Great Britain, with median Nash-Sutcliffe efficiency scores of 0.72-0.78 across all catchments. The performance of the four models was similar, and all models showed similar spatial





patterns of performance, and there was no single model that outperformed the others across all catchment characteristics and for both daily flows and peak flows. This demonstrated the value in using an ensemble of model structures when exploring national-scale hydrology.

We found that all models showed higher skill in simulating the wet catchments to the west, and all models failed in areas of Scotland and southeast England. Seasonal performance and analysis of the water balance suggested that these model failures could be at least in part attributed to missing snowmelt or frozen ground processes in Scotland and chalk geology in southeast England where water was able to move between catchment boundaries. In general, we found models performed poorly for catchments for catchments with unaccounted losses or gains of water, which could be due to measurement errors, water transfer between catchments due to groundwater aquifers and human modifications to the water system. Therefore, these factors would need to be considered in a national model of Great Britain.

We also evaluated model predictive capability for high flows, as good model performance in replicating the hydrograph, assessed using Nash-Sutcliffe efficiency, does not necessarily mean models are performing well for other hydrological signatures. We found that the FUSE models tended to underestimate peak flows, and there were variations in model ability between years with models performing particularly poorly for extremely wet years.

This benchmark series provides a useful baseline for assessing more complex modelling strategies. From this we can resolve how or where we can and need to improve models, to understand the value of different conceptualisations, linkages to human impacts, and levels of spatial complexity our model frameworks could deploy in the future.





**Code availability**

FUSE model code is introduced in Clark et al., (2008), and is available upon request from the lead author.

All model outputs from this study are available upon request from the lead author.

**Data availability**

All datasets used in this study are publicly available. The CEH-GEAR and CHESS-PE datasets are freely available from CEH's Environmental Information Data Centre, and can be accessed through https://doi.org/10.5285/5dc179dc-f692-49ba-9326-a6893a503f6e and https://doi.org/10.5285/8baf805d-39ce-4dac-b224-c926ada353b7 respectively. Observed discharge data from the National River Flow Archive is available from the NRFA website.

**Author contribution**

Jim Freer, Gemma Coxon and Rosie Lane were involved in the project conceptualization and formulating the methodology. Rosie Lane was responsible for most of the formal analysis, running the model simulations and analysing the results. Data visualization was split between Rosie Lane and Gemma Coxon, with guidance from Jim Freer and Thorsten Wagener. Rosie Lane prepared the original manuscript, with contributions from Gemma Coxon, Jim Freer and Thorsten Wagener. Finally Penny Johnes, John Bloomfield, Sheila Greene, Kit Macleod, and Sim Reaney helped shape the initial ideas for this research
as part of their involvement in the National Modelling workpackage of NERC's Environmental Virtual Observatory Pilot.

**Competing Interests**

The authors declare that they have no conflict of interest.

**Disclaimer**

**Acknowledgements**

This work is funded as part of the Water Informatics Science and Engineering Centre for Doctoral Training (WISE CDT) under a grant from the Engineering and Physical Sciences Research Council (EPSRC), grant number EP/L016214/1. Much of the national data sources to make this research possible were originally obtained from NERC grant NE/1002200/1 Environmental Virtual Observatory Pilot. John Bloomfield publishes with the permission of the Executive Director of the British Geological Survey (UKRI).





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





**Figures**

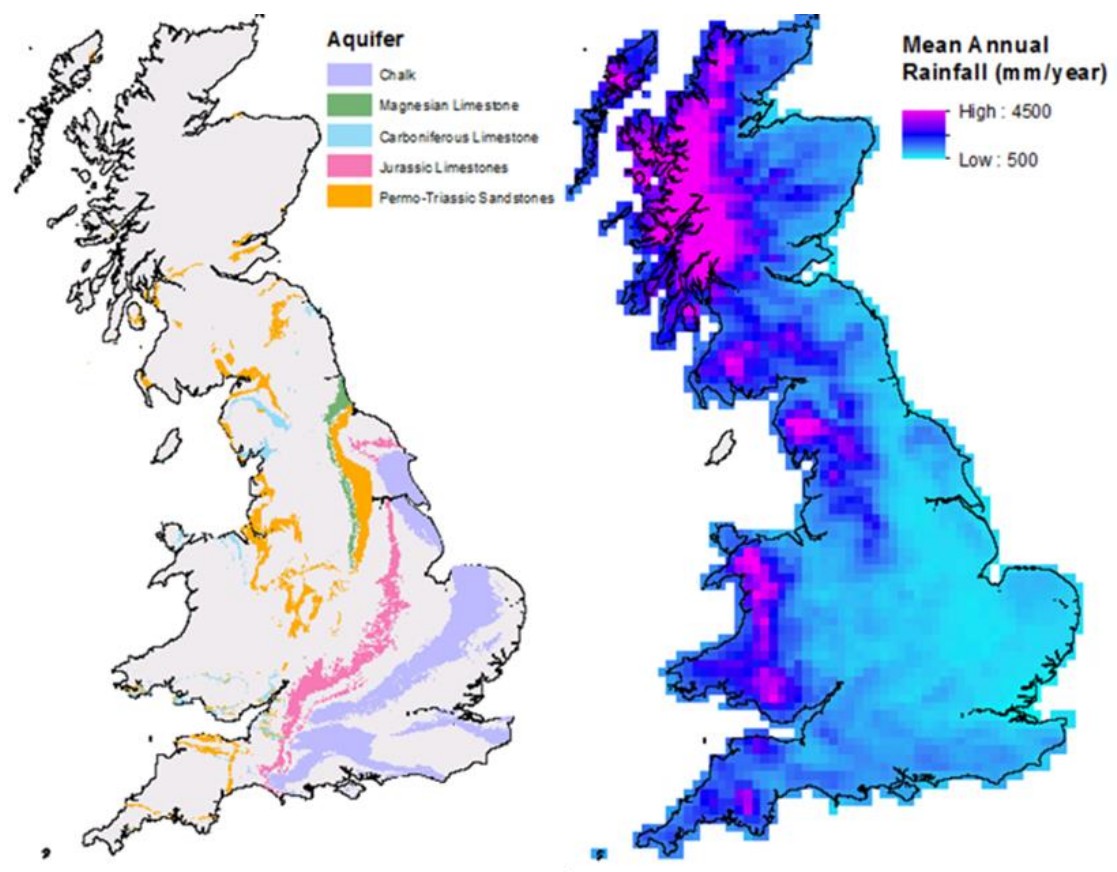

**Figure 1: A) Major aquifers across Great Britain, based upon BSS Geology 625k, with the permission of the British Geological Survey B) Mean annual rainfall for 10km² rainfall grid cells across Great Britain.**





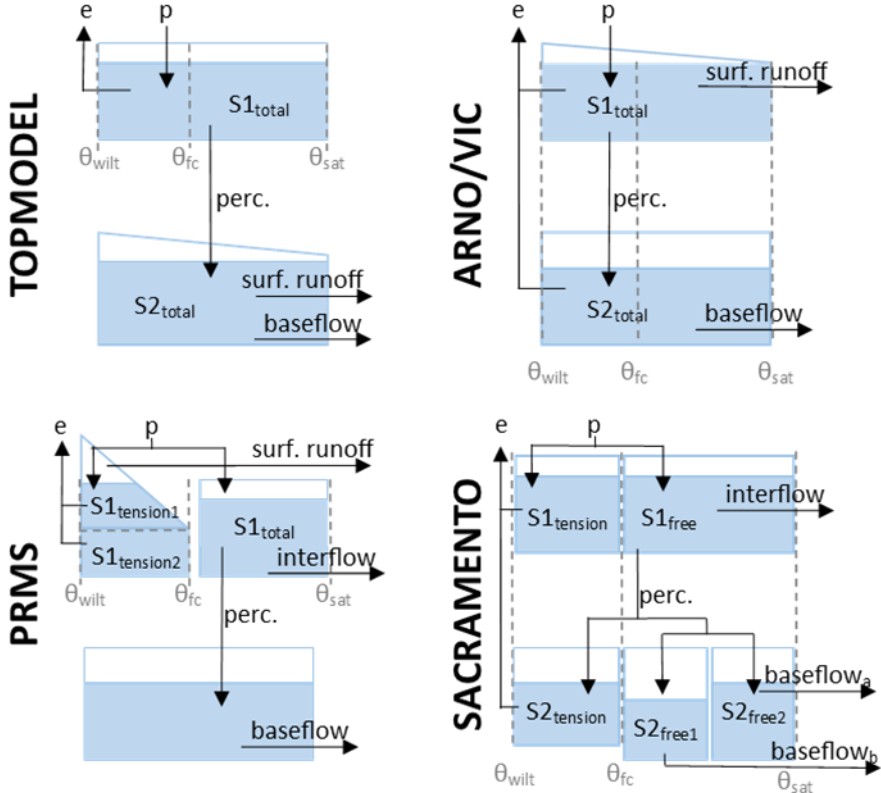

**Figure 2: FUSE wiring diagram. Adapted from Clark et al., (2008).**





**Figure 3: GB maps of model performance for each structure. Each point is a gauge location which is coloured based upon the best Nash Sutcliffe score attained by the model for that catchment.**



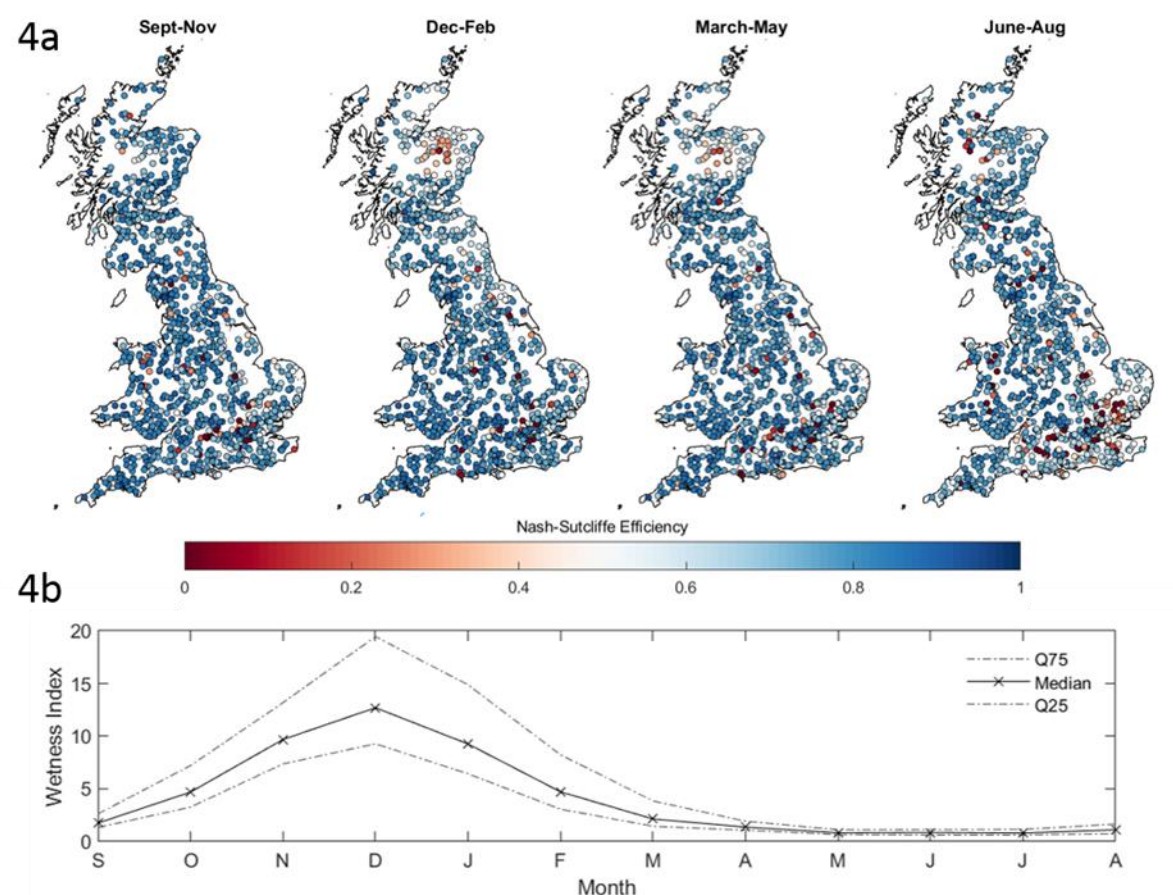

**Figure 4: GB maps of FUSE multi-model ensemble model performance for each season (4a) and observed seasonal variations in catchment wetness index (4b). Each point on 4a is a gauge location which is coloured based upon the best Nash Sutcliffe score attained by any of the four models sampled for that catchment and season. Figure 4b then shows how seasons vary hydrologically across GB, through the wetness index (precipitation/PET) calculated from the observed data, split by month, used to drive the hydrological models across all catchments shown in 4a.**





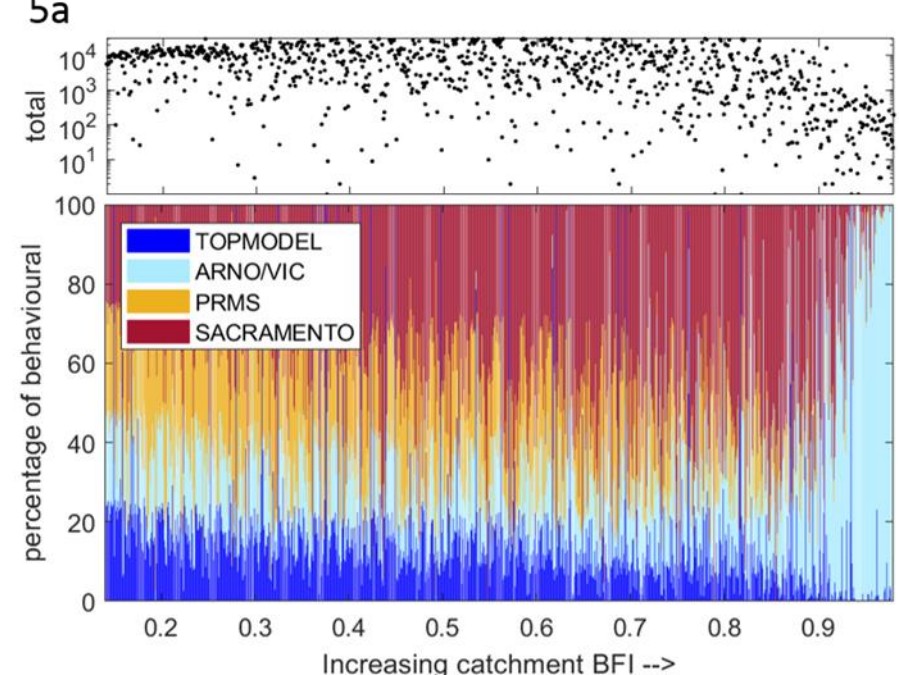

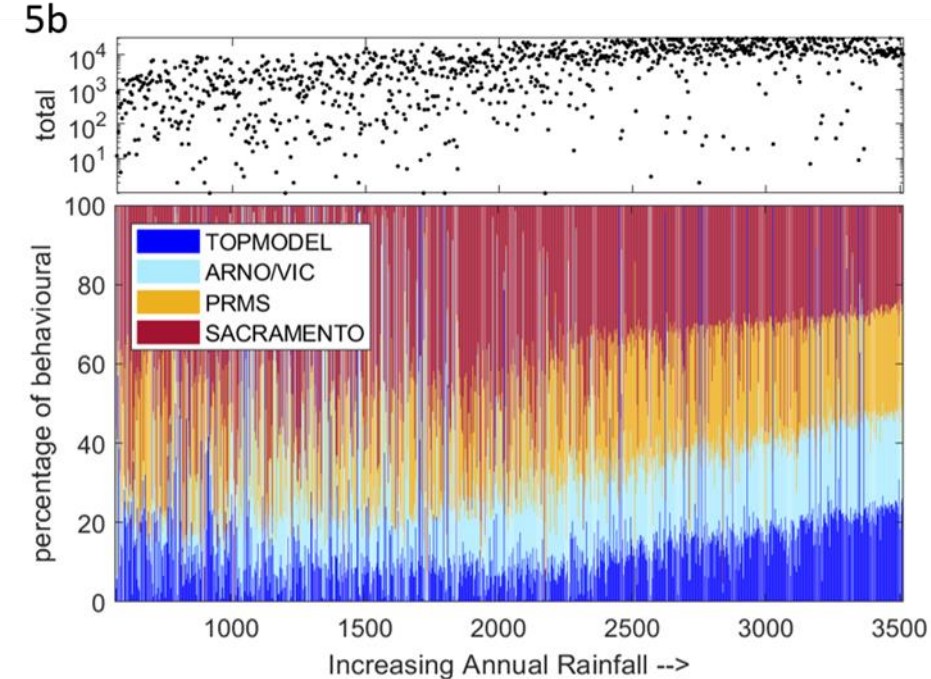

**Figure 5: Relative performance of the four FUSE model structures, depending on catchment characteristics. Scatter plots show the total number of behavioural simulations, from all model structures, forming each line on the stacked bar graph. Each line on this stacked bar chart represents 1 catchment, and the colour shows the proportion of the behavioural simulations from each model structure. Catchments have been ordered by BFI (5a) and Annual Rainfall (5b).**





**Figure 6: Cumulative distribution function (CDF) plots showing parameter values of the behavioural simulations for each catchment. Each line represents a catchment and is coloured by that catchment's BFI. The 4 rows show different parameters controlling different parts of the hydrograph. Surface runoff is given by the LOGLAMB (TOPMODEL), AXV_BEXP (ARNO) and SAREMAX (PRMS and SAC) as there was no common surface runoff parameter used for all 4 models. Each column is a different hydrological model.**





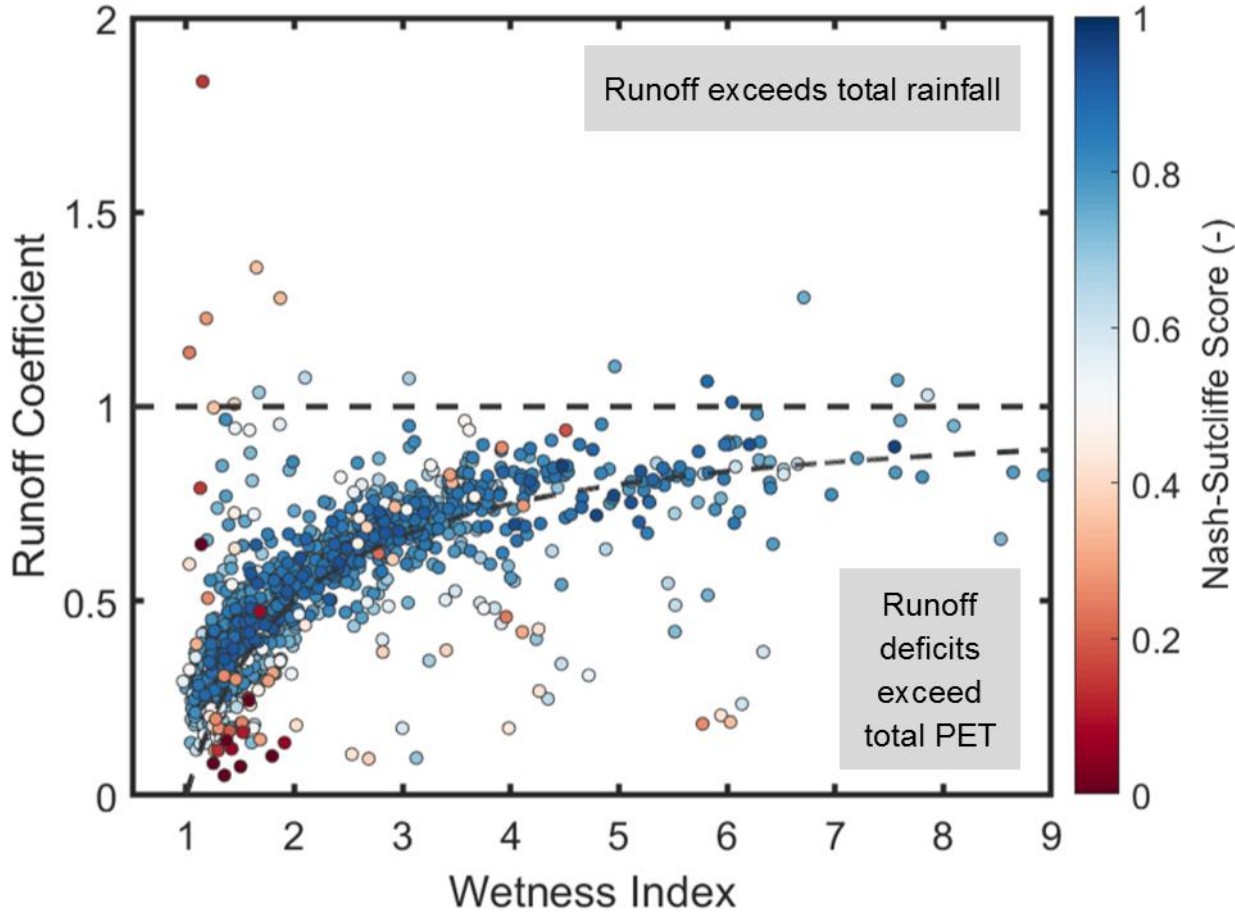

**Figure 7: Scatter plot of the relationship between wetness index, runoff coefficient and best sampled model performance. Each point represents a catchment, coloured by the best Nash-Sutcliffe score for that catchment from the model structure ensemble. The plotting order was modified to ensure catchments with more extreme (high and low) NSE values would be plotted on top. Any points above the horizontal dotted line are where runoff exceeds total rainfall in a catchment and any points below the curved line are where runoff deficits exceed total PET in a catchment.**





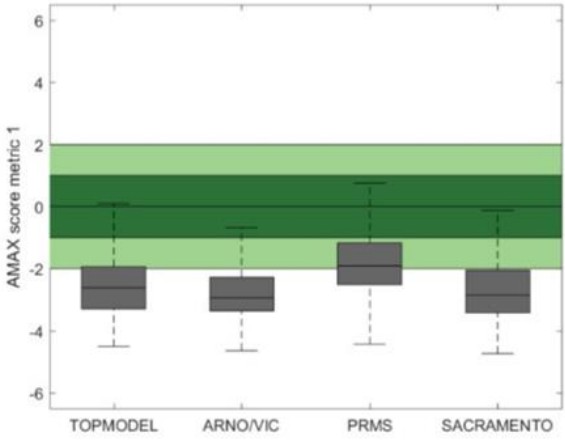

**Figure 8: Predictive capability of 4 hydrological models for annual maximum (AMAX) flows across Great Britain. Shows behavioural model ensemble (NSE>0.5) median performance in replicating the observed AMAX flows, with a value of 0 being a perfect score and a value of 1 meaning the simulated AMAX value was at the limits of the observational uncertainty. The spread covers all catchments.**

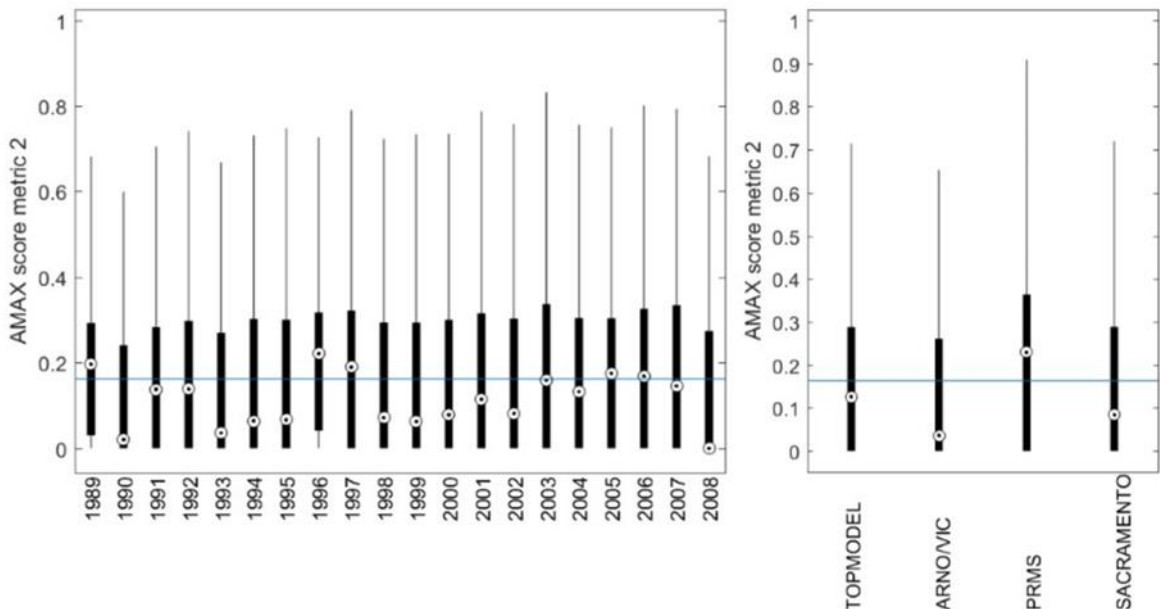

**Figure 9: Predictive capability of 4 hydrological models for annual maximum (AMAX) flows across Great Britain. Boxplots show the overlap of the simulated and observed uncertainty bounds, as a fraction of the total uncertainty. This metric ranges from 0 to 1, with 0 indicating no overlap between observed and simulated AMAX discharge and 1 indicating a perfect overlap of observed and simulated discharge bounds. The range in the left plot is over all catchments and all models, whilst the right-hand plot shows the range across all catchments.**





## Tables

**Table 1: Characteristics of the 1128 catchments included in this study. Values for Mean annual rainfall, runoff, loss, flood peaks and peak daily flows were calculated from the model input timeseries. Other values were taken from the UK hydrometric register (Marsh and Hannaford, 2008).**

| Variable | 95th percentile | Median | 5th percentile |
|---|---|---|---|
| Catchment Area [km$^2$] | 1364 | 129 | 15 |
| Baseflow Index [-] | 0.86 | 0.47 | 0.3 |
| Mean Annual Rainfall [mm] | 2369 | 964 | 619 |
| Mean Annual Runoff [mm] | 1920 | 521 | 139 |
| Mean Annual Loss [mm] | 702 | 460 | 216 |
| Median Annual Flood Peak [mm] | 51 | 13 | 2 |
| Peak Daily Flow [mm] | 100 | 29 | 4 |
| Gauge Elevation [m] | 228 | 39 | 5 |
| Urban Extent [%] | 22 | 1 | 0 |



**Table 2: FUSE parameters and defined upper and lower bounds.**

| Parameter | Description | Units | Lower Bound | Upper Bound | Model(s) using parameter |
|---|---|---|---|---|---|
| MAXWATER₁ | Depth of upper soil layer | mm | 25 | 500 | TOPMODEL, ARNO, PRMS, SAC |
| MAXWATER₂ | Depth of lower soil layer | mm | 50 | 5000 | TOPMODEL, ARNO, PRMS, SAC |
| FRACTEN | Fraction total storage in tension storage | - | 0.05 | 0.95 | TOPMODEL, ARNO, PRMS, SAC |
| FRCHZNE | Fraction tension storage in recharge zone | - | 0.05 | 0.95 | PRMS |
| FPRIMQB | Fraction storage in 1st baseflow reservoir | - | 0.05 | 0.95 | SACRAMENTO |
| RTFRAC1 | Fraction of roots in the upper layer | - | 0.05 | 0.95 | ARNO |
| PERCRTE | Percolation rate | mm day⁻ | 0.01 | 1000 | TOPMODEL, ARNO, PRMS |
| PERCEXP | Percolation exponent | - | 1 | 20 | TOPMODEL, ARNO, PRMS |
| SACPMLT | SAC model percolation multiplier for dry soil layer | - | 1 | 250 | SACRAMENTO |
| SACPEXP | SAC model percolation exponent for dry soil layer | - | 1 | 5 | SACRAMENTO |
| PERCFRAC | Fraction of percolation to tension storage | - | 0.5 | 0.95 | SACRAMENTO |
| FRACLOWZ | Fraction of soil excess to lower zone | - | 0.5 | 0.95 | PRMS |
| IFLWRTE | Interflow rate | mm day⁻ | 0.1 | 1000 | PRMS, SACRAMENTO |
| BASERTE | Baseflow rate | mm day⁻ | 0.001 | 1000 | TOPMODEL, ARNO |
| QB_POWR | Baseflow exponent | - | 1 | 10 | TOPMODEL, ARNO |
| QB_PRMS | Baseflow depletion rate | day-1 | 0.001 | 0.25 | PRMS |
| QBRATE_2A | Baseflow depletion rate 1st reservoir | day-1 | 0.001 | 0.25 | SACRAMENTO |
| QBRATE_2B | Baseflow depletion rate 2nd reservoir | day-1 | 0.001 | 0.25 | SACRAMENTO |
| SAREAMAX | Maximum saturated area | - | 0.05 | 0.95 | PRMS, SACRAMENTO |
| AXV_BEXP | ARNO/VIC b exponent | - | 0.001 | 3 | ARNO |
| LOGLAMB | Mean value of the topographic index | m | 5 | 10 | TOPMODEL |
| TISHAPE | Shape parameter for the topographic | - | 2 | 5 | TOPMODEL |
| TIMEDELAY | Time delay in runoff | days | 0.01 | 7 | TOPMODEL, ARNO, PRMS, SAC |