# Peer review of "Benchmarking the predictive capability of hydrological models for river flow and flood peak predictions across over 1000 catchments in Great Britain"

_Hydrology and Earth System Sciences, 2018_

## Referee Comment (RC1) · Thibault Mathevet (Referee) · 17 Feb 2019

I carefully read the paper by Lane et al.. This paper appeared to be particularly clear, well written and easy to follow. Scope and objectives are stated clearly, the presentation of results is rather straightforward. As you probably know it, I appreciate this kind of study on a large sample of watersheds. I am very happy to know that such a large sample exists for GB. Studies on large sample give generality and robustness to the results. This paper gives insights on the general hydrology of GB and predictive capabilities of 4 simple rainfall-runoff models. I really appreciated §4 and §5, par-

ticularly analyses linked to the seasonality (fig 4), BFI (fig 5, 6), and water balance closure (fig 7). Thanks to this large sample of watersheds in GB with a variety of hydrologic/hydrogeologic functioning (even in the same country), these results appear to be robusts, with a general interest. The link between BFI (main undergroud processes) and model structure agility is really interesting.

»> Main comments Evaluation of model performance and selection of model :

Authors decided to use the classical Nash-Sutcliffe efficiency (NSE) index to evaluate model performances (and select behavioural models, NSE > 0.5). NSE index is famous and widely used in Rainfall-Runoff modeling. Even if the perfect efficiency index do not exists, this index is also known to have some drawbacks (Schaefli and Gupta, 2007, among many references). Gupta et al. (2009) introduced the Kling-Gupta efficiency index that allows to explicitly account for bias (mean and variability) and correlation, in the evaluation of model performances. Given the ambition of this paper, I would recommend the authors to consider in their analyses the Kling-Gupta efficiency index, or at least to decompose their results in terms of correlation and mean bias.

Poor performances on floods :

Authors found that the different models had poor performances on floods, which is generally the case when classical modeling schemes are used to optimise or select parameter sets. I appreciate the simple way authors evaluate models on flood values, however I would add a figure to explain the two metrics. One of the main drawback of the NSE (and linear regression as well) is that the standard deviation of the simulated time-series is biased and underestimated, i.e. flood underestimated and drought overestimated. Among other arguments, this drawback partly explain why flood values are underestimated. I would add at least a comment on the fact that this statement is dependent on the behavioural model selection metrics in §5.4. If authors update their paper using KGE to select their behavioural models, they might revise (a bit) their findings on model performances for floods.

[Figure]

Focus on droughts ? :

Given the ambition of this paper, I think that this paper would also benefit from a focus on droughts. Hence, analyses on doughts could be complementary to analyses on relative model performances (among the 4 tested structures), since droughts might also be driven by BFI in GB ? The link with groundwater flows could also be shown, if a focus on droughts is done. Authors could use the same metrics as for floods. It could be better to use the 10 days or 30 days annual minimal value, instead of the annual minimal value, which could be highly impacted and uncertain.

»> Minor comments :

In §1.1 : authors discuss the benefits of national scale hydrological modelling. Another benefits could be the production of parameter libraries, which could be used for regional studies or model calibration on poorly gauged to ungauged basins or engineering studies. Authors can make references to papers on this subject (Perrin et al., 2008 ; Rojas-Serna et al., 2016 ; or some other works by Seibert).

In §5.1 : authors did not use a snow accumulation and melt routine in their modeling framework. Very simple snow routine are availables, in the spirit of the simple models proposed in FUSE. The CemaNeige routine could be a good candidate to improve model simulations on the few catchments where it's necessary. Depending on the proportion of snow impacted catchments, using a snow routine would improve model performances and the paper, as it could give answers to some hypotheses of the paper. Valéry, A., Andréassian, V., Perrin, C., 2014. 'As simple as possible but not simpler': What is useful in a temperature-based snow-accounting routine? Part 1 – Comparison of six snow accounting routines on 380 catchments, Journal of Hydrology, 517(0): 1166-1175.

Valéry, A., Andréassian, V., Perrin, C., 2014. 'As simple as possible but not simpler': What is useful in a temperature-based snow-accounting routine? Part 2 – Sensitivity analysis of the Cemaneige snow accounting routine on 380 catchments, Journal of

[Figure]

Hydrology, 517(0): 1176-1187.

In §5.3 : authors discuss about groudwater flows between catchments, with losses or gain of waters. This problem is not new and some conceptual modelisation could be found in the litterature since one or two decades. In a natural context, authors could make a reference to Le Moine at al. (2007, 2008) papers about groundwater flows and water balance closure. The existence of such groundwater flows in permeable geological context (chalk, limestones and/or karstic systems, etc.) was one of the reasons of the development of a groundwater exchange function within the GR model family. The use of this function should be motivated by (hydrogeologic) evidences of such groundwater flows (in order to avoid "overfitting" of the water balance, i.e. fudge factor), but might be usefull in catchments where water balance is difficult to close, such as the one influenced by chalk aquifers in southeast england.

Le Moine, N., Andréassian, V., Perrin, C. & Michel, C., 2007. How can rainfall-runoff models handle intercatchment groundwater flows? Theoretical study based on 1040 French catchments. Water Resources Research 43(6), W06428

Le Moine, N., V. Andréassian, T. Mathevet, 2008. Confronting surface- and ground-water balances on the La Rochefoucauld-Touvre karstic system (Charente, France), Water Resour. Res., 44, W03403

»> Last comments :

P4, l22 : I would also make a reference to Perrin et al. 2001 here

P7, l20 : mistake with O (mean of observed discharge)

P10, l22 : values instead of vales

P17, l8 : for catchments, repeated 2 times

In §2, I would give an estimation of the proportion of watersheds where snowmelt processes are observable (solid precipitation >20% of total precipitation ?)
Table 1 is not cited within §2

In §3.3, the +/- 13% concerning streamflow uncertainties for flood should be a bit more explained. To which probability range this uncertainty refers ? Is it one or two standard deviation (or something else) ?

In Figure 2, I would put the number of free parameters to calibrate.

Conclusion : I really appreciated this paper and I would like to congratulate the authors for their work.

---

## Referee Comment (RC2) · Anonymous Referee #2 · 5 Mar 2019

This study compares four structures from the framework FUSE in 1100 UK catchments. This is, in itself, a significant achievement. The authors highlight which structures perform best in different regions (Results Section) and then discuss more generally why models fail and which improvements would be necessary to improve performance (Discussions Section). I think that, ultimately, the goal of such model intercomparison is to provide guidance on i) model selection (i.e., can specific models/modules be recommended based on basin attributes?) and ii) model development (i.e., are there specific process parameterisations that are currently missing, but are needed to improve the

simulations?). In my view, the latter point is addressed quite well (although I suggest restructuring the text to makes these results stand out more, and to go beyond FUSE structures by discussing modelling decisions more generally) but the former point could be addressed in a more systematic and comprehensive way. Overall, I consider that, after revisions, this paper has the potential to become a timely and welcome addition to the literature.

**Major comments**

Model intercomparison vs. benchmarking: Since the authors use the term "benchmarking" in the title and throughout the manuscript, I encourage them to clarify in the introduction what differentiates model benchmarking from model intercomparison. As the authors compare FUSE structures with each other, isn't their study rather a model intercomparison? Do the authors mean that their runs can be used as benchmark by future studies, as suggested on P12L12? Please clarify.

Model evaluation using NSE: Since the authors aim to better understand "where and why these simple models may fail" the choice of NSE is somewhat suprising, since NSE is a measure of overall performance, which provides limited insights into the reasons for high or low performance. Although an evaluation based on hydrological signatures would have enabled a more process-based diagnostic of model failures, I am not requiring this, since it would imply significant additional analyses. However, if the authors stick to NSE (or use KGE), I suggest that they use benchmarks (as suggested by Seibert et al., 2018) to account for the fact that high NSE/KGE values can be relatively easy to reach depending on the catchment and the season. I believe this would enable a more fair and enlightening assessment of the hydrological models across the catchments.

Relevance for the broad hydrological modelling community: A challenge here is to provide guidance for model selection, which is also relevant for modellers not using FUSE. Overall, the most interesting question is not really which FUSE model performs best,

but why. I encourage the authors to discuss and highlight specific model elements that contribute to poor/good simulations, rather than focussing FUSE models themselves (e.g., TOPMODEL or PRMS). For instance, the fact that ARNO-VIC performs particularly well in high-BFI catchments is only an intermediary result, which is mostly relevant to FUSE users. The reasons why this is the case (e.g., last paragraph of Section 5.2), on the other hand, are relevant to a much wider group. I suggest a stronger emphasis on modelling decisions, as opposed to FUSE models, in particular in the most critical parts of the manuscripts (abstract and conclusions).

Which process parameterisation are missing to capture the range of hydrological behaviours across the UK? The authors identify catchments in which the four model structures perform poorly, and reflect on characteristics of these catchments to which the poor performance can be attributed (e.g., chalk, snow, high human impacts). I suggest that the authors dedicate a subsection in the Discussion Section to these findings, which are relevant for both model development and selection. Can they formulate hypotheses on why annual maximum flows are underestimated, which could be tested by future studies?

How critical is the selection of model structure? There are cases of great equifinality (i.e., high NSE for all structures, mostly for humid catchments). As mentioned above, a high NSE is not a guarantee that the model structure is adapted, but as long as this is recognised (and this could be clearer throughout the manuscript), I think it is fine for this study. But in other (more interesting) catchments, some model structures clearly outperform other structures, and there, model choice is critical. I think this should be stressed more prominently, since these are cases in which the inadequacy of the model structure cannot be overcome by parameter tuning. Given the general tendency of using the same model structure across very diverse environments (as discussed e.g. by Addor and Melsen, 2019), I think this is an important result, which could be underscored more. A related question is: which catchment characteristics explain these large NSE differences between model structures?

This leads me to a set of comments related to the use of catchment attributes to explain model performance. Just like hydrological behaviour, model performance is not determined by a single catchment characteristic, but rather, by the interaction of multiple catchment characteristics. So, firstly, would it be possible to consider a wider range of catchment attributes? So far, the authors employ the BFI, annual rainfall, the wetness index and the runoff coefficient, but many more attributes could be used to describe each catchment (e.g., Beck et al., 2015). I encourage the authors to add other attributes, which they might have computed for other studies or retrieved from the UK hydrometric register, which they mention in Table 1, in order to describe the landscape in a more complete fashion (indicators of human interventions would also be useful, see below).

And secondly, I think it would be beneficial to better account for the interactions between these attributes. The authors combine several attributes in Figure 7 to explain model performance, which I find particularly interesting. Maybe that the analyses they will perform when revising this study will lead to more figures of this type, and enable a more systematic analysis of the interactions between these predictors (perhaps using regression trees, see Poncelet et al., 2017). This is critical to go from describing where models fail and to explaining why they fail.

Anthropogenic activities are repeatedly mentioned to explain poor model performance (e.g., P12L29, P14L16, P15L3). This is indeed plausible, but if qualitative or maybe quantitative indicators of the extent of human interventions could be included, so that their impacts on streamflow and model performance could be demonstrated or maybe even quantified, it would strengthen the study.

**Minor comments**

I find the introduction too long. It attempts to cover too much material, and hence ends up being too general and its different parts are not very well connected. I suggest that the authors focus on what is really necessary to introduce their study, transfer parts of

the text to the rest of the paper (e.g. the methods), and delete the rest.

Outlook: it might good to mention that, although this study focusses on four FUSE models, it is possible build additional FUSE model to transition progressively from one model to the next, and establish which modelling decisions contribute most to the differences in the simulations.

Data availability: "This study provides a useful benchmark of the performance and associated uncertainties of four commonly used lumped model structures across GB, for future model developments and model types to be compared against". I agree. But then, I think that instead of saying that "All model outputs from this study are available upon request from the lead author", the authors should make the runs available online, and provide the doi, before the paper is published. This is expected by AGU journals, and I think it is good practice in order to avoid data loss.

**Other suggested changes**

Title: the field is "large-sample hydrology", but here it should be "large sample"
P1L15: add "and support model selection"
P2L13: such as
P2L29: impacted by what?
P4L12-17: this belongs to Data and Methods
P4L20: I suggest removing "(i.e. the number of storage components)" as it an arbitrary measure of complexity.
P5L22: discharge
P6L5: please define "sufficient"
P7L2: I suggest mentioning here that none of these four models includes a snow routine
P7L4: please define "dynamically different" and what makes them "equally plausible"
P8L13: please be more explicit about how this 13P9L21: saying "snowmelt module" implies that accumulation is simulated but melt is not, use "snow module" instead

P10L29: SACRAMENTO

**References**

Addor, N. and Melsen, L. A.: Legacy, rather than adequacy, drives the selection of hydrological models, Water Resour. Res., 55, doi:10.1029/2018WR022958, 2019.

Beck, H. E., de Roo, A. and van Dijk, A. I. J. M.: Global maps of streamflow characteristics based on observations from several thousand catchments, J. Hydrometeorol., 16, 1478–1501, doi:10.1175/JHM-D-14-0155.1, 2015.

Poncelet, C., Merz, R., Merz, B., Parajka, J., Oudin, L., Andréassian, V. and Perrin, C.: Process-based interpretation of conceptual hydrological model performance using a multinational catchment set, Water Resour. Res., 53, 2742–2759, doi:10.1002/2016WR019991, 2017.

Seibert, J., Vis, M., Lewis, E. and van Meerveld, I.: Upper and lower benchmarks in hydrological modeling, Hydrol. Process., 32, 1120–1125, doi:10.1002/hyp.11476, 2018.

---

## Referee Comment (RC3) · Anonymous Referee #3 · 18 Mar 2019

Summary:

This paper provides a detailed investigation into the performance of four lumped conceptual models over large number of catchments in the UK. It demonstrates some very interesting findings, such as the fact that all four models have very similar performance on a catchment-by-catchment basis, and that only one of the models is deemed suitable for catchments with very high BFI. This paper is generally well written, set out and easy to follow, and the graphics provided assist the reader well in the interpretation of the results, I particularly like Figures 5 and 7. The discussion section should be syn-

thesised as it feels repetitive of the results section. Overall, I feel that the motivations of the research, and the implications of the results are not very well reasoned. The authors need to think a bit more carefully about how others may make use of these results, and in particular, should publish the model performance scores as supplementary information (see my comments below).

Major Comments:

1. You've "benchmarked" performance, but you haven't provided these benchmarks. If I were to now go and simulate a UK catchment, I still cannot easily compare my results with yours to see if I have a better model. For you to have achieved your aims, I would expect a supplementary table of the best scores the models achieved in each catchment, and the parameter values that produced them.

2. Section 3.2 – why NSE?

3. Section 3.2 – "results are stored for a number of additional metrics not reported here". Stored where? Why would I care about this if you haven't made them available to me? I suggest you summarise these additional metrics in supplementary information. This may also address the issue of only reporting on NSE here.

4. Your statement in the abstract L23 that NSE scores of 0.72-0.78 were achieved for all catchments is misleading. How useful a measure is the "median maximum NSE for all the catchments"? It's pretty cryptic. There are catchments in E Scotland, and Anglian region that are showing pink/red for all 4 models, so NSE must be <0.5. Having got to page 9 I now see what you meant, but it isn't clearly stated. The sentences on P12 L16-17 are a better summary of the performances across catchments. Same issue on P16 L 32.

5. Catchment characteristics and climate – do all FUSE models maintain the water balance? Can you comment on the existence of models that don't (e.g. GR4J), and how those may overcome such problems? What are the implications of maintaining

vs not maintaining water balance in conceptual lumped models? Are the four models you've chosen actually quite similar to each other? I think you need to make more of this somehow.

6. P8 L23 – only a 1 year warm up period? This is not sufficient for many GW dominated catchments in the SE.

7. P6 L6 – 2 years of data was your criteria for catchment selection, this doesn't seem sufficient to me...

8. Reading through your discussion seems very repetitive of the results chapter. Can these be better synthesised, to reduce the discussion section?

9. P2 L32 "a national scale model" – you're talking about applying a catchment model nationally. Can this be classified a national scale model?

10. P3 L16 - "Secondly, evaluating more complex hydrological models relative to benchmark performance of simple models ensures that the relative difficulty of simulating different catchments is implicitly considered (Seibert et al., 2018)." I don't think I understand what you're saying here.

11. P10 L22-24 – "For very low values of the ARNO-VIC 'b' exponent (AXV_BEXP) as seen for high BFI vales in Fig. 6 for behavioural model distributions means that only at very high, near full upper storage levels is any larger extent of saturated areas predicted" – I don't follow this sentence either.

12. P8 L3 - Can you explain conditional probabilities in more detail?

13. P11 L 23 – "the top row of plots" – there is only one plot in Fig 8!

14. P12 L 7-8 "However, variations between years are less apparent when looking at 25th and 75th percentiles in Fig. 8." We can't distinguish variation between years from Fig 8?

15. Please provide more sensible y axis labels for fig 8 and 9, e.g. "AMAX discharge

score", and "AMAX percentage overlap" respectively. Multiply Fig 9 y axis by 100 to make it an actual percentage value, as you have referred to it as such in the text.

16. Your discussion is longer than the rest of the paper put together!

17. P13 L 3 – you've made no reference to anthropogenic influences in Scotland. This statements seems a bit throwaway.

18. P13 L9 – it is not just the Thames basin that is affected by abstractions! A lot of Anglian region is VERY heavily influenced.

19. P13 L12 "we found that the ensemble of model structures produced better results overall than any single model" – can you validate that statement from your figures?

20. P13 L15 – "The ensemble of model structures was able to take advantage of this" - this seems to be a contradictory argument to the previous statement that the models all have similar performance to each other on a catchment by catchment basis. I think you need to tease these two arguments out better somehow. E.g. in some situations the choice of a different model can yield better results (e.g. high baseflow), but in other situations, none of the models can do well (e.g. abstractions). What are the implications of this?

21. P17 L 11-14 "We also evaluated model predictive capability for high flows, as good model performance in replicating the hydrograph, assessed using Nash-Sutcliffe efficiency, does not necessarily mean models are performing well for other hydrological signatures. We found that the FUSE models tended to underestimate peak flows, and there were variations in model ability between years with models performing particularly poorly for extremely wet years." – so what? What are the potential implications?

Typos and grammar:

1. P2 L27 – CAMELS and MOPEX datasets (what are they datasets of?)

2. P5 L14 - "these" should be "those"
3. P5 L22 - remove "Environment Agency", a catchment is a catchment, the EA don't own the catchments, even if they do own the gauges!

4. Amend "Rainfall is highest in the West and North of GB and lowest in the East and South varying from a minimum of 500mm to a maximum of 4496mm per year (see Fig. 1)" to "On average, rainfall is highest in the north and west of GB, and lowest in the south and east, with GB totals varying from a minimum of 500mm to a maximum of 4496mm per year (see Fig. 1).

5. P6 L1 - remove the "of" after "South-East"

6. P6 L12 – they are the "UK Met Office" not the "UK Meteorological Office".

7. P6 L14 and L20 – replace "laid" with "lay"

8. L6 L21 and elsewhere – "data" is plural, and should be followed by "were" instead of "was"

9. P8 L15 – "observational uncertainty certainty bounds" huh?? Can you not just remove the word certainty here?

10. P9 L15 – you haven't introduced the abbreviation "SAC"

11. P10 L5 – I'd call that northeast Scotland, not central Scotland

12. P11 L29 – "behavioural model" should be "behavioural models"

13. P13 L7 – do you mean model "structures"?

14. P16 L17 – "we also shown how"

15. P16 L19 – refer to Fig 6

16. P16/17 – "The performance of the four models was similar, and all models showed similar spatial patterns of performance, and there was no single model that outperformed the others across all catchment characteristics and for both daily flows and peak flows." – and, and, and.

17. P17 L8 – "we found models performed poorly for catchments for catchments with unaccounted losses"

HESS REVIEW CHECKLIST

1. Does the paper address relevant scientific questions within the scope of HESS? Yes

2. Does the paper present novel concepts, ideas, tools, or data? Yes

3. Are substantial conclusions reached? Nearly, the wider implications, and utility of the research need to be better considered

4. Are the scientific methods and assumptions valid and clearly outlined? Yes

5. Are the results sufficient to support the interpretations and conclusions? Yes

6. Is the description of experiments and calculations sufficiently complete and precise to allow their reproduction by fellow scientists (traceability of results)? Yes

7. Do the authors give proper credit to related work and clearly indicate their own new/original contribution? Yes

8. Does the title clearly reflect the contents of the paper? Yes

9. Does the abstract provide a concise and complete summary? Yes

10. Is the overall presentation well-structured and clear? Yes

11. Is the language fluent and precise? Yes

12. Are mathematical formulae, symbols, abbreviations, and units correctly defined and used? Yes

13. Should any parts of the paper (text, formulae, figures, tables) be clarified, reduced, combined, or eliminated? Yes, the discussion should be reduced

14. Are the number and quality of references appropriate? Yes
15. Is the amount and quality of supplementary material appropriate? No

---

## Author Comment (AC1) · 26 Apr 2019

We thank Thibault Mathevet for taking the time to review our manuscript, and for his helpful comments. Our responses to each individual comment are outlined in bold below.

**Summary**

I carefully read the paper by Lane et al.. This paper appeared to be particularly clear, well written and easy to follow. Scope and objectives are stated clearly, the presentation of results is rather straightforward. As you probably know it, I appreciate this kind of study on a large sample of watersheds. I am very happy to know that such a large sample exists for GB. Studies on large sample give generality and robustness to the results. This paper gives insights on the general hydrology of GB and predictive capabilities of 4 simple rainfall-runoff models. I really appreciated §4 and §5, particularly analyses linked to the seasonality (fig 4), BFI (fig 5, 6), and water balance closure (fig 7). Thanks to this large sample of watersheds in GB with a variety of hydrologic/ hydrogeologic functioning (even in the same country), these results appear to be robusts, with a general interest. The link between BFI (main underground processes) and model structure agility is really interesting.

Response: We thank Thibault Mathevet for taking the time to thoroughly review our manuscript, and for his positive comments.

**Main comments**

Evaluation of model performance and selection of model : Authors decided to use the classical Nash-Sutcliffe efficiency (NSE) index to evaluate model performances (and select behavioural models, NSE > 0.5). NSE index is famous and widely used in Rainfall-Runoff modeling. Even if the perfect efficiency index do not exists, this index is also known to have some drawbacks (Schaefli and Gupta, 2007, among many references). Gupta et al. (2009) introduced the Kling-Gupta efficiency index that allows to explicitly account for bias (mean and variability) and correlation, in the evaluation of model performances. Given the ambition of this paper, I would recommand the
authors to consider in their analyses the Kling-Gupta efficiency index, or at least to decompose their results in terms of correlation and mean bias.

Response: The NSE index was chosen for this analysis as it is so widely used and easy to interpret. Given our focus on floods, it is also a good choice as it emphasizes the fit to peaks more than KGE. However, we agree that there are drawbacks to only using the NSE index and so following this comment, we will provide additional analysis looking at the correlation, variance and mean bias.

Poor performances on floods : Authors found that the different models had poor performances on floods, which is generally the case when classical modeling schemes are used to optimise or select parameter sets. I appreciate the simple way authors evaluate models on flood values, however I would add a figure to explain the two metrics. One of the main drawback of the NSE (and linear regression as well) is that the standard deviation of the simulated time-series is biased and underestimated, i.e. flood underestimated and drought overestimated. Among other arguments, this drawback partly explain why flood values are underestimated. I would add at least a comment on the fact that this statement is dependent on the behavioural model selection metrics in §5.4. If authors update their paper using KGE to select their behavioural models, they might revise (a bit) their findings on model performances for floods.

Response: Thank you for pointing this out. In response to this comment, we will ensure the flood metrics are explained as simply as possible and consider adding a figure to help explain how they were calculated. We selected NSE as it emphasizes the fit to peaks, whilst KGE is more general, but we acknowledge that it has drawbacks and no global performance measure is useful in all situations, especially when looking at extremes. We will therefore keep using NSE to select behavioural models, but as suggested we will comment in the discussion on how behavioural model selection metrics influences estimation of flood values.
Focus on droughts ? : Given the ambition of this paper, I think that this paper would also benefit from a focus on droughts. Hence, analyses on droughts could be complementary to analyses on relative model performances (among the 4 tested structures), since droughts might also be driven by BFI in GB ? The link with groundwater flows could also be shown, if a focus on droughts is done. Authors could use the same metrics as for floods. It could be better to use the 10 days or 30 days annual minimal value, instead of the annual minimal value, which could be highly impacted and uncertain.

Response: We agree that focusing on droughts could be an interesting question in itself, however we feel that it is out of scope for this paper. We are aiming to give a general overview of the capability of models, with a focus on high flows. Drought and very low flows is a more complex problem to address, and more likely to be influenced by human impacts in managed catchments. We therefore think adding this would be too much for one paper.

**Minor comments**

In §1.1 : authors discuss the benefits of national scale hydrological modelling. Another benefits could be the production of parameter libraries, which could be used for regional studies or model calibration on poorly gauged to ungauged basins or engineering studies. Authors can make references to papers on this subject (Perrin et al., 2008; Rojas-Serna et al., 2016; or some other works by Seibert).

Response: Thank you for this idea, we will include discussion of this benefit in the manuscript, and ensure best-performing parameter sets are made available through a DOI or supplementary information.
In §5.1 : authors did not use a snow accumulation and melt routine in their modeling framework. Very simple snow routine are availables, in the spirit of the simple models proposed in FUSE. The CemaNeige routine could be a good candidate to improve model simulations on the few catchments where it's necessary. Depending on the proportion of snow impacted catchments, using a snow routine would improve model performances and the paper, as it could give answers to some hypotheses of the paper. Valéry, A., Andréassian, V., Perrin, C., 2014. 'As simple as possible but not simpler': What is useful in a temperature-based snow-accounting routine? Part 1 – Comparison of six snow accounting routines on 380 catchments, Journal of Hydrology, 517(0): 1166-1175.

Response: Thank you for this comment, but we do not think it would be feasible to run all simulations again with a snow routine. We originally decided not to use a snow routine as only a relatively few catchments were snow impacted. To check this, we have calculated snow fractions for all catchments, as the sum of the rainfall on days when daily mean temperature is less than 0 degrees Celsius divided by the total sum of the rainfall for the whole time period. This confirms that only a small proportion of catchments are snow impacted (13 catchments out of the 1127 have a snow fraction of more than 10%, and no catchments have a snow fraction of more than 17%). As the concept of the paper is focused on benchmarking the capability of these lumped models, and not model development, we feel that addition of a snow routine is out of scope.

In §5.3 : authors discuss about groundwater flows between catchments, with losses or gain of waters. This problem is not new and some conceptual modelisation could be found in the literature since one or two decades. In a natural context, authors could make a reference to Le Moine at al. (2007, 2008) papers about groundwater flows and water balance closure. The existence of such groundwater flows in permeable geological context (chalk, limestones and/or karstic systems, etc.) was one of the reasons of the development of a groundwater exchange function within the GR model
family. The use of this function should be motivated by (hydrogeologic) evidences of such groundwater flows (in order to avoid "overfitting" of the water balance, i.e. fudge factor), but might be useful in catchments where water balance is difficult to close, such as the one influenced by chalk aquifers in southeast england.

Response: Thank you for highlighting these interesting and very relevant papers. We will add this into the discussion section where relevant. However we have not yet done a comprehensive analyses of gaining and losing streams in the UK aquifer systems. This is indeed research that our group is currently conducting in more detail (separate PhD on improving ground water representation in models). Certainly from our preliminary analyses it is very difficult to attribute these losses and gains.

Last comments

P4, I22 : I would also make a reference to Perrin et al. 2001 here **Response: We agree this is a relevant paper, it has been added.**

P7, I20 : mistake with O (mean of observed discharge) Response: Thank you for noticing this, it has been corrected.

P10, l22 : values instead of vales **Response: This has been corrected.**

P17, l8 : for catchments, repeated 2 times **Response: The repetition has been removed.**

HESSD
In §2, I would give an estimation of the proportion of watersheds where snowmelt processes are observable (solid precipitation is more than 20% of total precipitation ?) **Response: We agree that this would be useful and will add a map of snow fractions to figure 1 (as discussed above).**

Table 1 is not cited within §2

Response: Thank you for spotting this, we have now added the citation: "The catchments cover all regions and include a wide variety of catchment characteristics including topography, geology and climate (see Table 1)."

In §3.3, the +/- 13% concerning streamflow uncertainties for flood should be a bit more explained. To which probability range this uncertainty refers ? Is it one or two standard deviation (or something else) ?

Response: The +/-13% represents the 95th percentile range of the discharge uncertainty bounds and was chosen as a representative discharge uncertainty for annual maximum flows from a national analysis of discharge uncertainties (Coxon et al, 2015). We will better clarify this in the text.

In Figure 2, I would put the number of free parameters to calibrate. **Response: These will be added to the figure caption.**

Conclusion : I really appreciated this paper and I would like to congratulate the authors for their work.

Response: We would like to thank Thibault Mathevet for his thorough review, and useful suggestions for the manuscript.

**HESSD**

---

## Author Comment (AC2) · 26 Apr 2019

We thank reviewer 2 for taking the time to review our manuscript, and provide useful feedback. Our responses to each review comment are given in bold below.

Summary

This study compares four structures from the framework FUSE in 1100 UK catchments. This is, in itself, a significant achievement. The authors highlight which structures perform best in different regions (Results Section) and then discuss more generally why models fail and which improvements would be necessary to improve performance (Discussions Section). I think that, ultimately, the goal of such model intercomparison is to provide guidance on i) model selection (i.e., can specific models/modules be recommended based on basin attributes?) and ii) model development (i.e., are there specific process parameterisations that are currently missing, but are needed to improve the simulations?). In my view, the latter point is addressed quite well (although I suggest restructuring the text to makes these results stand out more, and to go beyond FUSE structures by discussing modelling decisions more generally) but the former point could be addressed in a more systematic and comprehensive way. Overall, I consider that, after revisions, this paper has the potential to become a timely and welcome addition to the literature.

**Response: We thank reviewer 2 for these helpful comments, and for taking the time to review our manuscript.**

Main comments

Model intercomparison vs. benchmarking: Since the authors use the term "benchmarking" in the title and throughout the manuscript, I encourage them to clarify in the introduction what differentiates model benchmarking from model intercomparison. As the authors compare FUSE structures with each other, isn't their study rather a model intercomparison? Do the authors mean that their runs can be used as benchmark by future studies, as suggested on P12L12? Please clarify.

[Figure]

**Response: We agree that this could be made clearer, and will include clarification of this in the introduction. We used the term 'benchmark' to highlight that these results can be used as an indicator of the ability of lumped models, which future studies may use when evaluating the performance of other models (that are perhaps more complex or include additional processes). For example, our results would inform a modeller that gaining an NSE of 0.7 in SE England is a good achievement, whereas gaining the same score in west Wales is not an achievement as most models can easily gain higher NSE scores for these catchments. The use of simple models as benchmarks has been advocated in previous studies, for example Seibert et al., (2018).**
**Seibert, J., Vis, M. J., Lewis, E., & Meerveld, H. J. (2018). Upper and lower benchmarks in hydrological modelling. Hydrological Processes.**

Model evaluation using NSE: Since the authors aim to better understand "where and why these simple models may fail" the choice of NSE is somewhat suprising, since NSE is a measure of overall performance, which provides limited insights into the reasons for high or low performance. Although an evaluation based on hydrological signatures would have enabled a more process-based diagnostic of model failures, I am not requiring this, since it would imply significant additional analyses. However, if the authors stick to NSE (or use KGE), I suggest that they use benchmarks (as suggested by Seibert et al., 2018) to account for the fact that high NSE/KGE values can be relatively easy to reach depending on the catchment and the season. I believe this would enable a more fair and enlightening assessment of the hydrological models across the catchments.
**Response: We originally selected NSE as it is a widely used and easy to interpret measure of performance. However, we agree that in order to better understand model failures we will need to consider additional measures of performance. Therefore, we plan to also present correlation, variance and mean bias, as called for by the first reviewer, to support the seasonal analysis of model performance**

that we have already carried out. As our focus is on reasons for model failures, we feel that these additional decomposed metrics will be more informative than the use of benchmarks.

Relevance for the broad hydrological modelling community: A challenge here is to provide guidance for model selection, which is also relevant for modellers not using FUSE. Overall, the most interesting question is not really which FUSE model performs best, but why. I encourage the authors to discuss and highlight specific model elements that contribute to poor/good simulations, rather than focussing FUSE models themselves (e.g., TOPMODEL or PRMS). For instance, the fact that ARNO-VIC performs particularly well in high-BFI catchments is only an intermediary result, which is mostly relevant to FUSE users. The reasons why this is the case (e.g., last paragraph of Section 5.2), on the other hand, are relevant to a much wider group. I suggest a stronger emphasis on modelling decisions, as opposed to FUSE models, in particular in the most critical parts of the manuscripts (abstract and conclusions).

**Response: We agree that highlighting specific model elements that contribute to poor/good simulations would be of great use to the broad hydrological modelling community. To address this, we will explain in more detail how the modelling decisions differ between the four structures in the methods section. We will also clarify these different modelling decisions when discussing model performance in the abstract and conclusion However, as we have looked at the four parent models within the FUSE framework, and not all possible combinations of modelling decisions which would be very challenging over such a large sample of catchments, there is a limit to how far we can analyse which particular model elements contribute to good/poor results. Many elements differ between the model structures, and it is not always possible to decipher which individual element is responsible.**

Which process parameterisation are missing to capture the range of hydrological behaviours across the UK? The authors identify catchments in which the four model structures perform poorly, and reflect on characteristics of these catchments to which the poor performance can be attributed (e.g., chalk, snow, high human impacts). I suggest that the authors dedicate a subsection in the Discussion Section to these findings, which are relevant for both model development and selection. Can they formulate hypotheses on why annual maximum flows are underestimated, which could be tested by future studies?

**Response: Findings on which process parameterisations are missing to capture the range of hydrological behaviours are discussed in section 5.1. However, we agree that these should be made easier to find, and therefore we will split discussion section 5.1 into "Benchmarking performance of multiple lumped hydrological models across GB" and "Identifying missing process parameterisations for modelling in Great Britain". As suggested by the first reviewer, the choice of NSE could result in underestimation of flood peaks, and we will therefore comment in the discussion on how our choice of metrics could be a factor leading to the underestimation of flood values.**

How critical is the selection of model structure? There are cases of great equifinality (i.e., high NSE for all structures, mostly for humid catchments). As mentioned above, a high NSE is not a guarantee that the model structure is adapted, but as long as this is recognised (and this could be clearer throughout the manuscript), I think it is fine for this study. But in other (more interesting) catchments, some model structures clearly outperform other structures, and there, model choice is critical. I think this should be stressed more prominently, since these are cases in which the inadequacy of the model structure cannot be overcome by parameter tuning. Given the general tendency of using the same model structure across very diverse environments (as discussed e.g. by Addor and Melsen, 2019), I think this is an important result, which could be underscored more. A related question is: which catchment characteristics explain

these large NSE differences between model structures?

**Response: We explored the importance of model structure selection in figures 3, 5, 8 and 9, with figure 5 looking at catchment characteristics which were related to differences between the model structures. However, we agree that the question of how critical the selection of model structure is for different catchments is not well addressed in the manuscript. Therefore, we will add a short paragraph discussing this into section "5.2 Insights from applying an ensemble of model structures across a large sample of catchments" and make this key finding clearer in the abstract and conclusions of the paper.**

This leads me to a set of comments related to the use of catchment attributes to explain model performance. Just like hydrological behaviour, model performance is not determined by a single catchment characteristic, but rather, by the interaction of multiple catchment characteristics. So, firstly, would it be possible to consider a wider range of catchment attributes? So far, the authors employ the BFI, annual rainfall, the wetness index and the runoff coefficient, but many more attributes could be used to describe each catchment (e.g., Beck et al., 2015). I encourage the authors to add other attributes, which they might have computed for other studies or retrieved from the UK hydrometric register, which they mention in Table 1, in order to describe the landscape in a more complete fashion (indicators of human interventions would also be useful, see below).

And secondly, I think it would be beneficial to better account for the interactions between these attributes. The authors combine several attributes in Figure 7 to explain model performance, which I find particularly interesting. Maybe that the analyses they will perform when revising this study will lead to more figures of this type, and enable a more systematic analysis of the interactions between these predictors (perhaps using regression trees, see Poncelet et al., 2017). This is critical to go from describing where models fail and to explaining why they fail.

**Response: We selected the attributes of BFI, annual rainfall, wetness index**

**and runoff coefficient as they were observed to have the largest impact on model performance. However, it is possible to include additional attributes, to further explain model performance. We do not want to add more figures into the manuscript as we feel that this may detract from the main messages of the paper. However, we will produce results for a wider variety of attributes and include them as supplementary information. These will be in the form of scatter plots, showing interactions between different characteristics as suggested, in the same style as figure 7.**

Anthropogenic activities are repeatedly mentioned to explain poor model performance (e.g., P12L29, P14L16, P15L3). This is indeed plausible, but if qualitative or maybe quantitative indicators of the extent of human interventions could be included, so that their impacts on streamflow and model performance could be demonstrated or maybe even quantified, it would strengthen the study.

**Response: We agree that this is required to strengthen comments made regarding reasons for model failures. We have information on factors affecting runoff for all catchments in the hydrometric register. However, this only gives an indicator of which factors may affect runoff, and not to what extent, and therefore we decided not to include it in the original manuscript. In response to reviewer comments, we will investigate either a) providing maps of this information to assist in the interpretation of the results or b) Incorporate a summary of this information into figure 7 through changing the marker shape.**

Minor comments

I find the introduction too long. It attempts to cover too much material, and hence ends up being too general and its different parts are not very well connected. I suggest that

the authors focus on what is really necessary to introduce their study, transfer parts of the text to the rest of the paper (e.g. the methods), and delete the rest.
**Response: We agree that the introduction could be more concise and will shorten it in the revised manuscript of the paper.**

Outlook: it might good to mention that, although this study focusses on four FUSE models, it is possible build additional FUSE model to transition progressively from one model to the next, and establish which modelling decisions contribute most to the differences in the simulations.
**Response: this will be added.**

Data availability: "This study provides a useful benchmark of the performance and associated uncertainties of four commonly used lumped model structures across GB, for future model developments and model types to be compared against". I agree. But then, I think that instead of saying that "All model outputs from this study are available upon request from the lead author", the authors should make the runs available online, and provide the doi, before the paper is published. This is expected by AGU journals, and I think it is good practice in order to avoid data loss.
**Response: We completely agree with this, and are currently obtaining a DOI for the data. As also recommended by reviewer 3, we will add a table summarising the results for each catchment to be made available as supplementary information.**

Other suggested changes

Title: the field is "large-sample hydrology", but here it should be "large sample"
**Response: Thank you, this has been changed.**

P1L15: add "and support model selection"
**Response: This has been added.**

P2L13: such as **Response: Thank you for spotting this, it has been changed.**

P2L29: impacted by what?
**Response: We have clarified the sentence by adding "impacted by future changes to the hydrological regime."**

P4L12-17: this belongs to Data and Methods
**Response: These sentences have been moved to the data and methods section.**

P4L20: I suggest removing "(i.e. the number of storage components)" as it an arbitrary measure of complexity.
**Response: This has been removed.**

P5L22: discharge
**Response: We have clarified that it is a discharge data set.**

P6L5: please define "sufficient"
**Response: This was explained in the following sentence. We have re-arranged these sentences to make this clearer, now saying "Of these, 1128 had sufficient information (more than 2 years of discharge data available within the time period) available to include in this analysis."**

P7L2: I suggest mentioning here that none of these four models includes a snow

Routine

**Response: We have added this, "They all close the water balance and include the same processes, for example none of the models have a snow routine."**

P7L4: please define "dynamically different" and what makes them "equally plausible"

**Response: By "dynamically different" we meant that the models all represent the landscape in a different way, and have quite different and distinct structures as shown in figure 2. By "equally plausible" we are referring to the fact that we have no reason to expect one structure to behave better than the others, as all model structures are equally complete in terms of processes and all based on widely applied model structures. We will clarify this in the text.**

P8L13: please be more explicit about how this 13

**Response: We will include a better description of how this +/- 13% was calculated (see response to reviewer 1).**

P9L21: saying "snowmelt module" implies that accumulation is simulated but melt is not, use "snow module" instead.

**Response: We agree, and have changed "snowmelt module" to "snow module."**

---

## Author Comment (AC3) · 26 Apr 2019

We thank reviewer 3 for taking the time to review our manuscript, and provide useful feedback. Our responses to each comment are given in bold below.

Summary

This paper provides a detailed investigation into the performance of four lumped conceptual models over large number of catchments in the UK. It demonstrates some very interesting findings, such as the fact that all four models have very similar performance on a catchment-by-catchment basis, and that only one of the models is deemed suitable for catchments with very high BFI. This paper is generally well written, set out and easy to follow, and the graphics provided assist the reader well in the interpretation of the results, I particularly like Figures 5 and 7. The discussion section should be synthesised as it feels repetitive of the results section. Overall, I feel that the motivations of the research, and the implications of the results are not very well reasoned. The authors need to think a bit more carefully about how others may make use of these results, and in particular, should publish the model performance scores as supplementary information (see my comments below).

**Response: We would like to thank the reviewer for taking the time to read the paper in depth, and for their constructive comments.**

Main Comments

1. You've "benchmarked" performance, but you haven't provided these benchmarks. If I were to now go and simulate a UK catchment, I still cannot easily compare my results with yours to see if I have a better model. For you to have achieved your aims, I would expect a supplementary table of the best scores the models achieved in each catchment, and the parameter values that produced them.
**Response: We completely agree with this and are currently obtaining a DOI for the data. We will also add a table summarising the results for each catchment to be made available as supplementary information (as also discussed in response**

to reviewer 2).

2. Section 3.2 – why NSE?
**Response: We originally selected NSE as it is a widely used and easy to interpret measure of performance. However, as noted by the other reviewers, in order to better understand model failures we will consider additional metrics. Therefore, we plan to also present correlation and mean bias.**

3. Section 3.2 – "results are stored for a number of additional metrics not reported here". Stored where? Why would I care about this if you haven't made them available to me? I suggest you summarise these additional metrics in supplementary information. This may also address the issue of only reporting on NSE here.
**Response: We will follow this suggestion, and summarise results from additional metrics in supplementary information and also make these fully available in the open source database of simulations .**

4. Your statement in the abstract L23 that NSE scores of 0.72-0.78 were achieved for all catchments is misleading. How useful a measure is the "median maximum NSE for all the catchments"? It's pretty cryptic. There are catchments in E Scotland, and Anglian region that are showing pink/red for all 4 models, so NSE must be <0.5. Having got to page 9 I now see what you meant, but it isn't clearly stated. The sentences on P12 L16-17 are a better summary of the performances across catchments. Same issue on P16 L 32.
**Response: Thank you for pointing out that this is not clear. We have replaced the statement in abstract L23 with "Our results show that simple, lumped hydrological models were able to produce adequate simulations across most of Great Britain, with each model producing simulations exceeding 0.5 Nash Sutcliffe efficiency over at least 80% of catchments."**

5. Catchment characteristics and climate – do all FUSE models maintain the water balance? Can you comment on the existence of models that don't (e.g. GR4J), and how those may overcome such problems? What are the implications of maintaining vs not maintaining water balance in conceptual lumped models? Are the four models you've chosen actually quite similar to each other? I think you need to make more of this somehow.

**Response: Yes, all the FUSE models used in this study maintain the water balance. To address these questions we will add a paragraph in the discussion on how models that do not maintain the water balance have been used to improve modelling in groundwater dominated regions. In response to reviewer 1, this will include discussion of papers by Le Moine at al. (2007, 2008) about groundwater flows and water balance closure.**

6. P8 L23 – only a 1 year warm up period? This is not sufficient for many GW dominated catchments in the SE.

**Response: Thank you for this advice. We selected 1 year, as it is often considered sufficient for simple, lumped models such as the FUSE models. However, following this comment we carried out additional analysis of the simulated flows and found that whilst 1 year is a long enough warmup period for many catchments, it did not appear sufficient for some of the catchments in the SE as suggested. We will therefore increase the warmup period and re-analyse the data.**

7. P6 L6 – 2 years of data was your criteria for catchment selection, this doesn't seem sufficient to me

**Response: We originally aimed to keep as many catchments as possible for the analysis. However, you are correct that 2 years of data is not long for model**

evaluation. By increasing this threshold to 5, 10 or 15 years of data, we would lose 24, 83, and 155 catchments respectively. We will therefore implement a stricter threshold of 5 years.

8. Reading through your discussion seems very repetitive of the results chapter. Can these be better synthesised, to reduce the discussion section?
16. Your discussion is longer than the rest of the paper put together!
**Response: We agree the discussion is very long. We will make this section more concise, and remove repetitions of the results in the revised manuscript.**

9. P2 L32 "a national scale model" – you're talking about applying a catchment model nationally. Can this be classified a national scale model?
**Response: In this section we were aiming to discuss the importance of national scale modelling more generally, suggesting that our work could be informative for evaluation of a national scale model. We were not saying that our application of a catchment model across GB was a national scale model. We will ensure this is clarified in the text.**

10. P3 L16 - "Secondly, evaluating more complex hydrological models relative to benchmark performance of simple models ensures that the relative difficulty of simulating different catchments is implicitly considered (Seibert et al., 2018)." I don't think I understand what you're saying here.
**Response: This has been re-phrased to make the meaning clearer. It now reads: "Secondly, lumped hydrological models provide a particularly good benchmark for evaluating more complex models, as they give an indication of what it is possible to achieve given the particulars of a catchment and the available data (Seibert et al., 2018). This can help us identify whether a model is performing well in a catchment relative to how it should be expected to perform for the**

**particulars of that catchment."**

11. P10 L22-24 – "For very low values of the ARNO-VIC 'b' exponent (AXV_BEXP) as seen for high BFI vales in Fig. 6 for behavioural model distributions means that only at very high, near full upper storage levels is any larger extent of saturated areas predicted" – I don't follow this sentence either.
**Response: This has been re-phrased.**

12. P8 L3 - Can you explain conditional probabilities in more detail?
**Response: Yes, this will be added.**

13. P11 L 23 – "the top row of plots" – there is only one plot in Fig 8!
**Response: Thank you for noticing that! We had originally displayed figures 8 and 9 as a single plot. This has been corrected.**

14. P12 L 7-8 "However, variations between years are less apparent when looking at 25th and 75th percentiles in Fig. 8." We can't distinguish variation between years from Fig 8?
**Response: Again, thank you for noticing this. We have corrected it to Fig. 9.**

15. Please provide more sensible y axis labels for fig 8 and 9, e.g. "AMAX discharge score", and "AMAX percentage overlap" respectively. Multiply Fig 9 y axis by 100 to make it an actual percentage value, as you have referred to it as such in the text.
**Response: We agree with this comment and will change the figure axis and labels.**

17. P13 L 3 – you've made no reference to anthropogenic influences in Scotland. This statements seems a bit throwaway.
**Response: We will add a plot on factors affecting river flows to support this statement.**

18. P13 L9 – it is not just the Thames basin that is affected by abstractions! A lot of Anglian region is VERY heavily influenced.
**Response: we have changed this sentence to "a considerable proportion of river discharges throughout the Thames and Anglian region are abstracted."**

19. P13 L12 "we found that the ensemble of model structures produced better results overall than any single model" – can you validate that statement from your figures?
**Response: This can not be directly validated in a specific figure, but it can be seen across the figures, especially looking at Figure 5, where we see that no single model produces good results for all catchments.**

20. P13 L15 – "The ensemble of model structures was able to take advantage of this" - this seems to be a contradictory argument to the previous statement that the models all have similar performance to each other on a catchment by catchment basis. I think you need to tease these two arguments out better somehow. E.g. in some situations the choice of a different model can yield better results (e.g. high baseflow), but in other situations, none of the models can do well (e.g. abstractions). What are the implications of this?
**Response: We will clarify these arguments in the discussion.**

21. P17 L 11-14 "We also evaluated model predictive capability for high flows, as good model performance in replicating the hydrograph, assessed using Nash-Sutcliffe

efficiency, does not necessarily mean models are performing well for other hydrological signatures. We found that the FUSE models tended to underestimate peak flows, and there were variations in model ability between years with models performing particularly poorly for extremely wet years." – so what? What are the potential implications?

**Response: We will add discussion about the implications for flood modelling and forecasting here.**

Typos and grammar

1. P2 L27 – CAMELS and MOPEX datasets (what are they datasets of?)
**Response: This sentence has been clarified - "the CAMELS or MOPEX hydrom-eteorological and catchment attribute datasets."**

3. P5 L22 - remove "Environment Agency", a catchment is a catchment, the EA don't own the catchments, even if they do own the gauges!
**Response: "Environment Agency" has been removed.**

4. Amend "Rainfall is highest in the West and North of GB and lowest in the East and South varying from a minimum of 500mm to a maximum of 4496mm per year (see Fig. 1)" to "On average, rainfall is highest in the north and west of GB, and lowest in the south and east, with GB totals varying from a minimum of 500mm to a maximum of 4496mm per year (see Fig. 1).
**Response: Thank you for the suggestion, this sentence has been amended.**

2. P5 L14 - "these" should be "those" 5. P6 L1 - remove the "of" after "South-East"
6. P6 L12 – they are the "UK Met Office" not the "UK Meteorological Office". 7. P6 L14 and L20 – replace "laid" with "lay" 8. L6 L21 and elsewhere – "data" is plural, and should be followed by "were" instead of "was" 9. P8 L15 – "observational uncertainty certainty bounds" huh?? Can you not just remove the word certainty here? 10. P9 L15 – you haven't introduced the abbreviation "SAC" 11. P10 L5 – I'd call that northeast Scotland, not central Scotland 12. P11 L29 – "behavioural model" should be "behavioural models" 13. P13 L7 – do you mean model "structures"? 14. P16 L17 – "we also shown how" 15. P16 L19 – refer to Fig 6
**Response: Thank you for spotting these typos and grammatical errors, we have corrected these in the manuscript.**

16. P16/17 – "The performance of the four models was similar, and all models showed similar spatial patterns of performance, and there was no single model that outperformed the others across all catchment characteristics and for both daily flows and peak flows." – and, and, and
**Response: This sentence has been improved to "The performance of the four models was similar, with all models showing similar spatial patterns of performance, and no single model outperforming the others across all catchment characteristics for both daily flows and peak flows."**

17. P17 L8 – "we found models performed poorly for catchments for catchments with unaccounted losses"
**Response: we have removed the repetition.**

**HESS REVIEW CHECKLIST**

1. Does the paper address relevant scientific questions within the scope of HESS? Yes
2. Does the paper present novel concepts, ideas, tools, or data? Yes 3. Are substantial conclusions reached? Nearly, the wider implications, and utility of the research need to be better considered 4. Are the scientific methods and assumptions valid and clearly outlined? Yes 5. Are the results sufficient to support the interpretations and conclusions? Yes 6. Is the description of experiments and calculations sufficiently complete and precise to allow their reproduction by fellow scientists (traceability of results)? Yes 7. Do the authors give proper credit to related work and clearly indicate their own new/original contribution? Yes 8. Does the title clearly reflect the contents of the paper? Yes 9. Does the abstract provide a concise and complete summary? Yes 10. Is the overall presentation well-structured and clear? Yes 11. Is the language fluent and precise? Yes 12. Are mathematical formulae, symbols, abbreviations, and units correctly defined and used? Yes 13. Should any parts of the paper (text, formulae, figures, tables) be clarified, reduced, combined, or eliminated? Yes, the discussion should be reduced 14. Are the number and quality of references appropriate? Yes 15. Is the amount and quality of supplementary material appropriate? No

**Response: Thank you for this largely positive summary checklist. Our plans to address points 3, 13 and 15 are outlined in the response to individual points above, and the general response to reviewers.**

---

## Author Comment (AC4) · 26 Apr 2019

We thank the reviewers for taking the time to read the manuscript, and for their thorough and insightful comments. Their suggestions have helped us to ensure our results are more useful to the modelling community.

The main comments from the reviewers were regarding (1) the use of Nash-Sutcliffe Efficiency to evaluate model performance, (2) making data more easily accessible,

(3) synthesis of the introduction and discussion sections, and (4) plotting of additional catchment attributes and human influences on river flows.

In response to these reviewer comments, we plan to re-analyse the model output with consideration of additional performance metrics. We will ensure that a summary table of the key results is given as supplementary information, and that a more complete set of outputs is made available via a DOI. We will produce additional plots of factors affecting runoff across Great Britain, highlighting where streamflow is impacted by snowmelt and human influences. The manuscript will also be revised, to synthesize the introduction and discussion.

---

## Author Response (AR1)

**Benchmarking the predictive capability of hydrological models for river flow and flood peak predictions across a large sample of catchments in Great Britain**

**General response to reviewers**

We thank the reviewers for taking the time to read the manuscript, and for their thorough and insightful comments. Their suggestions have helped us to ensure our results are more useful to the modelling community.

The main comments from the reviewers were regarding (1) the use of Nash-Sutcliffe Efficiency to evaluate model performance, (2) making data more easily accessible, (3) synthesis of the introduction and discussion sections, and (4) plotting of additional catchment attributes and human influences on river flows.

In response to these reviewer comments, we have re-analysed the model output with consideration of additional performance metrics. We have supplied a complete set of outputs via a DOI. We have produced additional plots of factors affecting runoff across Great Britain, highlighting where streamflow is impacted by snowmelt and human influences in Figures 1 and 2. We have considered additional catchment attributes in supplementary information. The manuscript has also been revised, to synthesize the introduction and discussion, and to discuss the new metrics and factors affecting runoff plots in the methods and results sections.

Detailed responses to all reviewer comments are provided in bold below. We have also inserted the review comments as comments next to the relevant tracked changes in the manuscript.

Rosie Lane, June 2019

**Response to reviewer 1 (Thibault Mathevet)**

We thank Thibault Mathevet for taking the time to review our manuscript, and for his helpful comments. Our responses to each comment are outlined in bold below.

I carefully read the paper by Lane et al.. This paper appeared to be particularly clear, well written and easy to follow. Scope and objectives are stated clearly, the presentation of results is rather straightforward. As you probably know it, I appreciate this kind of study on a large sample of watersheds. I am very happy to know that such a large sample exists for GB. Studies on large sample give generality and robustness to the results. This paper gives insights on the general hydrology of GB and

10   predictive capabilities of 4 simple rainfall-runoff models. I really appreciated §4 and §5, particularly analyses linked to the seasonality (fig 4), BFI (fig 5, 6), and water balance closure (fig 7). Thanks to this large sample of watersheds in GB with a variety of hydrologic/ hydrogeologic functioning (even in the same country), these results appear to be robusts, with a general interest. The link between BFI (main underground processes) and model structure agility is really interesting.
**Response: We thank Thibault Mathevet for taking the time to thoroughly review our manuscript, and for his positive**

15   **comments.**

**Main comments:**

Evaluation of model performance and selection of model :

20   Authors decided to use the classical Nash-Sutcliffe efficiency (NSE) index to evaluate model performances (and select behavioural models, NSE > 0.5). NSE index is famous and widely used in Rainfall-Runoff modeling. Even if the perfect efficiency index do not exists, this index is also known to have some drawbacks (Schaefli and Gupta, 2007, among many references). Gupta et al. (2009) introduced the Kling-Gupta efficiency index that allows to explicitly account for bias (mean and variability) and correlation, in the evaluation of model performances. Given the ambition of this paper, I would recommend

25   the authors to consider in their analyses the Kling-Gupta efficiency index, or at least to decompose their results in terms of correlation and mean bias.
**Response: The NSE index was chosen for this analysis as it is so widely used and easy to interpret. Given our focus on floods, it is also a good choice as it emphasizes the fit to peaks more than KGE which focuses on balancing the contribution of the bias and correlation. However, we agree that there are drawbacks to only using the NSE index and**

30   **so following this comment, we have provided additional analysis looking at the correlation, variance and bias. This can be seen in Figures 5 and 9.**

Poor performances on floods :

Authors found that the different models had poor performances on floods, which is generally the case when classical modeling schemes are used to optimise or select parameter sets. I appreciate the simple way authors evaluate models on flood values, however I would add a figure to explain the two metrics. One of the main drawback of the NSE (and linear regression as well) is that the standard deviation of the simulated time-series is biased and underestimated, i.e. flood underestimated and drought overestimated. Among other arguments, this drawback partly explain why flood values are underestimated. I would add at least a comment on the fact that this statement is dependent on the behavioural model selection metrics in §5.4. If authors update their paper using KGE to select their behavioural models, they might revise (a bit) their findings on model performances for floods.

**Response: Thank you for pointing this out. In response to this comment, we have clarified the explanation of flood metrics. We selected NSE as it emphasizes the fit to peaks, whilst KGE is more general, but we acknowledge that it has drawbacks and no global performance measure is useful in all situations, especially when looking at extremes. We therefore decided to keep using NSE, but added a comment in the discussion on how behavioural model selection metrics influence estimation of flood values.**

Focus on droughts ? :

Given the ambition of this paper, I think that this paper would also benefit from a focus on droughts. Hence, analyses on droughts could be complementary to analyses on relative model performances (among the 4 tested structures), since droughts might also be driven by BFI in GB ? The link with groundwater flows could also be shown, if a focus on droughts is done. Authors could use the same metrics as for floods. It could be better to use the 10 days or 30 days annual minimal value, instead of the annual minimal value, which could be highly impacted and uncertain.

**Response: We agree that focusing on droughts could be an interesting question in itself, however we feel that it is out of scope for this paper. We are aiming to give a general overview of the capability of models, with a focus on high flows. Drought and very low flows is a more complex problem to address, and more likely to be influenced by human impacts in managed catchments. We therefore think adding this would be too much for one paper. We plan further research which better incorporates human influences on low flow river totals and thus will make such an assessment more fruitful.**

**Minor comments :**

In §1.1 : authors discuss the benefits of national scale hydrological modelling. Another benefits could be the production of parameter libraries, which could be used for regional studies or model calibration on poorly gauged to ungauged basins or engineering studies. Authors can make references to papers on this subject (Perrin et al., 2008 ; Rojas-Serna et al., 2016 ; or some other works by Seibert).

**Response: Thank you for this idea, we have referred to parameter libraries in the introduction, and have added tables of best parameter sets made available through a DOI.**

In §5.1 : authors did not use a snow accumulation and melt routine in their modeling framework. Very simple snow routine are available, in the spirit of the simple models proposed in FUSE. The CemaNeige routine could be a good candidate to improve model simulations on the few catchments where it's necessary. Depending on the proportion of snow impacted catchments, using a snow routine would improve model performances and the paper, as it could give answers to some hypotheses of the paper.

Valéry, A., Andréassian, V., Perrin, C., 2014. 'As simple as possible but not simpler': What is useful in a temperature-based snow-accounting routine? Part 1 – Comparison of six snow accounting routines on 380 catchments, Journal of Hydrology, 517(0): 1166-1175.

**Response: Thank you for this comment, but we do not think it would be feasible to run all simulations again with a snow routine. We originally decided not to use a snow routine as only a relatively few catchments were snow impacted. To check this, we have calculated snow fractions for all catchments, as the sum of the rainfall on days when daily mean temperature is less than 0 degrees Celsius divided by the total sum of the rainfall for the whole time period. This confirms that only a small proportion of catchments are snow impacted (13 catchments out of the 1127 have a snow fraction of more than 10%, and no catchments have a snow fraction of more than 17%). We have plotted these snow fractions in Figure 1, demonstrating that only a small proportion of catchments are impacted by snow. As the concept of the paper is focused on benchmarking the capability of these lumped models, and not model development, we feel that addition of a snow routine is out of scope.**

In §5.3 : authors discuss about groundwater flows between catchments, with losses or gain of waters. This problem is not new and some conceptual modelisation could be found in the literature since one or two decades. In a natural context, authors could make a reference to Le Moine at al. (2007, 2008) papers about groundwater flows and water balance closure. The existence of such groundwater flows in permeable geological context (chalk, limestones and/or karstic systems, etc.) was one of the reasons of the development of a groundwater exchange function within the GR model family. The use of this function should be motivated by (hydrogeologic) evidences of such groundwater flows (in order to avoid "overfitting" of the water balance, i.e. fudge factor), but might be useful in catchments where water balance is difficult to close, such as the one influenced by chalk aquifers in southeast england.

**Response: Thank you for highlighting these interesting and very relevant papers. We have added this into the discussion. However we have not yet done a comprehensive analyses of gaining and losing streams in the UK aquifer systems. This is indeed research that our group is currently conducting in more detail (separate PhD on improving ground water representation in models). Certainly from our preliminary analyses it is very difficult to attribute these losses and gains, and especially for lumped catchment model behaviours where the spatial partitioning needed might be too abstract to incorporate in the model outputs.**

**Last comments :**

P4, l22 : I would also make a reference to Perrin et al. 2001 here

**Response: We agree this is a relevant paper, it has been added.**

5  P7, l20 : mistake with O (mean of observed discharge)

**Response: Thank you for noticing this, it has been corrected.**

P10, l22 : values instead of vales

**Response: This has been corrected.**

P17, l8 : for catchments, repeated 2 times

**Response: The repetition has been removed.**

In §2, I would give an estimation of the proportion of watersheds where snowmelt processes are observable (solid precipitation

15  >20% of total precipitation ?)

**Response: We agree that this would be useful and have added a map of snow fractions to figure 2 which we refer to in section 2.**

Table 1 is not cited within §2

20  **Response: Thank you for spotting this, we have now added the citation: "The catchments cover all regions and include a wide variety of catchment characteristics including topography, geology and climate (see Table 1)."**

In §3.3, the +/- 13% concerning streamflow uncertainties for flood should be a bit more explained. To which probability range this uncertainty refers ? Is it one or two standard deviation (or something else) ?

25  **Response: The +/-13% represents the 95th percentile range of the discharge uncertainty bounds and was chosen as a representative discharge uncertainty for annual maximum flows from a national analysis of discharge uncertainties (Coxon et al, 2015). We have better clarified this, with the text now reading "This observed error value was selected following previous research on quantifying discharge uncertainty at 500 UK gauging stations for high flows, and represents the average 95$^{th}$ percentile range of the discharge uncertainty bounds for high flows (Coxon et al., 2015;**

30  **Mcmillan et al., 2012)."**

In Figure 2, I would put the number of free parameters to calibrate.

**Response: The following has been added to the figure caption "TOPMODEL and ARNO/VIC have 10 parameters, PRMS has 11 parameters and SACRAMENTO has 12 parameters. ".**

**Response to reviewer 2 (Anonymous)**

Comment 1: This study compares four structures from the framework FUSE in 1100 UK catchments. This is, in itself, a significant achievement. The authors highlight which structures perform best in different regions (Results Section) and then discuss more generally why models fail and which improvements would be necessary to improve performance (Discussions Section). I think that, ultimately, the goal of such model intercomparison is to provide guidance on i) model selection (i.e., can specific models/modules be recommended based on basin attributes?) and ii) model development (i.e., are there specific process parameterisations that are currently missing, but are needed to improve the simulations?). In my view, the latter point is addressed quite well (although I suggest restructuring the text to makes these results stand out more, and to go beyond FUSE structures by discussing modelling decisions more generally) but the former point could be addressed in a more systematic and comprehensive way. Overall, I consider that, after revisions, this paper has the potential to become a timely and welcome addition to the literature.

**Response: We thank reviewer 2 for these helpful comments, and for taking the time to review our manuscript.**

**Major comments:**

Comment 2: Model intercomparison vs. benchmarking: Since the authors use the term "benchmarking" in the title and throughout the manuscript, I encourage them to clarify in the introduction what differentiates model benchmarking from model intercomparison. As the authors compare FUSE structures with each other, isn't their study rather a model intercomparison? Do the authors mean that their runs can be used as benchmark by future studies, as suggested on P12L12? Please clarify.

**Response: We have included clarification of this in the introduction, including more explanation on how the performance of simple hydrological models can be used as a benchmark and making it clear that our results can be used as a benchmark. We used the term 'benchmark' to highlight that these results can be used as an indicator of the ability of lumped models, which future studies may use when evaluating the performance of other models (that are perhaps more complex or include additional processes). For example, our results would inform a modeller that gaining an NSE of 0.7 in SE England is a good achievement, whereas gaining the same score in west Wales is not an achievement as most models can easily gain higher NSE scores for these catchments. The use of simple models as benchmarks has been advocated in previous studies, for example Seibert et al., (2018).**

**Seibert, J., Vis, M. J., Lewis, E., & Meerveld, H. J. (2018). Upper and lower benchmarks in hydrological modelling. Hydrological Processes.**

Comment 3: Model evaluation using NSE: Since the authors aim to better understand "where and why these simple models may fail" the choice of NSE is somewhat suprising, since NSE is a measure of overall performance, which provides limited insights into the reasons for high or low performance. Although an evaluation based on hydrological signatures would have enabled a more process-based diagnostic of model failures, I am not requiring this, since it would imply significant additional analyses. However, if the authors stick to NSE (or use KGE), I suggest that they use benchmarks (as suggested by Seibert et al., 2018) to account for the fact that high NSE/KGE values can be relatively easy to reach depending on the catchment and the season. I believe this would enable a more fair and enlightening assessment of the hydrological models across the catchments.

**Response: We originally selected NSE as it is a widely used and easy to interpret measure of performance. However, we agree that in order to better understand model failures we will need to consider additional measures of performance. Therefore, we plan to also present correlation, variance and mean bias, as called for by the first reviewer, to support the seasonal analysis of model performance that we have already carried out. As our focus is on reasons for model failures, we feel that these additional decomposed metrics will be more informative than the use of benchmarks.**

Comment 4: Relevance for the broad hydrological modelling community: A challenge here is to provide guidance for model selection, which is also relevant for modellers not using FUSE. Overall, the most interesting question is not really which FUSE model performs best, but why. I encourage the authors to discuss and highlight specific model elements that contribute to poor/good simulations, rather than focussing FUSE models themselves (e.g., TOPMODEL or PRMS). For instance, the fact that ARNO-VIC performs particularly well in high-BFI catchments is only an intermediary result, which is mostly relevant to FUSE users. The reasons why this is the case (e.g., last paragraph of Section 5.2), on the other hand, are relevant to a much wider group. I suggest a stronger emphasis on modelling decisions, as opposed to FUSE models, in particular in the most critical parts of the manuscripts (abstract and conclusions).

**Response: We agree that highlighting specific model elements that contribute to poor/good simulations would be of great use to the broad hydrological modelling community. Where possible, we have tried to outline modelling decisions that may cause differences in the results. However, it is difficult to distinguish which model elements are causing good/poor model performance, as the model structures differ in multiple aspects, and further analysis would be required to fully explore which modelling decisions are contributing to good/poor simulations. We have however added an extra table explaining which modelling decisions were applied for each FUSE model, highlighting the different model elements.**

Comment 5: Which process parameterisation are missing to capture the range of hydrological behaviours across the UK? The authors identify catchments in which the four model structures perform poorly, and reflect on characteristics of these catchments to which the poor performance can be attributed (e.g., chalk, snow, high human impacts). I suggest that the authors

dedicate a subsection in the Discussion Section to these findings, which are relevant for both model development and selection. Can they formulate hypotheses on why annual maximum flows are underestimated, which could be tested by future studies?

**Response: We have dedicated a section of the discussion to "Identifying missing process parameterisations" in response to this comment. As suggested by the first reviewer, the choice of NSE could result in underestimation of flood peaks, and we have therefore commented in the discussion on how our choice of metrics could be a factor leading to the underestimation of flood values.**

Comment 6: How critical is the selection of model structure? There are cases of great equifinality (i.e., high NSE for all structures, mostly for humid catchments). As mentioned above, a high NSE is not a guarantee that the model structure is adapted, but as long as this is recognised (and this could be clearer throughout the manuscript), I think it is fine for this study. But in other (more interesting) catchments, some model structures clearly outperform other structures, and there, model choice is critical. I think this should be stressed more prominently, since these are cases in which the inadequacy of the model structure cannot be overcome by parameter tuning. Given the general tendency of using the same model structure across very diverse environments (as discussed e.g. by Addor and Melsen, 2019), I think this is an important result, which could be underscored more. A related question is: which catchment characteristics explain these large NSE differences between model structures?

**Response: We explored the importance of model structure selection in figures 4, 7, 8, 10 and 11, with figure 7 looking at catchment characteristics which were related to differences between the model structures. However, we agree that the question of how critical the selection of model structure is for different catchments was not well addressed in the manuscript. Therefore, we have clarified the discussion of this in the discussion section.**

This leads me to a set of comments related to the use of catchment attributes to explain model performance.

Comment 7: Just like hydrological behaviour, model performance is not determined by a single catchment characteristic, but rather, by the interaction of multiple catchment characteristics. So, firstly, would it be possible to consider a wider range of catchment attributes? So far, the authors employ the BFI, annual rainfall, the wetness index and the runoff coefficient, but many more attributes could be used to describe each catchment (e.g., Beck et al., 2015). I encourage the authors to add other attributes, which they might have computed for other studies or retrieved from the UK hydrometric register, which they mention in Table 1, in order to describe the landscape in a more complete fashion (indicators of human interventions would also be useful, see below).

Comment 8: And secondly, I think it would be beneficial to better account for the interactions between these attributes. The authors combine several attributes in Figure 7 to explain model performance, which I find particularly interesting. Maybe that the analyses they will perform when revising this study will lead to more figures of this type, and enable a more systematic analysis of the interactions between these predictors (perhaps using regression trees, see Poncelet et al., 2017). This is critical to go from describing where models fail and to explaining why they fail.

Response: We selected the attributes of BFI, annual rainfall, wetness index and runoff coefficient as they were observed to have the largest impact on model performance. In response to reviewer comments, we have created additional plots of snow fraction, and factors affecting runoff on catchments across GB which are given in Figures 1 and 2. We do not want to add many more figures into the manuscript as we feel that this may detract from the main messages of the paper, but have added additional plots looking at interactions between attributes as supplementary information. Also we believe the current analyses are in keeping with the abstract nature of lumped modelling systems where a greater range of catchment attributes might only be loosely related to the structure and parameterisation of the model design. We aim to explore these issues with more spatially orientated modelling approaches in future publications.

Comment 9: Anthropogenic activities are repeatedly mentioned to explain poor model performance (e.g., P12L29, P14L16, P15L3). This is indeed plausible, but if qualitative or maybe quantitative indicators of the extent of human interventions could be included, so that their impacts on streamflow and model performance could be demonstrated or maybe even quantified, it would strengthen the study.

Response: We agree that this is required to strengthen comments made regarding reasons for model failures. We have information on factors affecting runoff for all catchments in the hydrometric register. However, this only gives an indicator of which factors may affect runoff, and not to what extent, and therefore we decided not to include it in the original manuscript. In response to reviewer comments, we have added plots of factors affecting runoff in Figure 1.

**Minor comments:**

Comment 10: I find the introduction too long. It attempts to cover too much material, and hence ends up being too general and its different parts are not very well connected. I suggest that the authors focus on what is really necessary to introduce their study, transfer parts of the text to the rest of the paper (e.g. the methods), and delete the rest.

Response: We agree. We have shortened the introduction, by combining and shortening sections on large sample and national hydrology, and condensing the introduction of modelling uncertainties by moving sentences to the methods section or deleting where appropriate.

Comment 11: Outlook: it might good to mention that, although this study focusses on four FUSE models, it is possible build additional FUSE model to transition progressively from one model to the next, and establish which modelling decisions contribute most to the differences in the simulations.

Response: The following has been added, "The framework allows the user to select different combinations of modelling decisions, starting with four parent models based on the structures of widely used hydrological models, and allowing the user to combine these decisions to create over 1200 different model structures".

Comment 12: Data availability: "This study provides a useful benchmark of the performance and associated uncertainties of four commonly used lumped model structures across GB, for future model developments and model types to be compared against". I agree. But then, I think that instead of saying that "All model outputs from this study are available upon request from the lead author", the authors should make the runs available online, and provide the doi, before the paper is published. This is expected by AGU journals, and I think it is good practice in order to avoid data loss.

**Response: We completely agree with this, and have provided a DOI for the data.**

**Other suggested changes:**

Title: the field is "large-sample hydrology", but here it should be "large sample"

**Response: Thank you, this has been changed to "over 1000 catchments" as it is more informative than "large sample".**

P1L15: add "and support model selection"

**Response: This has been added.**

P2L13: such as

**Response: Thank you for spotting this, this sentence has now been re-phrased.**

P2L29: impacted by what?

**Response: We have clarified and re-written the sentence to say "These have great benefits, as applying a consistent methodology across a large area enables comparison between places and identification of areas that may be at most risk of future hydrological hazards. "**

P4L12-17: this belongs to Data and Methods

**Response: These sentences have been moved to the data and methods section.**

P4L20: I suggest removing "(i.e. the number of storage components)" as it an arbitrary measure of complexity.

**Response: This has been removed.**

P5L22: discharge

**Response: This is referring to all the catchment data – we refer to discharge specific data at a later point in the methods section.**

P6L5: please define "sufficient"

**Response: This was explained in the following sentence. We have re-arranged these sentences to make this clearer, now saying "Of these, 1013 had sufficient information (defined as more than 10 years of available discharge data during the model evaluation period) available to include in this analysis."**

P7L2: I suggest mentioning here that none of these four models includes a snow Routine

**Response: We have added this, "They all close the water balance, have a gamma routing function and include the same processes, for example none of the models have a snow routine or vegetation module."**

P7L4: please define "dynamically different" and what makes them "equally plausible"

**Response: By "dynamically different" we meant that the models all represent the landscape in a different way, and have quite different and distinct structures as shown in figure 3. By "equally plausible" we are referring to the fact that we have no reason to expect one structure to behave better than the others, as all model structures are equally complete in terms of processes and all based on widely applied model structures. We have clarified this in the text by saying, "this leads us to believe that the model structures are dynamically different, as they are representing hydrological processes in different ways, yet as all are based on widely used hydrological models they are equally plausible and we have no a priori expectations that one model should outperform the others ".**

P8L13: please be more explicit about how this 13

**Response: This has been further explained with the text now reading, "This observed error value was selected following previous research on quantifying discharge uncertainty at 500 UK gauging stations for high flows, and represents the average 95$^{th}$ percentile range of the discharge uncertainty bounds for high flows."**

P9L21: saying "snowmelt module" implies that accumulation is simulated but melt is not, use "snow module" instead.

**Response: We agree, and have changed "snowmelt module" to "snow module."**

**Response to reviewer 3 (Anonymous)**

This paper provides a detailed investigation into the performance of four lumped conceptual models over large number of catchments in the UK. It demonstrates some very interesting findings, such as the fact that all four models have very similar performance on a catchment-by-catchment basis, and that only one of the models is deemed suitable for catchments with very

5   high BFI. This paper is generally well written, set out and easy to follow, and the graphics provided assist the reader well in the interpretation of the results, I particularly like Figures 5 and 7. The discussion section should be synthesised as it feels repetitive of the results section. Overall, I feel that the motivations of the research, and the implications of the results are not very well reasoned. The authors need to think a bit more carefully about how others may make use of these results, and in particular, should publish the model performance scores as supplementary information (see my comments below).

10   **Response: We would like to thank the reviewer for taking the time to read the paper in depth, and for their constructive comments.**

Comment 1. You've "benchmarked" performance, but you haven't provided these benchmarks. If I were to now go and simulate a UK catchment, I still cannot easily compare my results with yours to see if I have a better model. For you to have

15   achieved your aims, I would expect a supplementary table of the best scores the models achieved in each catchment, and the parameter values that produced them.

**Response: We completely agree with this, and the results can now be accessed through a DOI.**

Comment 2. Section 3.2 – why NSE?

20   **Response: We originally selected NSE as it is a widely used and easy to interpret measure of performance. However, as noted by the other reviewers, in order to better understand model failures we will consider additional metrics. Therefore, we plan to also present correlation and mean bias.**

Comment 3. Section 3.2 – "results are stored for a number of additional metrics not reported here". Stored where? Why would

25   I care about this if you haven't made them available to me? I suggest you summarise these additional metrics in supplementary information. This may also address the issue of only reporting on NSE here.

**Response: We have removed this sentence from the methods and included additional metrics in the results which will also be provided thorough the DOI.**

30   Comment 4. Your statement in the abstract L23 that NSE scores of 0.72-0.78 were achieved for all catchments is misleading. How useful a measure is the "median maximum NSE for all the catchments"? It's pretty cryptic. There are catchments in E Scotland, and Anglian region that are showing pink/red for all 4 models, so NSE must be <0.5. Having got to page 9 I now see

what you meant, but it isn't clearly stated. The sentences on P12 L16-17 are a better summary of the performances across catchments. Same issue on P16 L 32.

**Response: Thank you for pointing out that this is not clear. We have replaced the statement in abstract L23 with "Our results show that simple, lumped hydrological models were able to produce adequate simulations across most of Great Britain, with each model producing simulations exceeding 0.5 Nash Sutcliffe efficiency over at least 80% of catchments."**

Comment 5. Catchment characteristics and climate – do all FUSE models maintain the water balance? Can you comment on the existence of models that don't (e.g. GR4J), and how those may overcome such problems? What are the implications of maintaining vs not maintaining water balance in conceptual lumped models? Are the four models you've chosen actually quite similar to each other? I think you need to make more of this somehow.

**Response: Yes, all the FUSE models used in this study maintain the water balance, and we have clarified this in the methods section. To address these questions we have added a paragraph in the discussion on how models that do not maintain the water balance have been used to improve modelling in groundwater dominated regions. In response to reviewer 1, this includes discussion of papers by Le Moine at al. (2007, 2008) about groundwater flows and water balance closure.**

Comment 6. P8 L23 – only a 1 year warm up period? This is not sufficient for many GW dominated
catchments in the SE.

**Response: Thank you for this advice. We initially selected 1 year, as it is often considered sufficient for simple, lumped models such as the FUSE models. However, following this comment we carried out additional analysis of the simulated flows and found that whilst 1 year is a long enough warmup period for many catchments, it did not appear sufficient for some of the catchments in the SE as suggested. We will have increased the warmup period to 5 years, re-analysed the data and re-made all the figures to reflect this.**

Comment 7. P6 L6 – 2 years of data was your criteria for catchment selection, this doesn't seem sufficient to me

**Response: We originally aimed to keep as many catchments as possible for the analysis. However, you are correct that 2 years of data is not long for model evaluation. We have now added a tougher criteriona for catchment selection, of more than 10 years of available discharge data during the model evaluation period. The figures have been re-made to reflect this.**

Comment 8. Reading through your discussion seems very repetitive of the results chapter. Can these be better synthesised, to reduce the discussion section?

16. Your discussion is longer than the rest of the paper put together!

**Response: We have reduced the length of the discussion section, and re-structured the old sections 5.1-5.3 to reduce repetition.**

Comment 9. P2 L32 "a national scale model" – you're talking about applying a catchment model nationally. Can this be classified a national scale model?

**Response: In this section we were aiming to discuss the importance of national scale modelling more generally, suggesting that our work could be informative for evaluation of a national scale model. We were not saying that our application of a catchment model across GB was a national scale model.**

Comment 10. P3 L16 - "Secondly, evaluating more complex hydrological models relative to benchmark performance of simple models ensures that the relative difficulty of simulating different catchments is implicitly considered (Seibert et al., 2018)." I don't think I understand what you're saying here.

**Response: This has been re-phrased and further explained to make the meaning clearer. It now reads "Secondly, lumped hydrological models provide a good benchmark for evaluating more complex models, as they give an indication of what it is possible to achieve for a specific catchment and the available data (Seibert et al., 2018). This can help us identify whether a model is performing well in a catchment relative to how it should be expected to perform for the particulars of that catchment. For example, if a modeller gains an efficiency score of 0.7 for their model in a specific catchment, it is subjective whether this is a good or poor performance. However, if lumped, conceptual models tend to have efficiency scores of around 0.9 for that catchment then the modeller knows that their model is performing poorly relative to what is possible."**

Comment 11. P10 L22-24 – "For very low values of the ARNO-VIC 'b' exponent (AXV_BEXP) as seen for high BFI vales in Fig. 6 for behavioural model distributions means that only at very high, near full upper storage levels is any larger extent of saturated areas predicted" – I don't follow this sentence either.

**Response: This has been re-phrased.**

Comment 12. P8 L3 - Can you explain conditional probabilities in more detail?

**Response: We have extended this paragraph, now saying "Conditional probabilities were assigned to each behavioural parameter set based on their behavioural Efficiency score, and these were normalised to sum to 1. This meant that the simulations which scored the highest efficiency value had larger conditional probabilities, and simulations which had efficiency values just above 0.5 would have very low conditional probabilities. For each daily timestep, a $5^{th}$, $50^{th}$ and $95^{th}$ simulated discharge bound was produced from these conditional probabilities, for each catchment and model structure individually as described in Beven and Freer (2001). This meant that simulations with a higher efficiency score were given a higher weighting when producing the discharge bounds." Simply the behavioural weights**

(probabilities) assigned to each model are conditional on the choices made in the modelling exercise, here dependent on the sample design, the choice of parameter ranges, the model performance metric, and hence conditional.

Comment 13. P11 L 23 – "the top row of plots" – there is only one plot in Fig 8!

**Response: Thank you for noticing that! We had originally displayed figures 8 and 9 as a single plot. This has been corrected.**

Comment 14. P12 L 7-8 "However, variations between years are less apparent when looking at 25th and 75th percentiles in Fig. 8." We can't distinguish variation between years from Fig 8?

**Response: Again, thank you for noticing this, it has been corrected to point to the right figure.**

Comment 15. Please provide more sensible y axis labels for fig 8 and 9, e.g. "AMAX discharge score", and "AMAX percentage overlap" respectively. Multiply Fig 9 y axis by 100 to make it an actual percentage value, as you have referred to it as such in the text.

**Response: We agree with this comment and have changed the figure.**

Comment 17. P13 L 3 – you've made no reference to anthropogenic influences in Scotland. This statements seems a bit throwaway.

**Response: We have removed this sentence.**

Comment 18. P13 L9 – it is not just the Thames basin that is affected by abstractions! A lot of Anglian region is VERY heavily influenced.

**Response: we have changed this sentence to "a considerable proportion of river discharges throughout the Anglian region are abstracted."**

Comment 19. P13 L12 "we found that the ensemble of model structures produced better results overall than any single model" – can you validate that statement from your figures?

**Response: This can not be directly validated in a specific figure, but it can be seen across the figures, especially looking at Figure 7, where we see that no single model produces good results for all catchments.**

Comment 20. P13 L15 – "The ensemble of model structures was able to take advantage of this" - this seems to be a contradictory argument to the previous statement that the models all have similar performance to each other on a catchment by catchment basis. I think you need to tease these two arguments out better somehow. E.g. in some situations the choice of a

different model can yield better results (e.g. high baseflow), but in other situations, none of the models can do well (e.g. abstractions). What are the implications of this?

**Response: We have clarified these arguments in the discussion.**

5 Comment 21. P17 L 11-14 "We also evaluated model predictive capability for high flows, as good model performance in replicating the hydrograph, assessed using Nash-Sutcliffe efficiency, does not necessarily mean models are performing well for other hydrological signatures. We found that the FUSE models tended to underestimate peak flows, and there were variations in model ability between years with models performing particularly poorly for extremely wet years." – so what? What are the potential implications?

10 **Response: We have added discussion about the implications for flood modelling and forecasting here.**

**Typos and grammar:**

1. P2 L27 – CAMELS and MOPEX datasets (what are they datasets of?)

**Response: This sentence has been clarified -  "the CAMELS or MOPEX hydrometeorological and catchment attribute**
15 **datasets."**

3. P5 L22 - remove "Environment Agency", a catchment is a catchment, the EA don't own the catchments, even if they do own the gauges!

**Response: "Environment Agency" has been removed.**

4. Amend "Rainfall is highest in the West and North of GB and lowest in the East and South varying from a minimum of 500mm to a maximum of 4496mm per year (see Fig. 1)" to "On average, rainfall is highest in the north and west of GB, and lowest in the south and east, with GB totals varying from a minimum of 500mm to a maximum of 4496mm per year (see Fig. 1).

25 **Response: Thank you for the suggestion, this sentence has been amended.**

2. P5 L14 - "these" should be "those"

5. P6 L1 - remove the "of" after "South-East"

6. P6 L12 – they are the "UK Met Office" not the "UK Meteorological Office".

30 7. P6 L14 and L20 – replace "laid" with "lay"

8. L6 L21 and elsewhere – "data" is plural, and should be followed by "were" instead of "was"

9. P8 L15 – "observational uncertainty certainty bounds" huh?? Can you not just remove the word certainty here?

10. P9 L15 – you haven't introduced the abbreviation "SAC"

11. P10 L5 – I'd call that northeast Scotland, not central Scotland

12. P11 L29 – "behavioural model" should be "behavioural models"

13. P13 L7 – do you mean model "structures"?

14. P16 L17 – "we also shown how"

15. P16 L19 – refer to Fig 6

**Response: Thank you for spotting these typos and grammatical errors, we have corrected these in the manuscript.**

16. P16/17 – "The performance of the four models was similar, and all models showed similar spatial patterns of performance, and there was no single model that outperformed the others across all catchment characteristics and for both daily flows and peak flows." – and, and, and

**Response: This sentence has been improved to "The performance of the four models was similar, with all models showing similar spatial patterns of performance, and no single model  outperforming the others across all catchment characteristics for both daily flows and peak flows."**

17. P17 L8 – "we found models performed poorly for catchments for catchments with unaccounted losses"

**Response: we have removed the repetition.**

HESS REVIEW CHECKLIST

1. Does the paper address relevant scientific questions within the scope of HESS? Yes

2. Does the paper present novel concepts, ideas, tools, or data? Yes

3. Are substantial conclusions reached? Nearly, the wider implications, and utility of the research need to be better considered

4. Are the scientific methods and assumptions valid and clearly outlined? Yes

5. Are the results sufficient to support the interpretations and conclusions? Yes

6. Is the description of experiments and calculations sufficiently complete and precise to allow their reproduction by fellow scientists (traceability of results)? Yes

7. Do the authors give proper credit to related work and clearly indicate their own new/original contribution? Yes

8. Does the title clearly reflect the contents of the paper? Yes

9. Does the abstract provide a concise and complete summary? Yes

10. Is the overall presentation well-structured and clear? Yes

11. Is the language fluent and precise? Yes

12. Are mathematical formulae, symbols, abbreviations, and units correctly defined and used? Yes

13. Should any parts of the paper (text, formulae, figures, tables) be clarified, reduced,

combined, or eliminated? Yes, the discussion should be reduced

14. Are the number and quality of references appropriate? Yes

15. Is the amount and quality of supplementary material appropriate? No

**Response: Thank you for this largely positive summary checklist. We have addressed points 3 and 13 through our changes to the discussion section, and point 15 by making our output data available through a DOI.**

**Benchmarking the predictive capability of hydrological models for river flow and flood peak predictions across over 1000 catchments in Great Britain**

Rosanna A. Lane[1], Gemma Coxon[1], Jim E. Freer[1,3], Thorsten Wagener[2,3], Penny J. Johnes[1,3], John P. Bloomfield[4], Sheila Greene[5], Christopher J. A. Macleod[6], Sim M. Reaney[7]

[1]School of Geographical Sciences, University of Bristol, Bristol, BS8 2NQ, United Kingdom
[2]Faculty of Engineering, University of Bristol, Bristol, BS8 2NQ, United Kingdom
[3]Cabot Institute, University of Bristol, Bristol, BS8 2NQ, United Kingdom
[4]British Geological Survey, Maclean Building, Wallingford, OX10 8BB, United Kingdom
[5]Trinity College Dublin, Dublin, Ireland
[6]The James Hutton Institute, Craigiebuckler, Aberdeen, AB15 8QH, United Kingdom
[7]Department of Geography, Durham University, Durham, DH1 3LE, United Kingdom

*Correspondence to*: Rosanna A. Lane (R.A.Lane@bristol.ac.uk)

**Abstract.** Benchmarking model performance across large samples of catchments is useful to guide model selection and  future model development. Given uncertainties in the observational data we use to drive and evaluate hydrological models, and uncertainties in the structure and parameterisation of models we use to produce hydrological simulations and predictions, it is essential that model evaluation is undertaken within an uncertainty analysis framework.

Here, we benchmark the capability of several lumped hydrological models across Great Britain, by focusing on daily flow and peak flow simulation. Four hydrological model structures from the Framework for Understanding Structural Errors (FUSE) were applied to over 1000 catchments in England, Wales and Scotland. Model performance was then evaluated using  standard performance metrics for daily flows, and  novel performance metrics for peak flows considering parameter uncertainty.

Our results show that lumped hydrological models were able to produce adequate simulations across most of Great Britain, with each model producing simulations exceeding 0.5 Nash Sutcliffe efficiency for at least 80% of catchments. All four models showed a similar spatial pattern of performance, producing better simulations in the wetter catchments to the west, and poor model performance in Scotland and southeast England. Poor model performance was often linked to the catchment water balance, with models unable to capture the catchment hydrology where the water balance did not close. Overall, performance was similar between model structures, but different models performed better for different catchment characteristics and metrics for assessing daily or peak flows, leading to the ensemble of model structures outperforming any single structure thus demonstrating the value of using  multi-model structures across a large sample of different catchment behaviours.

This research  evaluates what conceptual lumped models can achieve as a performance benchmark, as well as providing interesting insights into where and why these simple models may fail. The large number of river catchments included in this study makes it an appropriate benchmark for any future developments of a national model of Great Britain.

**Commented [RI1]:** R2 Page 10 Line 3 "Title: the field is "large-sample hydrology", but here it should be "large sample""

**Commented [RI2]:** R2C4 Page 7 Line 15: "Comment 4: Relevance for the broad hydrological modelling community: A challenge here is to provide guidance for model selection, which is also relevant for modellers not using FUSE."

**Commented [RI3]:** R3C4 Page 12 Line 30 "Your statement in the abstract L23 that NSE scores of 0.72-0.78 were achieved for all catchments is misleading. How useful a measure is the "median maximum NSE for all the catchments"? It's pretty cryptic. There are catchments in E Scotland, and Anglian region that are showing pink/red for all 4 models, so NSE must be <0.5. Having got to page 9 I now see what you meant, but it isn't clearly stated. The sentences on P12 L16-17 are a better summary of the performances across catchments. Same issue on P16 L 32."

**1 Introduction**

Lumped and semi-distributed hydrological models, applied singularly or within nested sub-catchment networks, are used for a wide range of applications. These include water resources planning, flood/drought impact assessment, comparative analyses of catchment and model behaviour, regionalisation studies, simulations at ungauged locations, process based analyses, and climate or land-use change impact studies (see for example Coxon et al., 2014; Formetta et al., 2017; Melsen et al., 2018; Parajka et al., 2007; Perrin et al., 2008; Poncelet et al., 2017; Rojas-Serna et al., 2016; Salavati et al., 2015; van Werkhoven et al., 2008). However, model skill varies between catchments due to differing catchment characteristics such as climate, land use and  topography. Evaluating  where models perform well/poorly and the reasons for these variations in model performance, can provide a benchmark of model performance to help us better interpret modelling results across large samples of catchments (Newman et al., 2017) and lead to more targeted model improvements through synthesising those interpretations.

**1.1 Large sample hydrology**

  large-sample hydrological studies, also known as comparative hydrology, test hydrological models on many catchments of varying characteristics (Gupta et al., 2014; Sivapalan, 2009; Wagener et al., 2010).  science The use of a large range of   improved understanding of hydrological processes, robustness of    Evaluating model performance across a large sample of catchments can lead to improved understanding of hydrological processes and teach us a lot about hydrological models  the appropriateness of model structures for different types of catchment characteristics (i.e. Van Esse et al., 2013; Kollat et al. 2012), emergent properties and spatial patterns, key processes that we should be improving and identification of areas where models are unable to produce satisfactory results (e.g. Newman et al., 2015; Pechlivanidis and Arheimer, 2015). This can guide model selection, and also teach us about appropriate model parameter values for different catchment characteristics, with the production of parameter libraries which can be used for parameter calibration in ungauged basins, and increase robustness of calibration in poorly gauged basins (Perrin et al., 2008; Rojas-Serna et al., 2016).

At the same time, regional-continental  scale hydrological modelling studies are increasingly needed, to address large-scale challenges such as managing water supply, water scarcity and flood risk under climate change, and to inform large-scale policy decisions such as the European Union's Water Framework Directive (European Parliament, 2000).

**Commented [RI4]:** R2: Page 10 Line 7 "P1L15: add "and support model selection""

**Commented [RI5]:** R1 Page 3 Line 28 "authors discuss the benefits of national scale hydrological modelling. Another benefits could be the production of parameter libraries, which could be used for regional studies or model calibration on poorly gauged to ungauged basins or engineering studies. Authors can make references to papers on this subject (Perrin et al., 2008 ; Rojas-Serna et al., 2016 ; or some other works by Seibert)."

**1.1 National Scale Hydrological Modelling**

~~Large-scale challenges such as climate change and population growth will impact river flow regimes across country wide hydro-climatic gradients affecting some regions more than others. Solutions need to be found to better predict water supplies, water scarcity and improve modelling studies used in the evaluation of national water policies such as the European Union's Water Framework Directive (European Parliament, 2000).datasets(Addor et al., 2017; Duan et al., 2006). For these large-scale challenges, national scale modelling approaches using a consistent methodology allow for comparison between places and identification of areas which may be most impacted to the hydrological regime.~~

However, the range of catchment characteristics and hydrological processes across national scales pose a great challenge to the implementation and evaluation of a national-scale model (Lee et al., 2006), and we therefore need large-scale evaluations of model capability to identify which processes are important and which model structure(s) are most appropriate. ~~A national model ideally needs to represent varied catchment characteristics such as climate, topography and hydrogeology. Furthermore, whilst many hydrological studies focus only on natural catchments, a national scale approach must include catchments with human impacted modified flow regimes, for example heavily urbanised areas, rivers downstream from reservoirs and areas where abstractions and discharges are taking place to understand their impacts, albeit these can be difficult to disentangle (Salavati et al., 2015). Incorporating this influence of human activity has been identified as a major challenge to advancing societally relevant hydrological analyses (Montanari et al., 2013).~~

**1.2 Benchmarking hydrological models**

Model skill varies between places, and it is therefore important for a modeller to understand the relative model skill for their study region, and how that relates to their core objectives. A single model structure will vary in its ability to produce good flow time-series across different environments and time-periods (McMillan et al., 2016), expressed sometimes as model agility (Newman et al., 2017). It is important for a modeller to know the performance capabilities of their model when deciding whether to place confidence in any predictive skill. One way to evaluate this relative model skill is by comparing the model performance to a benchmark, which is an indicator of what it is possible to achieve in a catchment given the data available

**Commented [RI6]:** R3: Page 16 Line 10 "1. P2 L27 – CAMELS and MOPEX datasets (what are they datasets of?)"

**Commented [RI7]:** R2: Page 10 Line 12 "P2L29: impacted by what?"

**Commented [RI8]:** R2C10 Page 9 Line 10 "I find the introduction too long. It attempts to cover too much material, and hence ends up being too general and its different parts are not very well connected. I suggest that the authors focus on what is really necessary to introduce their study, transfer parts of the text to the rest of the paper (e.g. the methods), and delete the rest."

(Seibert, 2001). This helps a modeller make a more objective decision on whether their model is performing well. Examples of benchmarks that models can be evaluated against include climatology, mean observed discharge, or the performance of a simple, lumped hydrological model for the same conditions (Pappenberger et al., 2015; Schaefli and Gupta, 2007; Seibert, 2001; Seibert et al., 2018).

The creation of a national benchmark series of performance of simple, lumped models can therefore be useful for a variety of reasons. Firstly, a benchmark series of lumped model performance is a useful baseline upon which more complex or highly distributed modelling attempts can be evaluated (Newman et al., 2015). This would ensure that future model developments are improving upon our current capability therefore justifying additional model complexity. Secondly, lumped hydrological
10  models provide a particularly good benchmark for evaluating more complex models, as they give an indication of what it is possible to achieve for a specific catchment and the available data (Seibert et al., 2018). This can help us identify Secondly, evaluating more complex hydrological models relative to benchmark performance of simple models ensures that the relative difficulty of simulating different catchments is implicitly considered (Seibert et al., 2018). Using benchmarks in this way can improve our evaluation of hydrological models, helping to identify whether a model is performing well in a catchment relative
15  to how it should be expected to perform for the particulars of that catchment. For example, if a modeller, using more complex modelling approaches, gains an efficiency score of 0.7 for their model in a specific catchment, itthere is some subjectivitye whether this is a good or poor performance depending on the modelling objective. However, if lumped, conceptual models already applied at the same catchment tend to have efficiency scores of around 0.9 for that catchment then the modeller knows that their model is performing poorly relative to what is possible. Thirdly, national benchmarks are useful for users of models
20  as they can highlight areas where models have more or less skill, and where model results should be treated with caution.

**1.3 Assessing Uncertainty**

Hydrological model output is always uncertain, due to uncertainties in the observational data used to drive and evaluate the
25  models, boundary conditions, uncertainties in selection of model parameters and in the choice of a model structure (Beven and Freer, 2001). There is a large and rapidly growing body of literature on uncertainty estimation in hydrological modelling, with many techniques emerging to assess the impact of different sources of uncertainty on model output, as summarised in Beven (2009). Despite this, uncertainty estimation is not yet routine practice in comparative or large-sample hydrology and few nationwide hydrological modelling studies have included uncertainty estimation, tending to look more at regionalization of
30  parameters, multi-objective calibration techniques, or the use of flow signatures in model evaluation (i.e. Donnelly et al., 2016; Kollat et al., 2012; Oudin et al., 2008; Parajka et al., 2007b).

Hydrological modelling studies often assume that the rainfall and evapotranspiration data used to drive hydrological models, and the discharge data used to evaluate models is correct. However, errors in observational data can be high, especially for

Commented [RI9]: R2: Page 6 Line 15 "Model intercomparison vs. benchmarking: Since the authors use the term "benchmarking" in the title and throughout the manuscript, I encourage them to clarify in the introduction what differentiates model benchmarking from model intercomparison. As the authors compare FUSE structures with each other, isn't their study rather a model intercomparison? Do the authors mean that their runs can be used as benchmark by future studies, as suggested on P12L12? Please clarify." – I have clarified why benchmarks may be used here.

Commented [RI10]: R3C10 Page 14 Line 10 "10. P3 L16 - "Secondly, evaluating more complex hydrological models relative to benchmark performance of simple models ensures that the relative difficulty of simulating different catchments is implicitly considered (Seibert et al., 2018)." I don't think I understand what you're saying here."

Field Code Changed

extreme events (Coxon et al., 2015; Mcmillan et al., 2012; Westerberg et al., 2016). In a recent review of observational uncertainties for hydrology, McMillan et al. (2012) found that measures of discharge typically have uncertainties in the range 2-19%, but low flows and out of bank flows have much higher uncertainties of 50-100% and 40% respectively. It is therefore important to consider these data uncertainties in the evaluation of hydrological models and any diagnostic evaluations (Coxon et al., 2014).

Parameter uncertainty can be evaluated through calibrating models within an uncertainty evaluation framework. There are many different uncertainty analysis procedures in hydrology such as the Parameter Solution (ParaSol) method (van Griensven and Meixner, 2006), the Integrated Bayesian Uncertainty Estimator (IBUNE) (Ajami et al., 2007), the Sequential Uncertainty Fitting algorithm (SUFI-2) (Abbaspour et al., 2007), and the Generalized Likelihood Uncertainty Estimation (GLUE) framework (Beven and Binley, 1992).

Parameter uncertainty is often evaluated through calibration models within an uncertainty evaluation framework (e.g. GLUE, (Beven and Binley, 1992) or ParaSol (van Griensven and Meixner, 2006)). Whilst many studies have explored parameter uncertainty, it is less common to evaluate the additional impact of model structural uncertainty on hydrological model output (Butts et al., 2004). Model structures can differ in their choice of processes to include, process parameterisations, model spatial and temporal resolution and model complexity (i.e. the number of storage components). Studies attempting to address model structural uncertainty often apply multiple hydrological model structures and compare the differences in output (i.e. Ambroise et al., 1996; Vansteenkiste et al., 2014; Velázquez et al., 2013), (Ambroise et al., 1996; Perrin et al., 2001; Vansteenkiste et al., 2014; Velázquez et al., 2013), and in climate impact studies (i.e. Bosshard et al., 2013; Karlsson et al., 2016; Samuel et al., 2012). These studies have found that the choice of hydrological model structure can strongly affect the model output, and therefore hydrological model structural uncertainty is an important component of the overall uncertainty in hydrological modelling and cannot be ignored.

The use of flexible model frameworks has emerged as a is a useful tool for exploring the impact of model structural uncertainty in a controlled way, and for identifying the different aspects of a model structure which are most influential to the model output. These flexible modelling frameworks allow a modeller to build many different model structures using combinations of generic model components (Fenicia et al., 2011). For example, the Modular Modelling System (MMS) of Leavesley et al., (1996) allows the modeller to combine different sub-models. The SUPERFLEX modelling framework presented by Fenicia et al., (2011) is based on generic building blocks such as reservoirs, junctions, and constitutive functions, and allows the modeller to generate new model configurations using these building blocks. There is also the and the Framework for Understanding Structural Errors (FUSE), developed by Clark et al., (2008), which combines process representations from four commonly used hydrological models to create over 79 1000 unique model structures.

**1.4 Study Scope and Objectives**

The main objective of this study is to comprehensively benchmark performance of an ensemble of lumped hydrological model structures across Great Britain, focusing on daily flow and peak flow simulation. This will be is the first evaluation of

**Commented [RI11]:** R2C10 Page 9 Line 10 "I find the introduction too long. It attempts to cover too much material, and hence ends up being too general and its different parts are not very well connected. I suggest that the authors focus on what is really necessary to introduce their study, transfer parts of the text to the rest of the paper (e.g. the methods), and delete the rest."

**Commented [RI12]:** R2 Page 10 Line 17 "P4L12-17: this belongs to Data and Methods"

**Commented [RI13]:** R2 Page 10 Line 19 "P4L20: I suggest removing "(i.e. the number of storage components)" as it an arbitrary measure of complexity."

**Commented [RI14]:** R1: Line 34 Page 4 "I would also make a reference to Perrin et al. 2001 here"

hydrological model ability across a large sample of British catchments whilst considering model structural and parameter uncertainty. This will be useful both as a benchmark of model performance against which other models can be evaluated and improved upon in Great Britain, and as a large-sample study which can provide general insights into the influence of catchment characteristics and selected model structure and parameterisation on model performance.

Commented [RI15]: R2: Page 6 Line 15 "Do the authors mean that their runs can be used as benchmark by future studies, as suggested on P12L12? Please clarify."

5      The specific research questions we investigate are:

   1. How well do simple, lumped hydrological model structures perform across Great Britain, when assessed over annual and seasonal time scales via standard performance metrics?

   2. Are there advantages in using an ensemble of model structures over any single model, and so are there any emergent patterns/characteristics in which a given structure and/or behavioural parameter set outperforms others?

10    3. What is the influence of certain catchment characteristics on model performance?

   4. What is the predictive capability of these those identified as behavioural models for then predicting annual maximum flows when applied in a parameter uncertainty framework?

Commented [RI16]: R3: Page 16 Line 24 "P5 L14 - "these" should be "those""

To address these questions, we have applied the four core conceptual hydrological models from the FUSE hydrological framework to 1128 1013 British catchments, within an uncertainty analysis framework. Model performance and predictive

15    capability have been evaluated at each catchment, providing a national overview of hydrological modelling capability for simpler lumped conceptualisations over Great Britain.

**2 Data and Catchment Selection**

**2.1 Catchment Data**

This study was national in scope, using a large discharge data set of 1128 1013 Environment Agency catchments distributed

20    across Great Britain (GB). The catchments cover all regions and include a wide variety of catchment characteristics including topography, geology and climate (see Table 1), and include both natural and human impacted catchments (see Figure 1Figure 12).

Commented [RI17]: Tougher selection criteria – still more than 1000 catchments.

Commented [RI18]: R3: Page 16 Line 14 "3. P5 L22 - remove "Environment Agency", a catchment is a catchment, the EA don't own the catchments, even if they do own the gauges!"

Commented [RI19]: R1: Page 5 Line 17 "Table 1 is not cited within §2"

On average, Rrainfall is highest in the West and Northnorth and west of GB, and lowest in the East and Southsouth and east, with GB totals varying from a minimum of 500mm to a maximum of 4496mm per year (see Figure 2Fig. 1). There is also

[revised manuscript text omitted]

**Commented [RI22]:** R1: Page 5 Line 12 "In §2, I would give an estimation of the proportion of watersheds where snowmelt processes are observable (solid precipitation >20% of total precipitation ?)"

**Commented [RI23]:** R2: Page 10 Line 27 "P6L5: please define "sufficient""

R3C7 : Page 13 Line 26 "P6 L6 – 2 years of data was your criteria for catchment selection, this doesn't seem sufficient to me"

**Commented [RI24]:** R3: Page 16 Line 26 "P6 L12 – they are the "UK Met Office" not the "UK Meteorological Office"."

**Commented [RI25]:** R3: Page 16 Line 27 "P6 L14 and L20 – replace "laid" with "lay""

**Commented [RI26]:** R3: Page 16 Line 28 "L6 L21 and elsewhere – "data" is plural, and should be followed by "were" instead of "was""

**Commented [RI27]:** R2C11 Page 9 Line 18 "Outlook: it might good to mention that, although this study focusses on four FUSE models, it is possible build additional FUSE model to transition progressively from one model to the next, and establish which modelling decisions contribute most to the differences in the simulations."

For this study, only the four parent models from the FUSE framework were selected due to the computational requirements of running the models across such as large number of catchments, and that the core models should provide the core differences of models compared to all the possible variants. These models are based on four widely used hydrological models; TOPMODEL (Beven and Kirkby, 1979), the Variable Infiltration Capacity (ARNO/VIC) model (Liang et al., 1994; Todini, 1996), the Precipitation-Runoff Modelling System (PRMS) (Leavesley et al., 1983) and the Sacramento model (Burnash et al., 1974). The models are all lumped, conceptual models of similar complexity and all run at a daily timestep within the FUSE framework. They all close the water balance, have a gamma routing function and include the same processes, for example none of the models have a snow routine or vegetation module. However, the structures of these models differ through the architecture of the upper and lower soil layers and parameterizations for simulation of evaporation, surface runoff, percolation from the upper to lower layer, interflow and baseflow (Clark et al., 2008), as shown in Figure 3 and Table 3. This leads us to believe that the model structures are dynamically different, as they are representing hydrological processes in different ways, yet as all are based on widely used hydrological models they are equally plausible and we have no a priori expectations that one model should outperform the others (Clark et al., 2008).

~~Parameter uncertainty can be evaluated through calibrating models within an uncertainty evaluation framework. There are many different uncertainty analysis procedures in hydrology such as the Parameter Solution (ParaSol) method (van Griensven and Meixner, 2006), the Integrated Bayesian Uncertainty Estimator (IBUNE) (Ajami et al., 2007), the Sequential Uncertainty Fitting algorithm (SUFI-2) (Abbaspour et al., 2007), and the Generalized Likelihood Uncertainty Estimation (GLUE) framework (Beven and Binley, 1992).~~

The models were run within a Monte-Carlo simulation framework. There are 23 adjustable parameters within the FUSE framework, as shown in Table 2. Each of these was assigned upper and lower bounds based upon feasible parameter ranges and behavioural ranges identified in previous research (Clark et al., 2008; Coxon et al., 2014). Monte-Carlo sampling was then used to generate 10,000 parameter sets within these given bounds. Therefore, for each of the 1013 catchments, the 4 hydrological model structures were each run using the 10,000 possible parameter sets over the 21 year period 1988-2008, resulting in.  >40 million simulations being carried out.

**3.2 Evaluation of Model Performance**

The objective of this study was to evaluate the model's ability to reproduce observed catchment behaviour with a focus on assessing the strengths and weaknesses of each model in different catchments. Given the large number of catchments evaluated, it was not possible to evaluate model performance against a large range of objective functions with this paper, here we aim to benchmark behaviour to metrics that capture different aspects of model performance. Consequently, we chose to evaluate the o v e r a l l performance of the hydrological models through the widely used Nash-Sutcliffe Efficiency Index (Nash and Sutcliffe, 1970) which is an easy to interpret measure of model performance that is often used in studies interested in high flows as it emphasizes fit to peaks. To further diagnose the reasons for model good/poor performance, the simulation

Commented [RI28]: R2: Page 10 Line 32 "P7L2: I suggest mentioning here that none of these four models includes a snow Routine"

R3C5: Page 13 Line 8 "Catchment characteristics and climate – do all FUSE models maintain the water balance? Can you comment on the existence of models that don't (e.g. GR4J), and how those may overcome such problems? What are the implications of maintaining vs not maintaining water balance in conceptual lumped models? Are the four models you've chosen actually quite similar to each other? I think you need to make more of this somehow."

Commented [RI29]: R2: Page 11 Line 1 "P7L4: please define "dynamically different" and what makes them "equally plausible""

Commented [RI30]: R3C2: Page 12 Line 19 "Section 3.2 – why NSE?"

[revised manuscript text omitted]

Commented [RI40]: R3C6: Page 13 Line 18" P8 L23 – only a 1 year warm up period? This is not sufficient for many GW dominated catchments in the SE."

Commented [RI41]: R3: Page 16 Line 30 "P9 L15 – you haven't introduced the abbreviation "SAC""

east/west divide in model performance, with models typically performing better in wetter western catchments compared to drier catchments in the east. Clusters of poorly performing catchments can be seen in the east of England around London and in central Scotland, where all models are failing to produce satisfactory simulations. There are also more localised catchments where all models are failing, such as in north Wales and northern England. Areas where all models are performing well include south Wales, southwest England and southwest Scotland.

However, looking at the decomposed performance metrics in Figure 5, differences between the model structures emerge that cannot be seen from the overall NSE scores. Firstly, the models show different biases (top row of plots, Fig. 5). The SACRAMENTO model is generally balanced, whilst best scoring simulations tend to underpredict flows for TOPMODEL, and overpredict flows for ARNO/VIC and PRMS. Secondly, all models tend to underpredict the standard deviation of flows (middle row of plots, Fig.5), with TOPMODEL generally underpredicting the most, but PRMS stands out as overpredicting the standard deviation for many catchments in the southeast. Thirdly, the pattern of correlation is similar between the models, and closely matches the patterns seen for NSE. This is unsurprising, as the correlation term is given a high weighting when calculating NSE (Gupta et al., 2009). It is particularly interesting that whilst the models are all calibrated in the same way and are producing similar NSE scores, the decomposed metrics show clear differences between the best simulations produced using each structure,

The decomposed metrics also help to identify which aspects of NSE are causing models to fail. Models have problems simulating the bias, standard deviation and correlation for catchments in southeast England (Fig. 5). The localised poorly performing catchments in north Wales are failing due to poor simulation of variance and correlation. Poor performance in northeast Scotland is due to poor correlation and underestimation of variance for all models. In central/northern Scotland all models except TOPMODEL overpredict bias, leading to TOPMODEL being the only model able to produce reasonable simulations for these catchments.

The overall performance of the four model structures was similar, with TOPMODEL, ARNO, PRMS and Sacramento producing simulations exceeding 0.5 NSE for 87%, 90%, 81% and 88% of catchments respectivelywith the median maximum NSE from all the catchments being 0.72 for TOPMODEL, 0.75 for ARNO, 0.73 for PRMS and 0.78 for SAC. A similar spatial pattern of performance was also seen across all four model structures, with certain catchments resulting in poor or good simulations for all four model structures (Figure 43). Across all catchments, the average range in maximum NSE between the four models is only 0.08. This is far smaller than the range of maximum NSE gained by each model across all catchments.

The similar performance of theSimilarities in overall model performance models could be partially due to the models all being run at the same spatial and temporal resolution, having a similar model architecture splitting the catchment into upper and lower stores, and including the same process representations (such as lack of a snowmelt module). However, there are important differences between the models, which may be contributing to the differences seen in the decomposed metrics (Fig. 5). The architecture of the upper and lower model layers differs, as can be seen in Figure 3Fig. 2. TOPMODEL and ARNO/VIC have more parsimonious structures with only one store in each layer, while PRMS has a more complex upper layer which is split into multiple stores, and SACRAMENTO splits both upper and lower layers into multiple stores. The modelling equations

**Commented [RI42]:** R3: Page 16 Line 30 "P9 L15 – you haven't introduced the abbreviation "SAC""

**Commented [RI43]:** R2 Page 11 Line 15 "P9L21: saying "snowmelt module" implies that accumulation is simulated but melt is not, use "snow module" instead."

[revised manuscript text omitted]

**Commented [RL46]:** R2C7: Page 8  Line 22 "Comment 7: Just like hydrological behaviour, model performance is not determined by a single catchment characteristic, but rather, by the interaction of multiple catchment characteristics. So, firstly, would it be possible to consider a wider range of catchment attributes? So far, the authors employ the BFI, annual rainfall, the wetness index and the runoff coefficient, but many more attributes could be used to describe each catchment (e.g., Beck et al., 2015). I encourage the authors to add other attributes, which they might have computed for other studies or retrieved from the UK hydrometric register, which they mention in Table 1, in order to describe the landscape in a more complete fashion (indicators of human interventions would also be useful, see below).
Comment 8: And secondly, I think it would be beneficial to better account for the interactions between these attributes. The authors combine several attributes in Figure 7 to explain model performance, which I find particularly interesting. Maybe that the analyses they will perform when revising this study will lead to more figures of this type, and enable a more systematic analysis of the interactions between these predictors (perhaps using regression trees, see Poncelet et al., 2017). This is critical to go from describing where models fail and to explaining why they fail."

**4.5 Benchmarking Predictive Capability for Annual Maximum Peak Flows**

[revised manuscript text omitted]

Commented [RI49]: Removed as repetition of results

Commented [RI50]: R3C8: Page 13 Line 32 "Reading through your discussion seems very repetitive of the results chapter. Can these be better synthesised, to reduce the discussion section? 16. Your discussion is longer than the rest of the paper put together!" – we have removed this paragraph to reduce repetition of the results.

Commented [RL51]: R2C5: Page 7 Line 27 "Comment 5: Which process parameterisation are missing to capture the range of hydrological behaviours across the UK? The authors identify catchments in which the four model structures perform poorly, and reflect on characteristics of these catchments to which the poor performance can be attributed (e.g., chalk, snow, high human impacts). I suggest that the authors dedicate a subsection in the Discussion Section to these findings, which are relevant for both model development and selection."

Commented [RI52]: R3: Page 15 Line 15 "Comment 17. P13 L 3 – you've made no reference to anthropogenic influences in Scotland. This statements seems a bit throwaway."

Commented [RI53]: R3C17 Page 15 Line 15 "17. P13 L 3 – you've made no reference to anthropogenic influences in Scotland. This statements seems a bit throwaway."

Commented [RI54]: R3C18 Page 15 Line 19 "P13 L9 – it is not just the Thames basin that is affected by abstractions! A lot of Anglian region is VERY heavily influenced."

For catchments where groundwater is the reason for model failure, a possible solution could be to use a conceptual model that allows for groundwater exchange (as opposed to the models used here which all maintain the water balance). Hydrological models such as GR4J and SMAR have been developed with functions tohat allow represent models to gain or lose water, to represent inter-catchment groundwater flows (Le Moine et al., 2007). The use of these models where there is evidence of groundwater flows can help to improve model performance and reduce discrepancies between observed and simulated flows, but must be used with caution to avoid overfitting of the water balance where there is no physical reasoning for a catchment to be gaining or losing water. Whilst it has been noted that there is a general pattern of poor performance for catchments in southeast England, it is hard to disentangle the reasons that this may be the case. Both the underlying chalk geology causing water transfer between catchments, and heavily human modified flow regimes could explain model failures which are greatest during the summer. Interestingly, McMillan et al., (2016) found that whilst aquifer fraction was expected to have a strong link to model performance, no relationship was found for the TOPNET model applied in New Zealand. An approach such as this may help to improve simulations in catchments overlaying the chalk aquifer in southeast England for future modelling attempts.

**5.2 Insights from Applying an Ensemble of Model Structures Across a Large Sample of Catchments**

We found that the ensemble of model structures produced better results overall than any single model, showing that there is value in considering model structural uncertainty. Certain model structures were more likely to be considered behavioural, or resulted in the best sampled simulations, compared to others dependent on the catchment characteristics coupled to climatic conditions. This is clearly seen in Fig. 5,7,8 and 9. The ensemble of model structures was able to take advantage of this, as can be seen by the different proportion of the four model structures comprising the behavioural ensemble for different climate and catchment characteristics (Fig. 5). This supports previous research highlighting the importance of considering alternative model structures and using model structure ensembles (Butts et al., 2004; Clark et al., 2008; Perrin et al., 2001). Exploring the relative performance of the different model structures within an uncertainty framework has enabled us to identify scenarios where one model can become dominant or where all model structures are equally likely to generate behavioural simulations. Understanding why different model structures generally perform well for certain catchments can help us to identify parts of a model structure that may be particularly effective and can lead to model improvements. We found that for catchments with average annual rainfall values of around 2000mm/year or lower, the SACRAMENTO model structure is more dominant. As we move towards catchments with higher annual rainfall, the relative importance of the different structures shift until all structures are approximately equal for the catchments with the highest annual rainfalls. This shows that for very wet catchments, the model structure is less important as all models can produce behavioural simulations through some part of the parameter space, as seen by the relatively high number of behavioural simulations for wetter catchments (Fig. 5b). This agrees with previous studies, where models have been found to perform better for wetter catchments, which are likely to have more connected saturated areas, as there is a more direct link between rainfall and runoff (McMillan et al., 2016).

**Commented [RI55]:** R1: Page 4 Line 19 "authors discuss about groundwater flows between catchments, with losses or gain of waters. This problem is not new and some conceptual modelisation could be found in the literature since one or two decades. In a natural context, authors could make a reference to Le Moine at al. (2007, 2008) papers about groundwater flows and water balance closure. The existence of such groundwater flows in permeable geological context (chalk, limestones and/or karstic systems, etc.) was one of the reasons of the development of a groundwater exchange function within the GR model family. The use of this function should be motivated by (hydrogeologic) evidences of such groundwater flows (in order to avoid "overfitting" of the water balance, i.e. fudge factor), but might be useful in catchments where water balance is difficult to close, such as the one influenced by chalk aquifers in southeast england."

R2C5: Page 13 Line 8 "Catchment characteristics and climate – do all FUSE models maintain the water balance? Can you comment on the existence of models that don't (e.g. GR4J), and how those may overcome such problems? What are the implications of maintaining vs not maintaining water balance in conceptual lumped models? Are the four models you've chosen actually quite similar to each other? I think you need to make more of this somehow."

[revised manuscript text omitted]

**Commented [RI56]:** R3C8: Page 13 Line 32 "Reading through your discussion seems very repetitive of the results chapter. Can these be better synthesised, to reduce the discussion section? 16. Your discussion is longer than the rest of the paper put together!"

to perform better for wetter catchments, which are likely to have more connected saturated areas, as there is a more direct link between rainfall and runoff (McMillan et al., 2016).

5     Our results highlight the difficulty in national and large-scale modelling studies, which for GB must incorporate human modified hydrological regimes, complex groundwater processes, a range of different climates and the potential of dis- informative data, or at least a lack of process understanding to adjust model conceptualisations. Whilst simple, lumped hydrological models can produce adequate simulations for most catchments, the model structures are put under too much stress when trying to simulate catchments where the water balance does not close or is increasingly departing more normal conditions.

10   The models fail or produce poor simulations when large volumes of water enter or leave the catchment due to human activities or groundwater processes, indicating the importance of considering these influences in any national study. What is striking here in these results, is that general hydrological processes, defined by water availability and BFI metrics to infer the extent of slower flow pathways, are important in defining the quality of simulated output and differences in model structures and parameter ranges, even though nationally many catchments are impacted by additional anthropogenic activities such as

15   abstractions and multiple flow structures.

**5.4 Predictive Capability of Models for Predicting Annual Maximum Flows**

Predictions of annual maximum discharge using behavioural models based on Nash-Sutcliffe Efficiency (NSE) posed a larger challenge for the models, even when allowing for an estimate of observational uncertainty from results generalised in Coxon et al., (2015). It was found that all model structures systematically underpredicted annual maximum flows across most

20   catchments, which could have large implications if these structures were used for flood modelling or forecasting. These results are in line with previous large-scale modelling efforts. McMillan et al., (2016) report that their TOPNET model applied across New Zealand showed a smoothing of the modelled hydrograph relative to the observations, which resulted in overestimation of low flows and underestimation of annual maximum flows. Newman et al., (2015) found the same effect in their study covering 617 catchments across the US. This underestimation of peaks could be in part due to the use of NSE in selection of

25   the behavioural models. NSE is often used in flood studies, as it emphasises correct prediction of flood peaks relative to low flows (For example, Tian et al., 2013). However, NSE tends to underestimate the  overall variance  of the time-series, resulting in underprediction of floods and overprediction of low flows (Gupta et al., 2009).

It was found that there were some variations in the ability of models to simulate AMAX flows between years, and this often related to the wetness of a particular year. Models tended to perform worse in wetter years, and better in drier years. This could

30   be linked to the fact that all models tended to underestimate annual maximum flows, and therefore are closer to observations in years with lower annual maximum flows.

Commented [RI57]: R2C6 Page 8 Line 4 "How critical is the selection of model structure? There are cases of great equifinality (i.e., high NSE for all structures, mostly for humid catchments). As mentioned above, a high NSE is not a guarantee that the model structure is adapted, but as long as this is recognised (and this could be clearer throughout the manuscript), I think it is fine for this study. But in other (more interesting) catchments, some model structures clearly outperform other structures, and there, model choice is critical. I think this should be stressed more prominently, since these are cases in which the inadequacy of the model structure cannot be overcome by parameter tuning. Given the general tendency of using the same model structure across very diverse environments (as discussed e.g. by Addor and Melsen, 2019), I think this is an important result, which could be underscored more. A related question is: which catchment characteristics explain these large NSE differences between model structures?"

R3C20 Page 20 Line 29 "20. P13 L15 – "The ensemble of model structures was able to take advantage of this" - this seems to be a contradictory argument to the previous statement that the models all have similar performance to each other on a catchment by catchment basis. I think you need to tease these two arguments out better somehow. E.g. in some situations the choice of a different model can yield better results (e.g. high baseflow), but in other situations, none of the models can do well (e.g. abstractions). What are the implications of this? "

Commented [RI58]: R3C21: Page 16 Line 1 "21. P17 L 11-14 "We also evaluated model predictive capability for high flows, as good model performance in replicating the hydrograph, assessed using Nash-Sutcliffe efficiency, does not necessarily mean models are performing well for other hydrological signatures. We found that the FUSE models tended to underestimate peak flows, and there were variations in model ability between years with models performing particularly poorly for extremely wet years." – so what? What are the potential implications?"

Commented [RI59]: R1 Page 3 Line 3: "One of the main drawback of the NSE (and linear regression as well) is that the standard deviation of the simulated time-series is biased and underestimated, i.e. flood underestimated and drought overestimated. Among other arguments, this drawback partly explain why flood values are underestimated. I would add at least a comment on the fact that this statement is dependent on the behavioural model selection metrics in §5.4."

R2C5: Page 7 Line 27 "Can they formulate hypotheses on why annual maximum flows are underestimated, which could be tested by future studies?"

**5.5 4 Uncertainty Evaluation in Hydrological Modelling**

[revised manuscript text omitted]

> **Commented [RI65]:** R1: Page 5 Line 12 "In §2, I would give an estimation of the proportion of watersheds where snowmelt processes are observable (solid precipitation >20% of total precipitation ?)"

[Figure]

**Figure 3**: FUSE wiring diagram, showing the model structure decisions. TOPMODEL and ARNO/VIC have 10 parameters, PRMS has 11 parameters and SACRAMENTO has 12 parameters.. Adapted from Clark et al., (2008).

**Commented [RI66]:** R1: Page 5 Line 30 "In Figure 2, I would put the number of free parameters to calibrate."

[Figure]

**Figure 443: GB maps of model performance for each structure. Each point is a gauge location which is coloured based upon the best Nash Sutcliffe score attained by the model for that catchment.**

[Figure]

**Figure 5. GB maps of model performance for each structure for 3 different metrics. Top row shows model relative bias or relative error in simulated mean runoff (%), middle row shows relative error in the standard deviation of runoff (%), and the third row shows correlation between observed and simulated streamflow. Each point is a gauge location which is coloured based upon the best score for that metric.**

**Commented [RI67]:** R1 Page 2 Line 18: "Authors decided to use the classical Nash-Sutcliffe efficiency (NSE) index to evaluate model performances (and select behavioural models, NSE > 0.5). NSE index is famous and widely used in Rainfall-Runoff modeling. Even if the perfect efficiency index do not exists, this index is also known to have some drawbacks (Schaefli and Gupta, 2007, among many references). Gupta et al. (2009) introduced the Kling-Gupta efficiency index that allows to explicitly account for bias (mean and variability) and correlation, in the evaluation of model performances. Given the ambition of this paper, I would recommand the authors to consider in their analyses the Kling-Gupta efficiency index, or at least to decompose their results in terms of correlation and mean bias."

R2: Page 6 Line 30 "Since the authors aim to better understand "where and why these simple models may fail" the choice of NSE is somewhat suprising, since NSE is a measure of overall performance, which provides limited insights into the reasons for high or low performance. Although an evaluation based on hydrological signatures would have enabled a more process-based diagnostic of model failures, I am not requiring this, since it would imply significant additional analyses. However, if the authors stick to NSE (or use KGE), I suggest that they use benchmarks (as suggested by Seibert et al., 2018) to account for the fact that high NSE/KGE values can be relatively easy to reach depending on the catchment and the season. I believe this would enable a more fair and enlightening assessment of the hydrological models across the catchments."

[revised manuscript text omitted]

**Supplementary Information 1 – Plots looking at the relationship between catchment characteristics and model performance**

In the main body of the paper, we looked at the relationship between the catchment wetness index and runoff coefficient and model performance. This was selected as these variables strongly were related to model performance and explained differences between catchments. Here, we give additional plots looking at the relationship between model performance and many different catchment characteristics. These characteristics were either taken from the hydrometric register or calculated from the model input data timeseries (Centre for Ecology and Hydrology, 2016; Marsh and Hannaford, 2008a; Robinson et al., 2015a).

Figure S1 summarises the overall performance of the four models, and was used to help identify overall trends in model structure performance across all catchments. Figures S2 – S5 are scatter plots looking at the relationship between model performance (assessed using NSE, bias, error in standard deviation and correlation respectively) and different catchment attributes. Figures S6 onwards are plots looking at interactions between different catchment attributes and model performance.

From Figures S2-S5 we can see that Small catchments ($<200km^2$) tend to have more variable NSE scores (both high and low), whilst large catchments ($>3000km^2$) always do fairly well. This is seen with all the decomposed metrics – with small catchments more likely to have errors in bias and standard deviation of flows, and having more variable correlation scores. A possible explanation for this could be that daily data is less able to capture the variation in these catchments.
Baseflow dominated catchments (BFI > 0.7) are more likely to gain really low NSE values (although some high BFI catchments can be simulated well). Interestingly, BFI seems to have a relationship with error in the standard deviation, with baseflow dominated catchments the only catchments where the best simulations tend to overpredict variation. This can be explained, as groundwater would dampen variation in flows.
Gauge elevation seems to cap overall model performance – with higher elevation gauges unable to achieve performance scores as high as low elevation gauges. Or this could potentially be a pattern because there are fewer high elevation gauges. This could also be a problem with using NSE, as lower scores are naturally given to catchments with more seasonal variation. We see that for these high elevation gauges, the best simulations always underpredict the flow standard deviation.
Surprisingly, urbanisation does not seem to decrease model performance.

From figures S6 onwards we can see that the worst NSE performaning catchments are grouped being small catchments less than $120km^2$, with elevations below 125m, mid to high BFIs (>0.5), low annual rain less than 1000mm and annual runoff values which differ from other catchments with similar annual rainfall totals. Poor NSE ~0.5 is achieved for wetter catchments (annual rain > 1200mm), which have relatively low annual runoff generally less than 900mm. Many have flow attenuation from reservoirs and lakes, and for these catchments correlation is poor.
The largest problems with standard deviations seem to be small catchments, where standard deviation is generally underpredicted except for catchments with a high BFI where it is sometimes overpredicted. This can be explained as baseflow

dominated catchments may have less variation in flow so it is most likely that variation is overpredicted for these catchments. For small catchments relative errors are also most likely to be high.

Bias is very clearly linked to the climatic variables. There is a clear linear relationship with rainfall and runoff, and catchments deviating from this underpredict mean flows and have poorer correlations. Wetter catchments have reduced relative biases, which is likely to be due to biases being normalised by mean flow.

[Figure]

Figure S1. Cumulative distribution functions showing performance of the 4 model structures across all catchments, when assessed using NSE and decomposed metrics for the simulation with the highest NSE.

[Figure]

Figure S2: Relationship between NSE and a selection of 15 catchment descriptor variables. Column 1 is more general catchment attributes from the hydrometric register (Marsh and Hannaford, 2008). Column 2 gives hydroclimatic attributes calculated from our data, and proportion catchment is wet from the hydrometric register. Annual Loss is Rainfall-Runoff,

whilst Annual flood is the Median Annual maximum flood peak. Column 3 gives land-use and bedrock permeability descriptors, also from the UK hydrometric register.

[Figure]

Figure S3: Relationship between bias and a selection of 15 catchment descriptor variables, as in Figure S2.

[Figure]

Figure S4: Relationship between error in standard deviation and a selection of 15 catchment descriptor variables, as in Figure S2.

[Figure]

Figure S5: Relationship between correlation and a selection of 15 catchment descriptor variables, as in Figure S2.

[Figure]

Figure S6a: Relationship between general catchment characteristics, coloured by model ensemble NSE score for that catchment.

[Figure]

Figure S6b: Relationship between general catchment characteristics, coloured by model ensemble bias score for that catchment.

[Figure]

Figure S6c: Relationship between general catchment characteristics, coloured by model ensemble error in standard deviation for that catchment.

[Figure]

Figure S6d: Relationship between general catchment characteristics, coloured by model ensemble correlation for that catchment.

[Figure]

Figure 7a. Same as figure S6, but this time looking at hydroclimatic catchment descriptors.

[Figure]

Figure 7b. Same as figure S6, but this time looking at hydroclimatic catchment descriptors.

[Figure]

Figure 7c. Same as figure S6, but this time looking at hydroclimatic catchment descriptors.

[Figure]

Figure 7d. Same as figure S6, but this time looking at hydroclimatic catchment descriptors.

---

## Author Response (ED1)

**Benchmarking the predictive capability of hydrological models for river flow and flood peak predictions across a large sample of catchments in Great Britain**

**General response to reviewers**

We thank the reviewers for taking the time to read the manuscript, and for their thorough and insightful comments. Their suggestions have helped us to ensure our results are more useful to the modelling community.

The main comments from the reviewers were regarding (1) the use of Nash-Sutcliffe Efficiency to evaluate model performance, (2) making data more easily accessible, (3) synthesis of the introduction and discussion sections, and (4) plotting of additional catchment attributes and human influences on river flows.

In response to these reviewer comments, we have re-analysed the model output with consideration of additional performance metrics. We have supplied a complete set of outputs via a DOI. We have produced additional plots of factors affecting runoff across Great Britain, highlighting where streamflow is impacted by snowmelt and human influences in Figures 1 and 2. We have considered additional catchment attributes in supplementary information. The manuscript has also been revised, to synthesize the introduction and discussion, and to discuss the new metrics and factors affecting runoff plots in the methods and results sections.

Detailed responses to all reviewer comments are provided in bold below. We have also inserted the review comments as comments next to the relevant tracked changes in the manuscript.

Rosie Lane, June 2019

**Response to reviewer 1 (Thibault Mathevet)**

We thank Thibault Mathevet for taking the time to review our manuscript, and for his helpful comments. Our responses to each comment are outlined in bold below.

I carefully read the paper by Lane et al.. This paper appeared to be particularly clear, well written and easy to follow. Scope and objectives are stated clearly, the presentation of results is rather straightforward. As you probably know it, I appreciate this kind of study on a large sample of watersheds. I am very happy to know that such a large sample exists for GB. Studies on large sample give generality and robustness to the results. This paper gives insights on the general hydrology of GB and

10    predictive capabilities of 4 simple rainfall-runoff models. I really appreciated §4 and §5, particularly analyses linked to the seasonality (fig 4), BFI (fig 5, 6), and water balance closure (fig 7). Thanks to this large sample of watersheds in GB with a variety of hydrologic/ hydrogeologic functioning (even in the same country), these results appear to be robusts, with a general interest. The link between BFI (main underground processes) and model structure agility is really interesting.

**Response: We thank Thibault Mathevet for taking the time to thoroughly review our manuscript, and for his positive**

15    **comments.**

**Main comments:**

Evaluation of model performance and selection of model :

20    Authors decided to use the classical Nash-Sutcliffe efficiency (NSE) index to evaluate model performances (and select behavioural models, NSE > 0.5). NSE index is famous and widely used in Rainfall-Runoff modeling. Even if the perfect efficiency index do not exists, this index is also known to have some drawbacks (Schaefli and Gupta, 2007, among many references). Gupta et al. (2009) introduced the Kling-Gupta efficiency index that allows to explicitly account for bias (mean and variability) and correlation, in the evaluation of model performances. Given the ambition of this paper, I would recommend

25    the authors to consider in their analyses the Kling-Gupta efficiency index, or at least to decompose their results in terms of correlation and mean bias.

**Response: The NSE index was chosen for this analysis as it is so widely used and easy to interpret. Given our focus on floods, it is also a good choice as it emphasizes the fit to peaks more than KGE which focuses on balancing the contribution of the bias and correlation. However, we agree that there are drawbacks to only using the NSE index and**

30    **so following this comment, we have provided additional analysis looking at the correlation, variance and bias. This can be seen in Figures 5 and 9.**

Poor performances on floods :

Authors found that the different models had poor performances on floods, which is generally the case when classical modeling schemes are used to optimise or select parameter sets. I appreciate the simple way authors evaluate models on flood values, however I would add a figure to explain the two metrics. One of the main drawback of the NSE (and linear regression as well) is that the standard deviation of the simulated time-series is biased and underestimated, i.e. flood underestimated and drought

5 overestimated. Among other arguments, this drawback partly explain why flood values are underestimated. I would add at least a comment on the fact that this statement is dependent on the behavioural model selection metrics in §5.4. If authors update their paper using KGE to select their behavioural models, they might revise (a bit) their findings on model performances for floods.

**Response: Thank you for pointing this out. In response to this comment, we have clarified the explanation of flood**

10 **metrics. We selected NSE as it emphasizes the fit to peaks, whilst KGE is more general, but we acknowledge that it has drawbacks and no global performance measure is useful in all situations, especially when looking at extremes. We therefore decided to keep using NSE, but added a comment in the discussion on how behavioural model selection metrics influence estimation of flood values.**

15 Focus on droughts ? :

Given the ambition of this paper, I think that this paper would also benefit from a focus on droughts. Hence, analyses on droughts could be complementary to analyses on relative model performances (among the 4 tested structures), since droughts might also be driven by BFI in GB ? The link with groundwater flows could also be shown, if a focus on droughts is done. Authors could use the same metrics as for floods. It could be better to use the 10 days or 30 days annual minimal value, instead

20 of the annual minimal value, which could be highly impacted and uncertain.

**Response: We agree that focusing on droughts could be an interesting question in itself, however we feel that it is out of scope for this paper. We are aiming to give a general overview of the capability of models, with a focus on high flows. Drought and very low flows is a more complex problem to address, and more likely to be influenced by human impacts in managed catchments. We therefore think adding this would be too much for one paper. We plan further research**

25 **which better incorporates human influences on low flow river totals and thus will make such an assessment more fruitful.**

**Minor comments :**

In §1.1 : authors discuss the benefits of national scale hydrological modelling. Another benefits could be the production of

30 parameter libraries, which could be used for regional studies or model calibration on poorly gauged to ungauged basins or engineering studies. Authors can make references to papers on this subject (Perrin et al., 2008 ; Rojas-Serna et al., 2016 ; or some other works by Seibert).

**Response: Thank you for this idea, we have referred to parameter libraries in the introduction, and have added tables of best parameter sets made available through a DOI.**

In §5.1 : authors did not use a snow accumulation and melt routine in their modeling framework. Very simple snow routine are available, in the spirit of the simple models proposed in FUSE. The CemaNeige routine could be a good candidate to improve model simulations on the few catchments where it's necessary. Depending on the proportion of snow impacted catchments, using a snow routine would improve model performances and the paper, as it could give answers to some hypotheses of the paper.

Valéry, A., Andréassian, V., Perrin, C., 2014. 'As simple as possible but not simpler': What is useful in a temperature-based snow-accounting routine? Part 1 – Comparison of six snow accounting routines on 380 catchments, Journal of Hydrology, 517(0): 1166-1175.

**Response: Thank you for this comment, but we do not think it would be feasible to run all simulations again with a snow routine. We originally decided not to use a snow routine as only a relatively few catchments were snow impacted. To check this, we have calculated snow fractions for all catchments, as the sum of the rainfall on days when daily mean temperature is less than 0 degrees Celsius divided by the total sum of the rainfall for the whole time period. This confirms that only a small proportion of catchments are snow impacted (13 catchments out of the 1127 have a snow fraction of more than 10%, and no catchments have a snow fraction of more than 17%). We have plotted these snow fractions in Figure 1, demonstrating that only a small proportion of catchments are impacted by snow. As the concept of the paper is focused on benchmarking the capability of these lumped models, and not model development, we feel that addition of a snow routine is out of scope.**

In §5.3 : authors discuss about groundwater flows between catchments, with losses or gain of waters. This problem is not new and some conceptual modelisation could be found in the literature since one or two decades. In a natural context, authors could make a reference to Le Moine at al. (2007, 2008) papers about groundwater flows and water balance closure. The existence of such groundwater flows in permeable geological context (chalk, limestones and/or karstic systems, etc.) was one of the reasons of the development of a groundwater exchange function within the GR model family. The use of this function should be motivated by (hydrogeologic) evidences of such groundwater flows (in order to avoid "overfitting" of the water balance, i.e. fudge factor), but might be useful in catchments where water balance is difficult to close, such as the one influenced by chalk aquifers in southeast england.

**Response: Thank you for highlighting these interesting and very relevant papers. We have added this into the discussion. However we have not yet done a comprehensive analyses of gaining and losing streams in the UK aquifer systems. This is indeed research that our group is currently conducting in more detail (separate PhD on improving ground water representation in models). Certainly from our preliminary analyses it is very difficult to attribute these losses and gains, and especially for lumped catchment model behaviours where the spatial partitioning needed might be too abstract to incorporate in the model outputs.**

**Last comments :**

P4, l22 : I would also make a reference to Perrin et al. 2001 here

**Response: We agree this is a relevant paper, it has been added.**

5  P7, l20 : mistake with O (mean of observed discharge)

**Response: Thank you for noticing this, it has been corrected.**

P10, l22 : values instead of vales

**Response: This has been corrected.**

P17, l8 : for catchments, repeated 2 times

**Response: The repetition has been removed.**

In §2, I would give an estimation of the proportion of watersheds where snowmelt processes are observable (solid precipitation

15  >20% of total precipitation ?)

**Response: We agree that this would be useful and have added a map of snow fractions to figure 2 which we refer to in section 2.**

Table 1 is not cited within §2

20  **Response: Thank you for spotting this, we have now added the citation: "The catchments cover all regions and include a wide variety of catchment characteristics including topography, geology and climate (see Table 1)."**

In §3.3, the +/- 13% concerning streamflow uncertainties for flood should be a bit more explained. To which probability range this uncertainty refers ? Is it one or two standard deviation (or something else) ?

25  **Response: The +/-13% represents the 95th percentile range of the discharge uncertainty bounds and was chosen as a representative discharge uncertainty for annual maximum flows from a national analysis of discharge uncertainties (Coxon et al, 2015). We have better clarified this, with the text now reading "This observed error value was selected following previous research on quantifying discharge uncertainty at 500 UK gauging stations for high flows, and represents the average 95$^{th}$ percentile range of the discharge uncertainty bounds for high flows (Coxon et al., 2015;**

30  **Mcmillan et al., 2012)."**

In Figure 2, I would put the number of free parameters to calibrate.

**Response: The following has been added to the figure caption "TOPMODEL and ARNO/VIC have 10 parameters, PRMS has 11 parameters and SACRAMENTO has 12 parameters. ".**

**Response to reviewer 2 (Anonymous)**

Comment 1: This study compares four structures from the framework FUSE in 1100 UK catchments. This is, in itself, a significant achievement. The authors highlight which structures perform best in different regions (Results Section) and then discuss more generally why models fail and which improvements would be necessary to improve performance (Discussions Section). I think that, ultimately, the goal of such model intercomparison is to provide guidance on i) model selection (i.e., can specific models/modules be recommended based on basin attributes?) and ii) model development (i.e., are there specific process parameterisations that are currently missing, but are needed to improve the simulations?). In my view, the latter point is addressed quite well (although I suggest restructuring the text to makes these results stand out more, and to go beyond FUSE structures by discussing modelling decisions more generally) but the former point could be addressed in a more systematic and comprehensive way. Overall, I consider that, after revisions, this paper has the potential to become a timely and welcome addition to the literature.

**Response: We thank reviewer 2 for these helpful comments, and for taking the time to review our manuscript.**

**Major comments:**

Comment 2: Model intercomparison vs. benchmarking: Since the authors use the term "benchmarking" in the title and throughout the manuscript, I encourage them to clarify in the introduction what differentiates model benchmarking from model intercomparison. As the authors compare FUSE structures with each other, isn't their study rather a model intercomparison? Do the authors mean that their runs can be used as benchmark by future studies, as suggested on P12L12? Please clarify.

**Response: We have included clarification of this in the introduction, including more explanation on how the performance of simple hydrological models can be used as a benchmark and making it clear that our results can be used as a benchmark. We used the term 'benchmark' to highlight that these results can be used as an indicator of the ability of lumped models, which future studies may use when evaluating the performance of other models (that are perhaps more complex or include additional processes). For example, our results would inform a modeller that gaining an NSE of 0.7 in SE England is a good achievement, whereas gaining the same score in west Wales is not an achievement as most models can easily gain higher NSE scores for these catchments. The use of simple models as benchmarks has been advocated in previous studies, for example Seibert et al., (2018).**

**Seibert, J., Vis, M. J., Lewis, E., & Meerveld, H. J. (2018). Upper and lower benchmarks in hydrological modelling. Hydrological Processes.**

Comment 3: Model evaluation using NSE: Since the authors aim to better understand "where and why these simple models may fail" the choice of NSE is somewhat suprising, since NSE is a measure of overall performance, which provides limited insights into the reasons for high or low performance. Although an evaluation based on hydrological signatures would have enabled a more process-based diagnostic of model failures, I am not requiring this, since it would imply significant additional

5 analyses. However, if the authors stick to NSE (or use KGE), I suggest that they use benchmarks (as suggested by Seibert et al., 2018) to account for the fact that high NSE/KGE values can be relatively easy to reach depending on the catchment and the season. I believe this would enable a more fair and enlightening assessment of the hydrological models across the catchments.

**Response: We originally selected NSE as it is a widely used and easy to interpret measure of performance. However,**

10 **we agree that in order to better understand model failures we will need to consider additional measures of performance. Therefore, we plan to also present correlation, variance and mean bias, as called for by the first reviewer, to support the seasonal analysis of model performance that we have already carried out. As our focus is on reasons for model failures, we feel that these additional decomposed metrics will be more informative than the use of benchmarks.**

[Figure]

15 Comment 4: Relevance for the broad hydrological modelling community: A challenge here is to provide guidance for model selection, which is also relevant for modellers not using FUSE. Overall, the most interesting question is not really which FUSE model performs best, but why. I encourage the authors to discuss and highlight specific model elements that contribute to poor/good simulations, rather than focussing FUSE models themselves (e.g., TOPMODEL or PRMS). For instance, the fact that ARNO-VIC performs particularly well in high-BFI catchments is only an intermediary result, which is mostly relevant to

20 FUSE users. The reasons why this is the case (e.g., last paragraph of Section 5.2), on the other hand, are relevant to a much wider group. I suggest a stronger emphasis on modelling decisions, as opposed to FUSE models, in particular in the most critical parts of the manuscripts (abstract and conclusions).

**Response: We agree that highlighting specific model elements that contribute to poor/good simulations would be of great use to the broad hydrological modelling community. Where possible, we have tried to outline modelling decisions**

25 **that may cause differences in the results. However, it is difficult to distinguish which model elements are causing good/poor model performance, as the model structures differ in multiple aspects, and further analysis would be required to fully explore which modelling decisions are contributing to good/poor simulations. We have however added an extra table explaining which modelling decisions were applied for each FUSE model, highlighting the different model elements.**

Comment 5: Which process parameterisation are missing to capture the range of hydrological behaviours across the UK? The authors identify catchments in which the four model structures perform poorly, and reflect on characteristics of these catchments to which the poor performance can be attributed (e.g., chalk, snow, high human impacts). I suggest that the authors

dedicate a subsection in the Discussion Section to these findings, which are relevant for both model development and selection. Can they formulate hypotheses on why annual maximum flows are underestimated, which could be tested by future studies?

**Response: We have dedicated a section of the discussion to "Identifying missing process parameterisations" in response to this comment. As suggested by the first reviewer, the choice of NSE could result in underestimation of flood peaks,**
5 **and we have therefore commented in the discussion on how our choice of metrics could be a factor leading to the underestimation of flood values.**

Comment 6: How critical is the selection of model structure? There are cases of great equifinality (i.e., high NSE for all structures, mostly for humid catchments). As mentioned above, a high NSE is not a guarantee that the model structure is
10 adapted, but as long as this is recognised (and this could be clearer throughout the manuscript), I think it is fine for this study. But in other (more interesting) catchments, some model structures clearly outperform other structures, and there, model choice is critical. I think this should be stressed more prominently, since these are cases in which the inadequacy of the model structure cannot be overcome by parameter tuning. Given the general tendency of using the same model structure across very diverse environments (as discussed e.g. by Addor and Melsen, 2019), I think this is an important result, which could be underscored
15 more. A related question is: which catchment characteristics explain these large NSE differences between model structures?

**Response: We explored the importance of model structure selection in figures 4, 7, 8, 10 and 11, with figure 7 looking at catchment characteristics which were related to differences between the model structures. However, we agree that the question of how critical the selection of model structure is for different catchments was not well addressed in the manuscript. Therefore, we have clarified the discussion of this in the discussion section.**

[Figure]

This leads me to a set of comments related to the use of catchment attributes to explain model performance.

Comment 7: Just like hydrological behaviour, model performance is not determined by a single catchment characteristic, but rather, by the interaction of multiple catchment characteristics. So, firstly, would it be possible to consider a wider range of catchment attributes? So far, the authors employ the BFI, annual rainfall, the wetness index and the runoff coefficient, but
25 many more attributes could be used to describe each catchment (e.g., Beck et al., 2015). I encourage the authors to add other attributes, which they might have computed for other studies or retrieved from the UK hydrometric register, which they mention in Table 1, in order to describe the landscape in a more complete fashion (indicators of human interventions would also be useful, see below).

Comment 8: And secondly, I think it would be beneficial to better account for the interactions between these attributes. The
30 authors combine several attributes in Figure 7 to explain model performance, which I find particularly interesting. Maybe that the analyses they will perform when revising this study will lead to more figures of this type, and enable a more systematic analysis of the interactions between these predictors (perhaps using regression trees, see Poncelet et al., 2017). This is critical to go from describing where models fail and to explaining why they fail.

**Response: We selected the attributes of BFI, annual rainfall, wetness index and runoff coefficient as they were observed to have the largest impact on model performance. In response to reviewer comments, we have created additional plots of snow fraction, and factors affecting runoff on catchments across GB which are given in Figures 1 and 2. We do not want to add many more figures into the manuscript as we feel that this may detract from the main messages of the paper, but have added additional plots looking at interactions between attributes as supplementary information. Also we believe the current analyses are in keeping with the abstract nature of lumped modelling systems where a greater range of catchment attributes might only be loosely related to the structure and parameterisation of the model design. We aim to explore these issues with more spatially orientated modelling approaches in future publications.**

Comment 9: Anthropogenic activities are repeatedly mentioned to explain poor model performance (e.g., P12L29, P14L16, P15L3). This is indeed plausible, but if qualitative or maybe quantitative indicators of the extent of human interventions could be included, so that their impacts on streamflow and model performance could be demonstrated or maybe even quantified, it would strengthen the study.

**Response: We agree that this is required to strengthen comments made regarding reasons for model failures. We have information on factors affecting runoff for all catchments in the hydrometric register. However, this only gives an indicator of which factors may affect runoff, and not to what extent, and therefore we decided not to include it in the original manuscript. In response to reviewer comments, we have added plots of factors affecting runoff in Figure 1.**

**Minor comments:**

Comment 10: I find the introduction too long. It attempts to cover too much material, and hence ends up being too general and its different parts are not very well connected. I suggest that the authors focus on what is really necessary to introduce their study, transfer parts of the text to the rest of the paper (e.g. the methods), and delete the rest.

**Response: We agree. We have shortened the introduction, by combining and shortening sections on large sample and national hydrology, and condensing the introduction of modelling uncertainties by moving sentences to the methods section or deleting where appropriate.**

Comment 11: Outlook: it might good to mention that, although this study focusses on four FUSE models, it is possible build additional FUSE model to transition progressively from one model to the next, and establish which modelling decisions contribute most to the differences in the simulations.

**Response: The following has been added, "The framework allows the user to select different combinations of modelling decisions, starting with four parent models based on the structures of widely used hydrological models, and allowing the user to combine these decisions to create over 1200 different model structures".**

Comment 12: Data availability: "This study provides a useful benchmark of the performance and associated uncertainties of four commonly used lumped model structures across GB, for future model developments and model types to be compared against". I agree. But then, I think that instead of saying that "All model outputs from this study are available upon request from the lead author", the authors should make the runs available online, and provide the doi, before the paper is published. This is expected by AGU journals, and I think it is good practice in order to avoid data loss.

**Response: We completely agree with this, and have provided a DOI for the data.**

**Other suggested changes:**

Title: the field is "large-sample hydrology", but here it should be "large sample"

**Response: Thank you, this has been changed to "over 1000 catchments" as it is more informative than "large sample".**

P1L15: add "and support model selection"

**Response: This has been added.**

P2L13: such as

**Response: Thank you for spotting this, this sentence has now been re-phrased.**

P2L29: impacted by what?

**Response: We have clarified and re-written the sentence to say "These have great benefits, as applying a consistent methodology across a large area enables comparison between places and identification of areas that may be at most risk of future hydrological hazards. "**

P4L12-17: this belongs to Data and Methods

**Response: These sentences have been moved to the data and methods section.**

P4L20: I suggest removing "(i.e. the number of storage components)" as it an arbitrary measure of complexity.

**Response: This has been removed.**

P5L22: discharge

**Response: This is referring to all the catchment data – we refer to discharge specific data at a later point in the methods section.**

P6L5: please define "sufficient"

5   **Response: This was explained in the following sentence. We have re-arranged these sentences to make this clearer, now saying "Of these, 1013 had sufficient information (defined as more than 10 years of available discharge data during the model evaluation period) available to include in this analysis."**

P7L2: I suggest mentioning here that none of these four models includes a snow Routine

10   **Response: We have added this, "They all close the water balance, have a gamma routing function and include the same processes, for example none of the models have a snow routine or vegetation module."**

P7L4: please define "dynamically different" and what makes them "equally plausible"

**Response: By "dynamically different" we meant that the models all represent the landscape in a different way, and have quite different and distinct structures as shown in figure 3. By "equally plausible" we are referring to the fact that**

15   **we have no reason to expect one structure to behave better than the others, as all model structures are equally complete in terms of processes and all based on widely applied model structures. We have clarified this in the text by saying, "this leads us to believe that the model structures are dynamically different, as they are representing hydrological processes in different ways, yet as all are based on widely used hydrological models they are equally plausible and we have no a priori expectations that one model should outperform the others ".**

P8L13: please be more explicit about how this 13

**Response: This has been further explained with the text now reading, "This observed error value was selected following previous research on quantifying discharge uncertainty at 500 UK gauging stations for high flows, and represents the average 95$^{th}$ percentile range of the discharge uncertainty bounds for high flows."**

P9L21: saying "snowmelt module" implies that accumulation is simulated but melt is not, use "snow module" instead.

**Response: We agree, and have changed "snowmelt module" to "snow module."**

**Response to reviewer 3 (Anonymous)**

This paper provides a detailed investigation into the performance of four lumped conceptual models over large number of catchments in the UK. It demonstrates some very interesting findings, such as the fact that all four models have very similar performance on a catchment-by-catchment basis, and that only one of the models is deemed suitable for catchments with very high BFI. This paper is generally well written, set out and easy to follow, and the graphics provided assist the reader well in the interpretation of the results, I particularly like Figures 5 and 7. The discussion section should be synthesised as it feels repetitive of the results section. Overall, I feel that the motivations of the research, and the implications of the results are not very well reasoned. The authors need to think a bit more carefully about how others may make use of these results, and in particular, should publish the model performance scores as supplementary information (see my comments below).

**Response: We would like to thank the reviewer for taking the time to read the paper in depth, and for their constructive comments.**

Comment 1. You've "benchmarked" performance, but you haven't provided these benchmarks. If I were to now go and simulate a UK catchment, I still cannot easily compare my results with yours to see if I have a better model. For you to have achieved your aims, I would expect a supplementary table of the best scores the models achieved in each catchment, and the parameter values that produced them.

**Response: We completely agree with this, and the results can now be accessed through a DOI.**

Comment 2. Section 3.2 – why NSE?

**Response: We originally selected NSE as it is a widely used and easy to interpret measure of performance. However, as noted by the other reviewers, in order to better understand model failures we will consider additional metrics. Therefore, we plan to also present correlation and mean bias.**

[Figure]

Comment 3. Section 3.2 – "results are stored for a number of additional metrics not reported here". Stored where? Why would I care about this if you haven't made them available to me? I suggest you summarise these additional metrics in supplementary information. This may also address the issue of only reporting on NSE here.

**Response: We have removed this sentence from the methods and included additional metrics in the results which will also be provided thorough the DOI.**

Comment 4. Your statement in the abstract L23 that NSE scores of 0.72-0.78 were achieved for all catchments is misleading. How useful a measure is the "median maximum NSE for all the catchments"? It's pretty cryptic. There are catchments in E Scotland, and Anglian region that are showing pink/red for all 4 models, so NSE must be <0.5. Having got to page 9 I now see

what you meant, but it isn't clearly stated. The sentences on P12 L16-17 are a better summary of the performances across catchments. Same issue on P16 L 32.

**Response: Thank you for pointing out that this is not clear. We have replaced the statement in abstract L23 with "Our results show that simple, lumped hydrological models were able to produce adequate simulations across most of Great Britain, with each model producing simulations exceeding 0.5 Nash Sutcliffe efficiency over at least 80% of catchments."**

Comment 5. Catchment characteristics and climate – do all FUSE models maintain the water balance? Can you comment on the existence of models that don't (e.g. GR4J), and how those may overcome such problems? What are the implications of maintaining vs not maintaining water balance in conceptual lumped models? Are the four models you've chosen actually quite similar to each other? I think you need to make more of this somehow.

**Response: Yes, all the FUSE models used in this study maintain the water balance, and we have clarified this in the methods section. To address these questions we have added a paragraph in the discussion on how models that do not maintain the water balance have been used to improve modelling in groundwater dominated regions. In response to reviewer 1, this includes discussion of papers by Le Moine at al. (2007, 2008) about groundwater flows and water balance closure.**

Comment 6. P8 L23 – only a 1 year warm up period? This is not sufficient for many GW dominated catchments in the SE.

**Response: Thank you for this advice. We initially selected 1 year, as it is often considered sufficient for simple, lumped models such as the FUSE models. However, following this comment we carried out additional analysis of the simulated flows and found that whilst 1 year is a long enough warmup period for many catchments, it did not appear sufficient for some of the catchments in the SE as suggested. We will have increased the warmup period to 5 years, re-analysed the data and re-made all the figures to reflect this.**

Comment 7. P6 L6 – 2 years of data was your criteria for catchment selection, this doesn't seem sufficient to me

**Response: We originally aimed to keep as many catchments as possible for the analysis. However, you are correct that 2 years of data is not long for model evaluation. We have now added a tougher criteriona for catchment selection, of more than 10 years of available discharge data during the model evaluation period. The figures have been re-made to reflect this.**

Comment 8. Reading through your discussion seems very repetitive of the results chapter. Can these be better synthesised, to reduce the discussion section?

16. Your discussion is longer than the rest of the paper put together!

**Response: We have reduced the length of the discussion section, and re-structured the old sections 5.1-5.3 to reduce repetition.**

Comment 9. P2 L32 "a national scale model" – you're talking about applying a catchment model nationally. Can this be classified a national scale model?

**Response: In this section we were aiming to discuss the importance of national scale modelling more generally, suggesting that our work could be informative for evaluation of a national scale model. We were not saying that our application of a catchment model across GB was a national scale model.**

Comment 10. P3 L16 - "Secondly, evaluating more complex hydrological models relative to benchmark performance of simple models ensures that the relative difficulty of simulating different catchments is implicitly considered (Seibert et al., 2018)." I don't think I understand what you're saying here.

**Response: This has been re-phrased and further explained to make the meaning clearer. It now reads "Secondly, lumped hydrological models provide a good benchmark for evaluating more complex models, as they give an indication of what it is possible to achieve for a specific catchment and the available data (Seibert et al., 2018). This can help us identify whether a model is performing well in a catchment relative to how it should be expected to perform for the particulars of that catchment. For example, if a modeller gains an efficiency score of 0.7 for their model in a specific catchment, it is subjective whether this is a good or poor performance. However, if lumped, conceptual models tend to have efficiency scores of around 0.9 for that catchment then the modeller knows that their model is performing poorly relative to what is possible."**

Comment 11. P10 L22-24 – "For very low values of the ARNO-VIC 'b' exponent (AXV_BEXP) as seen for high BFI vales in Fig. 6 for behavioural model distributions means that only at very high, near full upper storage levels is any larger extent of saturated areas predicted" – I don't follow this sentence either.

**Response: This has been re-phrased.**

Comment 12. P8 L3 - Can you explain conditional probabilities in more detail?

**Response: We have extended this paragraph, now saying "Conditional probabilities were assigned to each behavioural parameter set based on their behavioural Efficiency score, and these were normalised to sum to 1. This meant that the simulations which scored the highest efficiency value had larger conditional probabilities, and simulations which had efficiency values just above 0.5 would have very low conditional probabilities. For each daily timestep, a 5th, 50th and 95th simulated discharge bound was produced from these conditional probabilities, for each catchment and model structure individually as described in Beven and Freer (2001). This meant that simulations with a higher efficiency score were given a higher weighting when producing the discharge bounds." Simply the behavioural weights**

(probabilities) assigned to each model are conditional on the choices made in the modelling exercise, here dependent on the sample design, the choice of parameter ranges, the model performance metric, and hence conditional.

Comment 13. P11 L 23 – "the top row of plots" – there is only one plot in Fig 8!

**Response: Thank you for noticing that! We had originally displayed figures 8 and 9 as a single plot. This has been corrected.**

Comment 14. P12 L 7-8 "However, variations between years are less apparent when looking at 25th and 75th percentiles in Fig. 8." We can't distinguish variation between years from Fig 8?

**Response: Again, thank you for noticing this, it has been corrected to point to the right figure.**

Comment 15. Please provide more sensible y axis labels for fig 8 and 9, e.g. "AMAX discharge score", and "AMAX percentage overlap" respectively. Multiply Fig 9 y axis by 100 to make it an actual percentage value, as you have referred to it as such in the text.

**Response: We agree with this comment and have changed the figure.**

Comment 17. P13 L 3 – you've made no reference to anthropogenic influences in Scotland. This statements seems a bit throwaway.

**Response: We have removed this sentence.**

Comment 18. P13 L9 – it is not just the Thames basin that is affected by abstractions! A lot of Anglian region is VERY heavily influenced.

**Response: we have changed this sentence to "a considerable proportion of river discharges throughout the Anglian region are abstracted."**

Comment 19. P13 L12 "we found that the ensemble of model structures produced better results overall than any single model" – can you validate that statement from your figures?

**Response: This can not be directly validated in a specific figure, but it can be seen across the figures, especially looking at Figure 7, where we see that no single model produces good results for all catchments.**

Comment 20. P13 L15 – "The ensemble of model structures was able to take advantage of this" - this seems to be a contradictory argument to the previous statement that the models all have similar performance to each other on a catchment by catchment basis. I think you need to tease these two arguments out better somehow. E.g. in some situations the choice of a

different model can yield better results (e.g. high baseflow), but in other situations, none of the models can do well (e.g. abstractions). What are the implications of this?

**Response: We have clarified these arguments in the discussion.**

[Figure]

5    Comment 21. P17 L 11-14 "We also evaluated model predictive capability for high flows, as good model performance in replicating the hydrograph, assessed using Nash-Sutcliffe efficiency, does not necessarily mean models are performing well for other hydrological signatures. We found that the FUSE models tended to underestimate peak flows, and there were variations in model ability between years with models performing particularly poorly for extremely wet years." – so what? What are the potential implications?

[Figure]

10    **Response: We have added discussion about the implications for flood modelling and forecasting here.**

**Typos and grammar:**

1. P2 L27 – CAMELS and MOPEX datasets (what are they datasets of?)

**Response: This sentence has been clarified - "the CAMELS or MOPEX hydrometeorological and catchment attribute**
15    **datasets."**

3. P5 L22 - remove "Environment Agency", a catchment is a catchment, the EA don't own the catchments, even if they do own the gauges!

**Response: "Environment Agency" has been removed.**

4. Amend "Rainfall is highest in the West and North of GB and lowest in the East and South varying from a minimum of 500mm to a maximum of 4496mm per year (see Fig. 1)" to "On average, rainfall is highest in the north and west of GB, and lowest in the south and east, with GB totals varying from a minimum of 500mm to a maximum of 4496mm per year (see Fig. 1).

25    **Response: Thank you for the suggestion, this sentence has been amended.**

2. P5 L14 - "these" should be "those"

5. P6 L1 - remove the "of" after "South-East"

6. P6 L12 – they are the "UK Met Office" not the "UK Meteorological Office".

30    7. P6 L14 and L20 – replace "laid" with "lay"

8. L6 L21 and elsewhere – "data" is plural, and should be followed by "were" instead of "was"

9. P8 L15 – "observational uncertainty certainty bounds" huh?? Can you not just remove the word certainty here?

10. P9 L15 – you haven't introduced the abbreviation "SAC"

11. P10 L5 – I'd call that northeast Scotland, not central Scotland

12. P11 L29 – "behavioural model" should be "behavioural models"

13. P13 L7 – do you mean model "structures"?

14. P16 L17 – "we also shown how"

15. P16 L19 – refer to Fig 6

**Response: Thank you for spotting these typos and grammatical errors, we have corrected these in the manuscript.**

16. P16/17 – "The performance of the four models was similar, and all models showed similar spatial patterns of performance, and there was no single model that outperformed the others across all catchment characteristics and for both daily flows and peak flows." – and, and, and

**Response: This sentence has been improved to "The performance of the four models was similar, with all models showing similar spatial patterns of performance, and no single model outperforming the others across all catchment characteristics for both daily flows and peak flows."**

17. P17 L8 – "we found models performed poorly for catchments for catchments with unaccounted losses"

**Response: we have removed the repetition.**

HESS REVIEW CHECKLIST

1. Does the paper address relevant scientific questions within the scope of HESS? Yes

2. Does the paper present novel concepts, ideas, tools, or data? Yes

3. Are substantial conclusions reached? Nearly, the wider implications, and utility of the research need to be better considered

4. Are the scientific methods and assumptions valid and clearly outlined? Yes

5. Are the results sufficient to support the interpretations and conclusions? Yes

6. Is the description of experiments and calculations sufficiently complete and precise to allow their reproduction by fellow scientists (traceability of results)? Yes

7. Do the authors give proper credit to related work and clearly indicate their own new/original contribution? Yes

8. Does the title clearly reflect the contents of the paper? Yes

9. Does the abstract provide a concise and complete summary? Yes

10. Is the overall presentation well-structured and clear? Yes

11. Is the language fluent and precise? Yes

12. Are mathematical formulae, symbols, abbreviations, and units correctly defined and used? Yes

13. Should any parts of the paper (text, formulae, figures, tables) be clarified, reduced,

combined, or eliminated? Yes, the discussion should be reduced

14. Are the number and quality of references appropriate? Yes

15. Is the amount and quality of supplementary material appropriate? No

**Response: Thank you for this largely positive summary checklist. We have addressed points 3 and 13 through our changes to the discussion section, and point 15 by making our output data available through a DOI.**

**Benchmarking the predictive capability of hydrological models for river flow and flood peak predictions across over 1000 catchments in Great Britain**

Rosanna A. Lane[1], Gemma Coxon[1], Jim E. Freer[1,3], Thorsten Wagener[2,3], Penny J. Johnes[1,3], John P. Bloomfield[4], Sheila Greene[5], Christopher J. A. Macleod[6], Sim M. Reaney[7]

[1]School of Geographical Sciences, University of Bristol, Bristol, BS8 2NQ, United Kingdom
[2]Faculty of Engineering, University of Bristol, Bristol, BS8 2NQ, United Kingdom
[3]Cabot Institute, University of Bristol, Bristol, BS8 2NQ, United Kingdom
[4]British Geological Survey, Maclean Building, Wallingford, OX10 8BB, United Kingdom
[5]Trinity College Dublin, Dublin, Ireland
[6]The James Hutton Institute, Craigiebuckler, Aberdeen, AB15 8QH, United Kingdom
[7]Department of Geography, Durham University, Durham, DH1 3LE, United Kingdom

*Correspondence to*: Rosanna A. Lane (R.A.Lane@bristol.ac.uk)

**Abstract.** Benchmarking model performance across large samples of catchments is useful to guide model selection and  future model development. Given uncertainties in the observational data we use to drive and evaluate hydrological models, and uncertainties in the structure and parameterisation of models we use to produce hydrological simulations and predictions, it is essential that model evaluation is undertaken within an uncertainty analysis framework.

Here, we benchmark the capability of several lumped hydrological models across Great Britain, by focusing on daily flow and peak flow simulation. Four hydrological model structures from the Framework for Understanding Structural Errors (FUSE) were applied to over 1000 catchments in England, Wales and Scotland. Model performance was then evaluated using  standard performance metrics for daily flows, and  novel performance metrics for peak flows considering parameter uncertainty.

Our results show that  lumped hydrological models were able to produce adequate simulations across most of Great Britain, with each model producing simulations exceeding 0.5 Nash Sutcliffe efficiency for at least 80% of catchments. All four models showed a similar spatial pattern of performance, producing better simulations in the wetter catchments to the west, and poor model performance in Scotland and southeast England. Poor model performance was often linked to the catchment water balance, with models unable to capture the catchment hydrology where the water balance did not close. Overall, performance was similar between model structures, but different models performed better for different catchment characteristics and metrics  for assessing daily or peak flows, leading to the ensemble of model structures outperforming any single structure thus demonstrating the value of using  multi-model structures across a large sample of different catchment behaviours.

This research  evaluates what conceptual lumped models can achieve as a performance benchmark, as well as providing interesting insights into where and why these simple models may fail. The large number of river catchments included in this study makes it an appropriate benchmark for any future developments of a national model of Great Britain.

**Commented [RI1]:** R2 Page 10 Line 3 "Title: the field is "large-sample hydrology", but here it should be "large sample""

**Commented [RI2]:** R2C4 Page 7 Line 15: "Comment 4: Relevance for the broad hydrological modelling community: A challenge here is to provide guidance for model selection, which is also relevant for modellers not using FUSE."

**Commented [RI3]:** R3C4 Page 12 Line 30 "Your statement in the abstract L23 that NSE scores of 0.72-0.78 were achieved for all catchments is misleading. How useful a measure is the "median maximum NSE for all the catchments"? It's pretty cryptic. There are catchments in E Scotland, and Anglian region that are showing pink/red for all 4 models, so NSE must be <0.5. Having got to page 9 I now see what you meant, but it isn't clearly stated. The sentences on P12 L16-17 are a better summary of the performances across catchments. Same issue on P16 L 32."

**1 Introduction**

Lumped and semi-distributed hydrological models, applied singularly or within nested sub-catchment networks, are used for a wide range of applications. These include water resources planning, flood/drought impact assessment, comparative analyses of catchment and model behaviour, regionalisation studies, simulations at ungauged locations, process based analyses, and
5 climate or land-use change impact studies (see for example Coxon et al., 2014; Formetta et al., 2017; Melsen et al., 2018; Parajka et al., 2007; Perrin et al., 2008; Poncelet et al., 2017; Rojas-Serna et al., 2016; Salavati et al., 2015; van Werkhoven et al., 2008). However, model skill varies between catchments due to differing catchment characteristics such as climate, land use and  topography. Evaluating  where models perform well/poorly and the reasons for these variations in model performance, can provide a benchmark of model performance
10 to help us better interpret modelling results across large samples of catchments (Newman et al., 2017) and lead to more targeted model improvements through synthesising those interpretations.

**1.1 Large sample hydrology**

  large-sample hydrological studies, also known as comparative hydrology,
15 test hydrological models on many catchments of varying characteristics (Gupta et al., 2014; Sivapalan, 2009; Wagener et al., 2010).  science Evaluating model performance across a
20 large sample of catchments can lead to improved understanding of hydrological processes and teach us a lot about hydrological models, for example,  the appropriateness of model structures for different types of catchment characteristics (i.e. Van Esse et al., 2013; Kollat et al. 2012), emergent properties and spatial patterns, key processes that we should be improving and identification of areas where models are unable to produce satisfactory results (e.g. Newman et al., 2015; Pechlivanidis and Arheimer, 2015). This can guide model selection, and also teach us about appropriate model parameter values for different
25 catchment characteristics, with the production of parameter libraries which can be used for parameter calibration in ungauged basins, and increase robustness of calibration in poorly gauged basins (Perrin et al., 2008; Rojas-Serna et al., 2016).
At the same time, regional-continental scale hydrological modelling studies are increasingly needed, to address large-scale challenges such as managing water supply, water scarcity and flood risk under climate change, and to inform large-scale policy decisions such as the European Union's Water Framework Directive (European Parliament, 2000).
30

**Commented [Rl4]:** R2: Page 10 Line 7 "P1L15: add "and support model selection""

**Commented [Rl5]:** R1 Page 3 Line 28 "authors discuss the benefits of national scale hydrological modelling. Another benefits could be the production of parameter libraries, which could be used for regional studies or model calibration on poorly gauged to ungauged basins or engineering studies. Authors can make references to papers on this subject (Perrin et al., 2008 ; Rojas-Serna et al., 2016 ; or some other works by Seibert)."

[revised manuscript text omitted]

Commented [RI11]: R2C10 Page 9 Line 10 "I find the introduction too long. It attempts to cover too much material, and hence ends up being too general and its different parts are not very well connected. I suggest that the authors focus on what is really necessary to introduce their study, transfer parts of the text to the rest of the paper (e.g. the methods), and delete the rest."

Commented [RI12]: R2 Page 10 Line 17 "P4L12-17: this belongs to Data and Methods"

Commented [RI13]: R2 Page 10 Line 19 "P4L20: I suggest removing "(i.e. the number of storage components)" as it an arbitrary measure of complexity."

Field Code Changed

Field Code Changed

Commented [RI14]: R1: Line 34 Page 4 "I would also make a reference to Perrin et al. 2001 here"

hydrological model ability across a large sample of British catchments whilst considering model structural and parameter uncertainty. This will be useful both as a benchmark of model performance against which other models can be evaluated and improved upon in Great Britain, and as a large-sample study which can provide general insights into the influence of catchment characteristics and selected model structure and parameterisation on model performance.

5 The specific research questions we investigate are:

1. How well do simple, lumped hydrological model structures perform across Great Britain, when assessed over annual and seasonal time scales via standard performance metrics?

2. Are there advantages in using an ensemble of model structures over any single model, and so are there any emergent patterns/characteristics in which a given structure and/or behavioural parameter set outperforms others?

10 3. What is the influence of certain catchment characteristics on model performance?

4. What is the predictive capability of  those identified as behavioural models for then predicting annual maximum flows when applied in a parameter uncertainty framework?

To address these questions, we have applied the four core conceptual hydrological models from the FUSE hydrological framework to  1013 British catchments, within an uncertainty analysis framework. Model performance and predictive

15 capability have been evaluated at each catchment, providing a national overview of hydrological modelling capability for simpler lumped conceptualisations over Great Britain.

**2 Data and Catchment Selection**

**2.1 Catchment Data**

This study was national in scope, using a large  data set of  1013  catchments distributed
20 across Great Britain (GB). The catchments cover all regions and include a wide variety of catchment characteristics including topography, geology and climate (see Table 1), and include both natural and human impacted catchments (see Figure 1).

On average, rainfall is highest in the north and west of GB, and lowest in the south and east, with GB totals varying from a minimum of 500mm to a maximum of 4496mm per year (see Figure 2). There is also
25 seasonal variation with the highest monthly rainfall totals generally occurring during the winter months and the lowest totals occurring in the summer months. This pattern is enhanced by seasonal variations in temperature with evaporation losses concentrated in the summer months from April – September. Besides climatic conditions, river flow patterns are also heavily influenced by groundwater contributions. Figure 1 shows the major aquifers in GB. In catchments overlying the Chalk outcrop in the South-East, flow is groundwater-dominated with a predominantly seasonal hydrograph that responds less quickly to
30 rainfall events. Land use and human modifications to river flows also significantly impact river flows, with river flows heavily modified in the South- and Midland regions of England due to high population densities (Figure 1). Most catchments

**Commented [RI15]:** R2: Page 6 Line 15 "Do the authors mean that their runs can be used as benchmark by future studies, as suggested on P12L12? Please clarify."

**Commented [RI16]:** R3: Page 16 Line 24 "P5 L14 - "these" should be "those""

**Commented [RI17]:** Tougher selection criteria – still more than 1000 catchments.

**Commented [RI18]:** R3: Page 16 Line 14 "3. P5 L22 - remove "Environment Agency", a catchment is a catchment, the EA don't own the catchments, even if they do own the gauges!"

**Commented [RI19]:** R1: Page 5 Line 17 "Table 1 is not cited within §2"

**Commented [RI20]:** R3: Page 16 Line 18 "4. Amend "Rainfall is highest in the West and North of GB and lowest in the East and South varying from a minimum of 500mm to a maximum of 4496mm per year (see Fig. 1)" to "On average, rainfall is highest in the north and west of GB, and lowest in the south and east, with GB totals varying from a minimum of 500mm to a maximum of 4496mm per year (see Fig. 1).""

**Commented [RI21]:** R3: Page 16 Line 25 "P6 L1 - remove the "of" after "South-East""

have very little or no snowfall in an average year, but there are some upland catchments in northern England and northeast Scotland where up to 15% of the annual precipitation falls as snow (Figure 2).

Catchments were selected from the National River Flow Archive (Centre for Ecology and Hydrology, 2016) based upon the quality and availability of rainfall, potential evapotranspiration (PET) and river discharge data over the period 1988-2008. The full NRFA dataset contains records for 1463 catchments across GB. Of these, 1013 had sufficient information (defined as more than 10 years of available discharge data during the model evaluation period of 1993-2008) available to include in this analysis.

**2.2 Observational Data**

Twenty-one years of daily rainfall and PET data covering the period 01/01/1988 to 31/12/2008 were used as hydrological model input. Rainfall timeseries were derived from The rainfall product used was the Centre for Ecology and Hydrology Gridded Estimates of Areal Rainfall, CEH-GEAR (Tanguy et al., 2014). This is a 1km$^2$ gridded product giving daily estimates of rainfall for Great Britain (Keller et al., 2015). It is based upon the national database of rain gauge observations collated by the UK Meteorological Office, with the natural neighbour interpolation methodology used to convert the point data to a gridded product (Keller et al., 2015). Catchment areal precipitation were then determined by averaging the values of all the grid squares that lied lay within the catchment boundaries to produce a daily rainfall time series for each of the 1128 1013 catchments.

The Climate Hydrology and Ecology research Support System Potential Evapotranspiration (CHESS-PE) dataset was used to estimate daily PET for each catchment. The CHESS-PE dataset is a 1km$^2$ gridded product for Great Britain, providing daily PET time-series (Robinson et al., 2015a). PET estimates were produced using the Penman-Monteith equation, calculated using meteorological variables from the CHESS-met dataset (Robinson et al., 2015b). Catchment areal daily precipitation and Daily PET time series were produced for each catchment by averaging values of all grid squares that lied lay within the catchment boundaries for each of the 1013 catchments.

Observed discharge data wereas used to evaluate model performance. Gauged daily flow data from the National River Flow Archive (NRFA) wasere used for all catchments where available (Centre for Ecology and Hydrology, 2016).

**3 Methodology**

**3.1 Hydrological Modelling**

The Framework for Understanding Structural Errors (FUSE) modelling framework was used to provide four alternative hydrological model structures. This framework was selected as it enables comparison between hydrological models with varying structural components (Clark et al., 2008) and the computational efficiency of these relatively simple hydrological models enabled modelling to be carried out across a large number of catchments within an uncertainty analysis framework. The framework allows the user to select different combinations of modelling decisions, starting with four parent models based on the structures of widely used hydrological models, and allowing the user to combine these decisions to create over 12000 different model structures.

**Commented [RI22]:** R1: Page 5 Line 12 "In §2, I would give an estimation of the proportion of watersheds where snowmelt processes are observable (solid precipitation >20% of total precipitation ?)"

**Commented [RI23]:** R2: Page 10 Line 27 "P6L5: please define "sufficient""

R3C7 : Page 13 Line 26 "P6 L6 – 2 years of data was your criteria for catchment selection, this doesn't seem sufficient to me"

**Commented [RI24]:** R3: Page 16 Line 26 "P6 L12 – they are the "UK Met Office" not the "UK Meteorological Office"."

**Commented [RI25]:** R3: Page 16 Line 27 "P6 L14 and L20 – replace "laid" with "lay""

**Commented [RI26]:** R3: Page 16 Line 28 "L6 L21 and elsewhere – "data" is plural, and should be followed by "were" instead of "was""

**Commented [RI27]:** R2C11 Page 9 Line 18 "Outlook: it might good to mention that, although this study focusses on four FUSE models, it is possible build additional FUSE model to transition progressively from one model to the next, and establish which modelling decisions contribute most to the differences in the simulations."

For this study, only the four parent models from the FUSE framework were selected due to the computational requirements of running the models across such as large number of catchments, and that the core models should provide the core differences of models compared to all the possible variants. These models are based on four widely used hydrological models; TOPMODEL (Beven and Kirkby, 1979), the Variable Infiltration Capacity (ARNO/VIC) model (Liang et al., 1994; Todini, 1996), the Precipitation-Runoff Modelling System (PRMS) (Leavesley et al., 1983) and the Sacramento model (Burnash et al., 1974). The models are all lumped, conceptual models of similar complexity and all run at a daily timestep within the FUSE framework. They all close the water balance, have a gamma routing function and include the same processes, for example none of the models have a snow routine or vegetation module. However, the structures of these models differ through the architecture of the upper and lower soil layers and parameterizations for simulation of evaporation, surface runoff, percolation from the upper to lower layer, interflow and baseflow (Clark et al., 2008), as shown in Fig. 2Figure 3 and Table 3Table 3. This leads us to believe that the model structures are dynamically different, as they are representing hydrological processes in different ways, yet as all are based on widely used hydrological models they are equally plausible and we have no a priori expectations that one model should outperform the others (Clark et al., 2008).

Parameter uncertainty can be evaluated through calibrating models within an uncertainty evaluation framework. There are many different uncertainty analysis procedures in hydrology such as the Parameter Solution (ParaSol) method (van Griensven and Meixner, 2006), the Integrated Bayesian Uncertainty Estimator (IBUNE) (Ajami et al., 2007), the Sequential Uncertainty Fitting algorithm (SUFI-2) (Abbaspour et al., 2007), and the Generalized Likelihood Uncertainty Estimation (GLUE) framework (Beven and Binley, 1992).

The models were run within a Monte-Carlo simulation framework. There are 23 adjustable parameters within the FUSE framework, as shown in Table 2. Each of these was assigned upper and lower bounds based upon feasible parameter ranges and behavioural ranges identified in previous research (Clark et al., 2008; Coxon et al., 2014). Monte-Carlo sampling was then used to generate 10,000 parameter sets within these given bounds. Therefore, for each of the 1128 1013 catchments, the four4 hydrological model structures were each run using the 10,000 possible parameter sets over the 21 year period 1988-2008, resulting in. This resulted in >45 40 million simulations being carried out.

**3.2 Evaluation of Model Performance**

The objective of this study was to evaluate the model's ability to reproduce observed catchment behaviour with a focus on assessing the strengths and weaknesses of each model in different catchments. Given the large number of catchments evaluated, it was not possible to evaluate model performance against multiple different a large range of objective functions with this paper, here we aim to benchmark behaviour to metrics that capture different aspects of model performance. Consequently, we chose to evaluate the overall performance of the hydrological models through the widely used Nash-Sutcliffe Efficiency Index (Nash and Sutcliffe, 1970), which is an easy to interpret measure of model performance that is often used in studies interested in high flows as it emphasizes fit to peaks. To further diagnose the reasons for model good/poor performance, the simulation

**Commented [RI28]:** R2: Page 10 Line 32 "P7L2: I suggest mentioning here that none of these four models includes a snow Routine"

R3C5: Page 13 Line 8 "Catchment characteristics and climate – do all FUSE models maintain the water balance? Can you comment on the existence of models that don't (e.g. GR4J), and how those may overcome such problems? What are the implications of maintaining vs not maintaining water balance in conceptual lumped models? Are the four models you've chosen actually quite similar to each other? I think you need to make more of this somehow."

**Commented [RI29]:** R2: Page 11 Line 1 "P7L4: please define "dynamically different" and what makes them "equally plausible""

**Commented [RI30]:** R3C2: Page 12 Line 19 "Section 3.2 – why NSE?"

with the highest efficiency value was then analysed further using the decomposed metrics of bias, error in the standard deviation and correlation.

however results are stored for a number of additional metrics not reported here.. All metrics were calculated for the period 1993-2008, with the first 5 simulation years being used as a model warm-up period.

5    The Nash-Sutcliffe efficiency index was calculated for each individual simulation using;

$$E = 1 - \frac{\sum(O_i - S_i)^2}{\sum(O_i - \overline{O})^2}$$    (1)1)

where $O_i$ refers to the observed discharge at each timestep, $S_i$ refers to the simulated discharge at each timestep and $\overline{O}$ $\overline{O}$ is the mean of the observed discharge values. This results in values of $E$ between 1 (perfect fit) and $-\infty$, where a value of zero means that the model simulation has the same skill as using the mean of the observed discharges.

10    To gain insights into model agility and time varying model performance during different times of the year, we also assess differences in seasonal performance by splitting the observed and simulated discharge into March-May (Spring), June-August (Summer), September-November (Autumn) and December-February (Winter). Seasonal Nash-Sutcliffe Efficiency values were then re-calculated for all the catchments, using only data extracted for that season. This allowed us to see if there were any seasonal patterns in model performance, for example during periods of higher or lower general flow conditions.

15    The Nash-Sutcliffe efficiency can be decomposed into three distinct components; the correlation, bias and a measure of the error in predicting the standard deviation of flows (Gupta et al., 2009). Understanding how the models perform for these different components can help us diagnose why models are producing good/poor simulations. We therefore calculated these simpler metrics, for the simulations of each model gaining the highest efficiency values. The relative bias was calculated using:

$$\triangle \mu = \frac{\mu_s - \mu_o}{\mu_o}$$    (2)

where $\mu_s$ and $\mu_o$ refer to the mean of the simulated and observed annual cycle. Using this equation, an unbiased model would score 0 (a perfect score), and a model that underestimated or overestimated the mean annual flow would score a negative or positive value respectively. A value of +/- 1 would indicate an overestimation/underestimation of flow by 100%.
The relative difference in standard deviation was calculated using:

25    $$\triangle \sigma = \frac{\sigma_s - \sigma_o}{\sigma_o}$$    (3)

where $\sigma_s$ and $\sigma_o$ represent the standard deviation of the simulated and observed mean annual cycle. Again, a value of zero indicates a perfect score with no error, and positive/negative values indicate an overestimation/underestimation of the amplitude of the mean annual cycle respectively.

**Commented [RI31]:** R3C3: Page 12 Line 24 "Section 3.2 – "results are stored for a number of additional metrics not reported here". Stored where? Why would I care about this if you haven't made them available to me? I suggest you summarise these additional metrics in supplementary information. This may also address the issue of only reporting on NSE here."

**Commented [RI32]:** R3C6: Page 13 Line 18" P8 L23 – only a 1 year warm up period? This is not sufficient for many GW dominated catchments in the SE."

**Commented [RI33]:** R1: Page 5 Line 3 "mistake with O (mean of observed discharge)"

**Commented [RI34]:** Additional metrics were suggested by all reviewers

[revised manuscript text omitted]

**Commented [RL46]:** R2C7: Page 8 Line 22 "Comment 7: Just like hydrological behaviour, model performance is not determined by a single catchment characteristic, but rather, by the interaction of multiple catchment characteristics. So, firstly, would it be possible to consider a wider range of catchment attributes? So far, the authors employ the BFI, annual rainfall, the wetness index and the runoff coefficient, but many more attributes could be used to describe each catchment (e.g., Beck et al., 2015). I encourage the authors to add other attributes, which they might have computed for other studies or retrieved from the UK hydrometric register, which they mention in Table 1, in order to describe the landscape in a more complete fashion (indicators of human interventions would also be useful, see below).
Comment 8: And secondly, I think it would be beneficial to better account for the interactions between these attributes. The authors combine several attributes in Figure 7 to explain model performance, which I find particularly interesting. Maybe that the analyses they will perform when revising this study will lead to more figures of this type, and enable a more systematic analysis of the interactions between these predictors (perhaps using regression trees, see Poncelet et al., 2017). This is critical to go from describing where models fail and to explaining why they fail."

**4.5 Benchmarking Predictive Capability for Annual Maximum Peak Flows**

Model predictive capability for simulating annual maximum (AMAX) flows from behavioural models defined from the NSE measure is shown in Figure 10 and Figure 11.  Figure 10 assesses the ability of models to produce AMAX discharge estimates which are as close as possible to observations. Here, a value of 0 means simulated AMAX discharge is equal to observed, up to 1 means simulated AMAX discharge is within the bounds of the observational uncertainties applied and larger values such as 2 indicate that simulated discharge is double the limit of observational uncertainties away from the observed discharge (negative values mean that the model simulations are lower than the observed). Median $E_{amax}$ values from Eq. (2) are around -2.4 to -3.2 across all four models, with PRMS producing slightly better predictions in general than the other models. This shows that the models are underestimating peak annual discharges across the majority of GB catchments even though behavioural model have been selected using NSE which favours models that perform well at higher flows.

Figure 11  shows the percentage overlap between the simulated 5th and 95th AMAX bounds and the observed AMAX uncertainty bounds. Here, the boxplot on the left shows the variation of results across all catchments and models for each year, whilst the boxplot on the right summarizes results across all catchments and years for each model. The median value across all catchments is 0.16, meaning that there is a 16% overlap between the observed and simulated AMAX bounds averaged across all 20 years.

There are large variations in model ability to simulate observed annual maximum flows between years, when looking at median predictions. For example, 1990 and 2008, which were wetter than average years across most of GB, model ability to represent annual maximum discharge is poor. However, in 1996, which was a particularly dry year following the 1995 drought (Marsh et al., 2007), the models do a much better job of representing the annual maximum discharge. This may be in part due to the model tendency to underestimate discharge as seen in Figure 10. However, variations between years are less apparent when looking at 25th and 75th percentiles in 8. This could suggest that there are some catchments where predictions are more consistent between years, or that the large climatic variation across GB may conceal some of the effects of inter-year differences.

**5 Discussion**

This study provides a useful benchmark of the performance and associated uncertainties of four commonly used lumped model structures across GB, for future model developments and model types to be compared against. The large number of catchments included  makes this assessment a fair benchmark for any future national modelling studies, as well as smaller scale modelling efforts. A full list of models scores can be found at https://doi.org/10.5523/bris.3ma509dlakcf720aw8x82aq4tm.

Commented [RI47]: R3C13: Page 15 Line 1 "13. P11 L 23 – "the top row of plots" – there is only one plot in Fig 8!"

Commented [RI48]: R3C14: Page 15 Line 6 "14. P12 L 7-8 "However, variations between years are less apparent when looking at 25th and 75th percentiles in Fig. 8." We can't distinguish variation between years from Fig 8?"

[revised manuscript text omitted]

**Commented [RI55]:** R1: Page 4 Line 19 "authors discuss about groundwater flows between catchments, with losses or gain of waters. This problem is not new and some conceptual modelisation could be found in the literature since one or two decades. In a natural context, authors could make a reference to Le Moine at al. (2007, 2008) papers about groundwater flows and water balance closure. The existence of such groundwater flows in permeable geological context (chalk, limestones and/or karstic systems, etc.) was one of the reasons of the development of a groundwater exchange function within the GR model family. The use of this function should be motivated by (hydrogeologic) evidences of such groundwater flows (in order to avoid "overfitting" of the water balance, i.e. fudge factor), but might be useful in catchments where water balance is difficult to close, such as the one influenced by chalk aquifers in southeast england."

R2C5: Page 13 Line 8 "Catchment characteristics and climate – do all FUSE models maintain the water balance? Can you comment on the existence of models that don't (e.g. GR4J), and how those may overcome such problems? What are the implications of maintaining vs not maintaining water balance in conceptual lumped models? Are the four models you've chosen actually quite similar to each other? I think you need to make more of this somehow."

[revised manuscript text omitted]

> **Commented [RI65]:** R1: Page 5 Line 12 "In §2, I would give an estimation of the proportion of watersheds where snowmelt processes are observable (solid precipitation >20% of total precipitation ?)"

[Figure]

**Figure 3**: FUSE wiring diagram, showing the model structure decisions. TOPMODEL and ARNO/VIC have 10 parameters, PRMS has 11 parameters and SACRAMENTO has 12 parameters. Adapted from Clark et al., (2008).

[Figure]

**Figure 443: GB maps of model performance for each structure. Each point is a gauge location which is coloured based upon the best Nash Sutcliffe score attained by the model for that catchment.**

[Figure]

**Figure 5. GB maps of model performance for each structure for 3 different metrics. Top row shows model relative bias or relative error in simulated mean runoff (%), middle row shows relative error in the standard deviation of runoff (%), and the third row shows correlation between observed and simulated streamflow. Each point is a gauge location which is coloured based upon the best score for that metric.**

**Commented [RI67]:** R1 Page 2 Line 18: "Authors decided to use the classical Nash-Sutcliffe efficiency (NSE) index to evaluate model performances (and select behavioural models, NSE > 0.5). NSE index is famous and widely used in Rainfall-Runoff modeling. Even if the perfect efficiency index do not exists, this index is also known to have some drawbacks (Schaefli and Gupta, 2007, among many references). Gupta et al. (2009) introduced the Kling-Gupta efficiency index that allows to explicitly account for bias (mean and variability) and correlation, in the evaluation of model performances. Given the ambition of this paper, I would recommand the authors to consider in their analyses the Kling-Gupta efficiency index, or at least to decompose their results in terms of correlation and mean bias."

R2: Page 6 Line 30 "Since the authors aim to better understand "where and why these simple models may fail" the choice of NSE is somewhat suprising, since NSE is a measure of overall performance, which provides limited insights into the reasons for high or low performance. Although an evaluation based on hydrological signatures would have enabled a more process-based diagnostic of model failures, I am not requiring this, since it would imply significant additional analyses. However, if the authors stick to NSE (or use KGE), I suggest that they use benchmarks (as suggested by Seibert et al., 2018) to account for the fact that high NSE/KGE values can be relatively easy to reach depending on the catchment and the season. I believe this would enable a more fair and enlightening assessment of the hydrological models across the catchments."

[revised manuscript text omitted]

**Supplementary Information 1 – Plots looking at the relationship between catchment characteristics and model performance**

In the main body of the paper, we looked at the relationship between the catchment wetness index and runoff coefficient and model performance. This was selected as these variables strongly were related to model performance and explained differences between catchments. Here, we give additional plots looking at the relationship between model performance and many different catchment characteristics. These characteristics were either taken from the hydrometric register or calculated from the model input data timeseries (Centre for Ecology and Hydrology, 2016; Marsh and Hannaford, 2008a; Robinson et al., 2015a).

Figure S1 summarises the overall performance of the four models, and was used to help identify overall trends in model structure performance across all catchments. Figures S2 – S5 are scatter plots looking at the relationship between model performance (assessed using NSE, bias, error in standard deviation and correlation respectively) and different catchment attributes. Figures S6 onwards are plots looking at interactions between different catchment attributes and model performance.

From Figures S2-S5 we can see that Small catchments (<200km$^2$) tend to have more variable NSE scores (both high and low), whilst large catchments (>3000km$^2$) always do fairly well. This is seen with all the decomposed metrics – with small catchments more likely to have errors in bias and standard deviation of flows, and having more variable correlation scores. A possible explanation for this could be that daily data is less able to capture the variation in these catchments.

Baseflow dominated catchments (BFI > 0.7) are more likely to gain really low NSE values (although some high BFI catchments can be simulated well). Interestingly, BFI seems to have a relationship with error in the standard deviation, with baseflow dominated catchments the only catchments where the best simulations tend to overpredict variation. This can be explained, as groundwater would dampen variation in flows.

Gauge elevation seems to cap overall model performance – with higher elevation gauges unable to achieve performance scores as high as low elevation gauges. Or this could potentially be a pattern because there are fewer high elevation gauges. This could also be a problem with using NSE, as lower scores are naturally given to catchments with more seasonal variation. We see that for these high elevation gauges, the best simulations always underpredict the flow standard deviation.

Surprisingly, urbanisation does not seem to decrease model performance.

From figures S6 onwards we can see that the worst NSE performaning catchments are grouped being small catchments less than 120km$^2$, with elevations below 125m, mid to high BFIs (>0.5), low annual rain less than 1000mm and annual runoff values which differ from other catchments with similar annual rainfall totals. Poor NSE ~0.5 is achieved for wetter catchments (annual rain > 1200mm), which have relatively low annual runoff generally less than 900mm. Many have flow attenuation from reservoirs and lakes, and for these catchments correlation is poor.

The largest problems with standard deviations seem to be small catchments, where standard deviation is generally underpredicted except for catchments with a high BFI where it is sometimes overpredicted. This can be explained as baseflow

dominated catchments may have less variation in flow so it is most likely that variation is overpredicted for these catchments. For small catchments relative errors are also most likely to be high.

Bias is very clearly linked to the climatic variables. There is a clear linear relationship with rainfall and runoff, and catchments deviating from this underpredict mean flows and have poorer correlations. Wetter catchments have reduced relative biases, which is likely to be due to biases being normalised by mean flow.

[Figure]

Figure S1. Cumulative distribution functions showing performance of the 4 model structures across all catchments, when assessed using NSE and decomposed metrics for the simulation with the highest NSE.

[Figure]

Figure S2: Relationship between NSE and a selection of 15 catchment descriptor variables. Column 1 is more general catchment attributes from the hydrometric register (Marsh and Hannaford, 2008). Column 2 gives hydroclimatic attributes calculated from our data, and proportion catchment is wet from the hydrometric register. Annual Loss is Rainfall-Runoff,

whilst Annual flood is the Median Annual maximum flood peak. Column 3 gives land-use and bedrock permeability descriptors, also from the UK hydrometric register.

[Figure]

Figure S3: Relationship between bias and a selection of 15 catchment descriptor variables, as in Figure S2.

[Figure]

Figure S4: Relationship between error in standard deviation and a selection of 15 catchment descriptor variables, as in Figure S2.

[Figure]

Figure S5: Relationship between correlation and a selection of 15 catchment descriptor variables, as in Figure S2.

[Figure]

Figure S6a: Relationship between general catchment characteristics, coloured by model ensemble NSE score for that catchment.

[Figure]

Figure S6b: Relationship between general catchment characteristics, coloured by model ensemble bias score for that catchment.

[Figure]

Figure S6c: Relationship between general catchment characteristics, coloured by model ensemble error in standard deviation for that catchment.

[Figure]

Figure S6d: Relationship between general catchment characteristics, coloured by model ensemble correlation for that catchment.

[Figure]

Figure 7a. Same as figure S6, but this time looking at hydroclimatic catchment descriptors.

[Figure]

Figure 7b. Same as figure S6, but this time looking at hydroclimatic catchment descriptors.

[Figure]

Figure 7c. Same as figure S6, but this time looking at hydroclimatic catchment descriptors.

[Figure]

Figure 7d. Same as figure S6, but this time looking at hydroclimatic catchment descriptors.

---

## Author Response (AR2)

**Benchmarking the predictive capability of hydrological models for river flow and flood peak predictions across a large sample of catchments in Great Britain**

**Response to minor revisions – August 2019**

We thank the editor and reviewer 2 for their further comments. These were regarding 1) changes to the reply letter clarifying where we have addressed specific comments, 2) validating the statement that 'the ensemble of model structures produced better results overall than any single model', and 3) improvements to the supplementary information document. Changes we have made to respond to these comments are summarised in the table below:

| Description of comment | Changes made following comment |
|---|---|
| Editor requested changes to the reply letter to clarify where we have addressed specific comments. | Clarifications have been made in some replies to reviewer 2 and reviewer 3. These can be seen highlighted in red on pages 8, 9, 13 and 17 of this document. |
| Editor requested a reference to actual results in response to Ref #3, comment 19. "we found that the ensemble of model structures produced better results overall than any single model" – can you validate that statement from your figures?" | We have moved figure S1, showing the distribution of model performance across all catchments, from the supplementary information into the main text and modified it to also include the model ensemble (see new Figure 4). This clearly shows the ensemble outperforms each model structure. Text in the results section has been modified to include this new figure, and the new figure numbers. |
| Ref #2 suggests improvements for supplementary information: "I recommend that the authors i) clarify what the variables correspond to (e.g. what is DPSBAR?), ii) reduce the number of Supplementary Figures (there are 13 of them currently), possibly by describing the most relevant results in the text without showing the associated Figures, iii) revise and condense the text (it is currently quite casual), iv) possibly mention that future research will rely on a wider range on catchment attributes and will explore model performance in a more systematic way." | i) Figure captions in supplementary information have been extended to define all variables.
 ii) We have removed 6 plots from the supplementary information.
 iii) Text in the supplementary information has been revised and condensed as can be seen from the tracked changes.
 iv) We decided to not include this in the supplementary information. |

| Editor advised carefully checking the revised version of the manuscript before the next submission. | A few small changes have been made to the manuscript. These include removal of a repetitive sentence in section 1.2, minor grammatical/spelling corrections in section 1.3 and rephrasing the sentence 'a considerable proportion of river discharges throughout the Anglian region are abstracted' to 'the South-East has some of the highest population densities in the UK and human influences can significantly impact flows in this region' in section 5.1. |
| --- | --- |

**General response to reviewers – June 2019**

We thank the reviewers for taking the time to read the manuscript, and for their thorough and insightful comments. Their suggestions have helped us to ensure our results are more useful to the modelling community.

The main comments from the reviewers were regarding (1) the use of Nash-Sutcliffe Efficiency to evaluate model performance, (2) making data more easily accessible, (3) synthesis of the introduction and discussion sections, and (4) plotting of additional catchment attributes and human influences on river flows.

In response to these reviewer comments, we have re-analysed the model output with consideration of additional performance metrics. We have supplied a complete set of outputs via a DOI. We have produced additional plots of factors affecting runoff across Great Britain, highlighting where streamflow is impacted by snowmelt and human influences in Figures 1 and 2. We have considered additional catchment attributes in supplementary information. The manuscript has also been revised, to synthesize the introduction and discussion, and to discuss the new metrics and factors affecting runoff plots in the methods and results sections.

Detailed responses to all reviewer comments are provided in bold below. We have also inserted the review comments as comments next to the relevant tracked changes in the manuscript.

Rosie Lane, June 2019

**Response to reviewer 1 (Thibault Mathevet)**

We thank Thibault Mathevet for taking the time to review our manuscript, and for his helpful comments. Our responses to each comment are outlined in bold below.

I carefully read the paper by Lane et al.. This paper appeared to be particularly clear, well written and easy to follow. Scope and objectives are stated clearly, the presentation of results is rather straightforward. As you probably know it, I appreciate this kind of study on a large sample of watersheds. I am very happy to know that such a large sample exists for GB. Studies on large sample give generality and robustness to the results. This paper gives insights on the general hydrology of GB and predictive capabilities of 4 simple rainfall-runoff models. I really appreciated §4 and §5, particularly analyses linked to the seasonality (fig 4), BFI (fig 5, 6), and water balance closure (fig 7). Thanks to this large sample of watersheds in GB with a variety of hydrologic/ hydrogeologic functioning (even in the same country), these results appear to be robusts, with a general interest. The link between BFI (main underground processes) and model structure agility is really interesting.

**Response: We thank Thibault Mathevet for taking the time to thoroughly review our manuscript, and for his positive comments.**

**Main comments:**

Evaluation of model performance and selection of model :
Authors decided to use the classical Nash-Sutcliffe efficiency (NSE) index to evaluate model performances (and select behavioural models, NSE > 0.5). NSE index is famous and widely used in Rainfall-Runoff modeling. Even if the perfect efficiency index do not exists, this index is also known to have some drawbacks (Schaefli and Gupta, 2007, among many references). Gupta et al. (2009) introduced the Kling-Gupta efficiency index that allows to explicitly account for bias (mean and variability) and correlation, in the evaluation of model performances. Given the ambition of this paper, I would recommend the authors to consider in their analyses the Kling-Gupta efficiency index, or at least to decompose their results in terms of correlation and mean bias.

**Response: The NSE index was chosen for this analysis as it is so widely used and easy to interpret. Given our focus on floods, it is also a good choice as it emphasizes the fit to peaks more than KGE which focuses on balancing the contribution of the bias and correlation. However, we agree that there are drawbacks to only using the NSE index and so following this comment, we have provided additional analysis looking at the correlation, variance and bias. This can be seen in Figures 5 and 9.**

Poor performances on floods :

Authors found that the different models had poor performances on floods, which is generally the case when classical modeling schemes are used to optimise or select parameter sets. I appreciate the simple way authors evaluate models on flood values, however I would add a figure to explain the two metrics. One of the main drawback of the NSE (and linear regression as well) is that the standard deviation of the simulated time-series is biased and underestimated, i.e. flood underestimated and drought overestimated. Among other arguments, this drawback partly explain why flood values are underestimated. I would add at least a comment on the fact that this statement is dependent on the behavioural model selection metrics in §5.4. If authors update their paper using KGE to select their behavioural models, they might revise (a bit) their findings on model performances for floods.

**Response: Thank you for pointing this out. In response to this comment, we have clarified the explanation of flood metrics. We selected NSE as it emphasizes the fit to peaks, whilst KGE is more general, but we acknowledge that it has drawbacks and no global performance measure is useful in all situations, especially when looking at extremes. We therefore decided to keep using NSE, but added a comment in the discussion on how behavioural model selection metrics influence estimation of flood values.**

Focus on droughts ? :

Given the ambition of this paper, I think that this paper would also benefit from a focus on droughts. Hence, analyses on droughts could be complementary to analyses on relative model performances (among the 4 tested structures), since droughts might also be driven by BFI in GB ? The link with groundwater flows could also be shown, if a focus on droughts is done. Authors could use the same metrics as for floods. It could be better to use the 10 days or 30 days annual minimal value, instead of the annual minimal value, which could be highly impacted and uncertain.

**Response: We agree that focusing on droughts could be an interesting question in itself, however we feel that it is out of scope for this paper. We are aiming to give a general overview of the capability of models, with a focus on high flows. Drought and very low flows is a more complex problem to address, and more likely to be influenced by human impacts in managed catchments. We therefore think adding this would be too much for one paper. We plan further research which better incorporates human influences on low flow river totals and thus will make such an assessment more fruitful.**

**Minor comments :**

In §1.1 : authors discuss the benefits of national scale hydrological modelling. Another benefits could be the production of parameter libraries, which could be used for regional studies or model calibration on poorly gauged to ungauged basins or engineering studies. Authors can make references to papers on this subject (Perrin et al., 2008 ; Rojas-Serna et al., 2016 ; or some other works by Seibert).

**Response: Thank you for this idea, we have referred to parameter libraries in the introduction, and have added tables of best parameter sets made available through a DOI.**

In §5.1 : authors did not use a snow accumulation and melt routine in their modeling framework. Very simple snow routine are available, in the spirit of the simple models proposed in FUSE. The CemaNeige routine could be a good candidate to improve model simulations on the few catchments where it's necessary. Depending on the proportion of snow impacted catchments, using a snow routine would improve model performances and the paper, as it could give answers to some hypotheses of the paper.

Valéry, A., Andréassian, V., Perrin, C., 2014. 'As simple as possible but not simpler': What is useful in a temperature-based snow-accounting routine? Part 1 – Comparison of six snow accounting routines on 380 catchments, Journal of Hydrology, 517(0): 1166-1175.

**Response: Thank you for this comment, but we do not think it would be feasible to run all simulations again with a snow routine. We originally decided not to use a snow routine as only a relatively few catchments were snow impacted. To check this, we have calculated snow fractions for all catchments, as the sum of the rainfall on days when daily mean temperature is less than 0 degrees Celsius divided by the total sum of the rainfall for the whole time period. This confirms that only a small proportion of catchments are snow impacted (13 catchments out of the 1127 have a snow fraction of more than 10%, and no catchments have a snow fraction of more than 17%). We have plotted these snow fractions in Figure 1, demonstrating that only a small proportion of catchments are impacted by snow. As the concept of the paper is focused on benchmarking the capability of these lumped models, and not model development, we feel that addition of a snow routine is out of scope.**

In §5.3 : authors discuss about groundwater flows between catchments, with losses or gain of waters. This problem is not new and some conceptual modelisation could be found in the literature since one or two decades. In a natural context, authors could make a reference to Le Moine at al. (2007, 2008) papers about groundwater flows and water balance closure. The existence of such groundwater flows in permeable geological context (chalk, limestones and/or karstic systems, etc.) was one of the reasons of the development of a groundwater exchange function within the GR model family. The use of this function should be motivated by (hydrogeologic) evidences of such groundwater flows (in order to avoid "overfitting" of the water balance, i.e. fudge factor), but might be useful in catchments where water balance is difficult to close, such as the one influenced by chalk aquifers in southeast england.

**Response: Thank you for highlighting these interesting and very relevant papers. We have added this into the discussion. However we have not yet done a comprehensive analyses of gaining and losing streams in the UK aquifer systems. This is indeed research that our group is currently conducting in more detail (separate PhD on improving ground water representation in models). Certainly from our preliminary analyses it is very difficult to attribute these losses and gains, and especially for lumped catchment model behaviours where the spatial partitioning needed might be too abstract to incorporate in the model outputs.**

**Last comments :**

P4, l22 : I would also make a reference to Perrin et al. 2001 here

**Response: We agree this is a relevant paper, it has been added.**

5 P7, l20 : mistake with O (mean of observed discharge)

**Response: Thank you for noticing this, it has been corrected.**

P10, l22 : values instead of vales

**Response: This has been corrected.**

P17, l8 : for catchments, repeated 2 times

**Response: The repetition has been removed.**

In §2, I would give an estimation of the proportion of watersheds where snowmelt processes are observable (solid precipitation
15 >20% of total precipitation ?)

**Response: We agree that this would be useful and have added a map of snow fractions to figure 2 which we refer to in section 2.**

Table 1 is not cited within §2

20 **Response: Thank you for spotting this, we have now added the citation: "The catchments cover all regions and include a wide variety of catchment characteristics including topography, geology and climate (see Table 1)."**

In §3.3, the +/- 13% concerning streamflow uncertainties for flood should be a bit more explained. To which probability range this uncertainty refers ? Is it one or two standard deviation (or something else) ?

25 **Response: The +/-13% represents the 95th percentile range of the discharge uncertainty bounds and was chosen as a representative discharge uncertainty for annual maximum flows from a national analysis of discharge uncertainties (Coxon et al, 2015). We have better clarified this, with the text now reading "This observed error value was selected following previous research on quantifying discharge uncertainty at 500 UK gauging stations for high flows, and represents the average 95$^{th}$ percentile range of the discharge uncertainty bounds for high flows (Coxon et al., 2015;**
30 **Mcmillan et al., 2012)."**

In Figure 2, I would put the number of free parameters to calibrate.

**Response: The following has been added to the figure caption "TOPMODEL and ARNO/VIC have 10 parameters, PRMS has 11 parameters and SACRAMENTO has 12 parameters. ".**

**Response to reviewer 2 (Anonymous)**

Comment 1: This study compares four structures from the framework FUSE in 1100 UK catchments. This is, in itself, a
significant achievement. The authors highlight which structures perform best in different regions (Results Section) and then
discuss more generally why models fail and which improvements would be necessary to improve performance (Discussions
Section). I think that, ultimately, the goal of such model intercomparison is to provide guidance on i) model selection (i.e., can
specific models/modules be recommended based on basin attributes?) and ii) model development (i.e., are there specific
process parameterisations that are currently missing, but are needed to improve the simulations?). In my view, the latter point
is addressed quite well (although I suggest restructuring the text to makes these results stand out more, and to go beyond FUSE
structures by discussing modelling decisions more generally) but the former point could be addressed in a more systematic
and comprehensive way. Overall, I consider that, after revisions, this paper has the potential to become a timely and welcome
addition to the literature.

**Response: We thank reviewer 2 for these helpful comments, and for taking the time to review our manuscript.**

**Major comments:**

Comment 2: Model intercomparison vs. benchmarking: Since the authors use the term "benchmarking" in the title and
throughout the manuscript, I encourage them to clarify in the introduction what differentiates model benchmarking from model
intercomparison. As the authors compare FUSE structures with each other, isn't their study rather a model intercomparison?
Do the authors mean that their runs can be used as benchmark by future studies, as suggested on P12L12? Please clarify.

**Response: We have included clarification of this in the introduction, including more explanation on how the
performance of simple hydrological models can be used as a benchmark and making it clear that our results can be
used as a benchmark. We used the term 'benchmark' to highlight that these results can be used as an indicator of the
ability of lumped models, which future studies may use when evaluating the performance of other models (that are
perhaps more complex or include additional processes). For example, our results would inform a modeller that gaining
an NSE of 0.7 in SE England is a good achievement, whereas gaining the same score in west Wales is not an achievement
as most models can easily gain higher NSE scores for these catchments. The use of simple models as benchmarks has
been advocated in previous studies, for example Seibert et al., (2018).**

**Seibert, J., Vis, M. J., Lewis, E., & Meerveld, H. J. (2018). Upper and lower benchmarks in hydrological modelling.
Hydrological Processes.**

Comment 3: Model evaluation using NSE: Since the authors aim to better understand "where and why these simple models may fail" the choice of NSE is somewhat surprising, since NSE is a measure of overall performance, which provides limited insights into the reasons for high or low performance. Although an evaluation based on hydrological signatures would have enabled a more process-based diagnostic of model failures, I am not requiring this, since it would imply significant additional analyses. However, if the authors stick to NSE (or use KGE), I suggest that they use benchmarks (as suggested by Seibert et al., 2018) to account for the fact that high NSE/KGE values can be relatively easy to reach depending on the catchment and the season. I believe this would enable a more fair and enlightening assessment of the hydrological models across the catchments.

**Response: We originally selected NSE as it is a widely used and easy to interpret measure of performance. However, we agree that in order to better understand model failures we need to consider additional measures of performance. Therefore, we have also presented correlation, variance and mean bias, as called for by the first reviewer, to support the seasonal analysis of model performance that we have already carried out. These additional results can be seen in Figs 5 and 9, and the associated text. As our focus is on reasons for model failures, we feel that these additional decomposed metrics will be more informative than the use of benchmarks.**

Comment 4: Relevance for the broad hydrological modelling community: A challenge here is to provide guidance for model selection, which is also relevant for modellers not using FUSE. Overall, the most interesting question is not really which FUSE model performs best, but why. I encourage the authors to discuss and highlight specific model elements that contribute to poor/good simulations, rather than focussing FUSE models themselves (e.g., TOPMODEL or PRMS). For instance, the fact that ARNO-VIC performs particularly well in high-BFI catchments is only an intermediary result, which is mostly relevant to FUSE users. The reasons why this is the case (e.g., last paragraph of Section 5.2), on the other hand, are relevant to a much wider group. I suggest a stronger emphasis on modelling decisions, as opposed to FUSE models, in particular in the most critical parts of the manuscripts (abstract and conclusions).

**Response: We agree that highlighting specific model elements that contribute to poor/good simulations would be of great use to the broad hydrological modelling community. Where possible, we have tried to outline modelling decisions that may cause differences in the results. However, it is difficult to distinguish which model elements are causing good/poor model performance, as the model structures differ in multiple aspects, and further analysis would be required to fully explore which modelling decisions are contributing to good/poor simulations. We have however added an extra table (Table 3) explaining which modelling decisions were applied for each FUSE model, highlighting the different model elements.**

Comment 5: Which process parameterisation are missing to capture the range of hydrological behaviours across the UK? The authors identify catchments in which the four model structures perform poorly, and reflect on characteristics of these catchments to which the poor performance can be attributed (e.g., chalk, snow, high human impacts). I suggest that the authors

dedicate a subsection in the Discussion Section to these findings, which are relevant for both model development and selection. Can they formulate hypotheses on why annual maximum flows are underestimated, which could be tested by future studies?

**Response: We have dedicated a section of the discussion to "Identifying missing process parameterisations" in response to this comment. As suggested by the first reviewer, the choice of NSE could result in underestimation of flood peaks, and we have therefore commented in the discussion on how our choice of metrics could be a factor leading to the underestimation of flood values.**

Comment 6: How critical is the selection of model structure? There are cases of great equifinality (i.e., high NSE for all structures, mostly for humid catchments). As mentioned above, a high NSE is not a guarantee that the model structure is adapted, but as long as this is recognised (and this could be clearer throughout the manuscript), I think it is fine for this study. But in other (more interesting) catchments, some model structures clearly outperform other structures, and there, model choice is critical. I think this should be stressed more prominently, since these are cases in which the inadequacy of the model structure cannot be overcome by parameter tuning. Given the general tendency of using the same model structure across very diverse environments (as discussed e.g. by Addor and Melsen, 2019), I think this is an important result, which could be underscored more. A related question is: which catchment characteristics explain these large NSE differences between model structures?

**Response: We explored the importance of model structure selection in figures 4, 7, 8, 10 and 11, with figure 7 looking at catchment characteristics which were related to differences between the model structures. However, we agree that the question of how critical the selection of model structure is for different catchments was not well addressed in the manuscript. Therefore, we have clarified the discussion of this on page 15 lines 19-32.**

This leads me to a set of comments related to the use of catchment attributes to explain model performance.

Comment 7: Just like hydrological behaviour, model performance is not determined by a single catchment characteristic, but rather, by the interaction of multiple catchment characteristics. So, firstly, would it be possible to consider a wider range of catchment attributes? So far, the authors employ the BFI, annual rainfall, the wetness index and the runoff coefficient, but many more attributes could be used to describe each catchment (e.g., Beck et al., 2015). I encourage the authors to add other attributes, which they might have computed for other studies or retrieved from the UK hydrometric register, which they mention in Table 1, in order to describe the landscape in a more complete fashion (indicators of human interventions would also be useful, see below).

Comment 8: And secondly, I think it would be beneficial to better account for the interactions between these attributes. The authors combine several attributes in Figure 7 to explain model performance, which I find particularly interesting. Maybe that the analyses they will perform when revising this study will lead to more figures of this type, and enable a more systematic analysis of the interactions between these predictors (perhaps using regression trees, see Poncelet et al., 2017). This is critical to go from describing where models fail and to explaining why they fail.

**Response: We selected the attributes of BFI, annual rainfall, wetness index and runoff coefficient as they were observed to have the largest impact on model performance. In response to reviewer comments, we have created additional plots of snow fraction, and factors affecting runoff on catchments across GB which are given in Figures 1 and 2. We do not want to add many more figures into the manuscript as we feel that this may detract from the main messages of the paper, but have added additional plots looking at interactions between attributes as supplementary information. Also we believe the current analyses are in keeping with the abstract nature of lumped modelling systems where a greater range of catchment attributes might only be loosely related to the structure and parameterisation of the model design. We aim to explore these issues with more spatially orientated modelling approaches in future publications.**

Comment 9: Anthropogenic activities are repeatedly mentioned to explain poor model performance (e.g., P12L29, P14L16, P15L3). This is indeed plausible, but if qualitative or maybe quantitative indicators of the extent of human interventions could be included, so that their impacts on streamflow and model performance could be demonstrated or maybe even quantified, it would strengthen the study.

**Response: We agree that this is required to strengthen comments made regarding reasons for model failures. We have information on factors affecting runoff for all catchments in the hydrometric register. However, this only gives an indicator of which factors may affect runoff, and not to what extent, and therefore we decided not to include it in the original manuscript. In response to reviewer comments, we have added plots of factors affecting runoff in Figure 1.**

**Minor comments:**

Comment 10: I find the introduction too long. It attempts to cover too much material, and hence ends up being too general and its different parts are not very well connected. I suggest that the authors focus on what is really necessary to introduce their study, transfer parts of the text to the rest of the paper (e.g. the methods), and delete the rest.

**Response: We agree. We have shortened the introduction, by combining and shortening sections on large sample and national hydrology, and condensing the introduction of modelling uncertainties by moving sentences to the methods section or deleting where appropriate.**

Comment 11: Outlook: it might good to mention that, although this study focusses on four FUSE models, it is possible build additional FUSE model to transition progressively from one model to the next, and establish which modelling decisions contribute most to the differences in the simulations.

**Response: The following has been added, "The framework allows the user to select different combinations of modelling decisions, starting with four parent models based on the structures of widely used hydrological models, and allowing the user to combine these decisions to create over 1200 different model structures".**

Comment 12: Data availability: "This study provides a useful benchmark of the performance and associated uncertainties of four commonly used lumped model structures across GB, for future model developments and model types to be compared against". I agree. But then, I think that instead of saying that "All model outputs from this study are available upon request from the lead author", the authors should make the runs available online, and provide the doi, before the paper is published. This is expected by AGU journals, and I think it is good practice in order to avoid data loss.

**Response: We completely agree with this, and have provided a DOI for the data.**

**Other suggested changes:**

Title: the field is "large-sample hydrology", but here it should be "large sample"

**Response: Thank you, this has been changed to "over 1000 catchments" as it is more informative than "large sample".**

P1L15: add "and support model selection"

**Response: This has been added.**

P2L13: such as

**Response: Thank you for spotting this, this sentence has now been re-phrased.**

P2L29: impacted by what?

**Response: We have clarified and re-written the sentence to say "These have great benefits, as applying a consistent methodology across a large area enables comparison between places and identification of areas that may be at most risk of future hydrological hazards. "**

P4L12-17: this belongs to Data and Methods

**Response: These sentences have been moved to the data and methods section.**

P4L20: I suggest removing "(i.e. the number of storage components)" as it an arbitrary measure of complexity.

**Response: This has been removed.**

P5L22: discharge

**Response: This is referring to all the catchment data – we refer to discharge specific data at a later point in the methods section.**

P6L5: please define "sufficient"

**Response: This was explained in the following sentence. We have re-arranged these sentences to make this clearer, now saying "Of these, 1013 had sufficient information (defined as more than 10 years of available discharge data during the model evaluation period) available to include in this analysis."**

P7L2: I suggest mentioning here that none of these four models includes a snow Routine

**Response: We have added this, "They all close the water balance, have a gamma routing function and include the same processes, for example none of the models have a snow routine or vegetation module."**

P7L4: please define "dynamically different" and what makes them "equally plausible"

**Response: By "dynamically different" we meant that the models all represent the landscape in a different way, and have quite different and distinct structures as shown in figure 3. By "equally plausible" we are referring to the fact that we have no reason to expect one structure to behave better than the others, as all model structures are equally complete in terms of processes and all based on widely applied model structures. We have clarified this in the text by saying, "this leads us to believe that the model structures are dynamically different, as they are representing hydrological processes in different ways, yet as all are based on widely used hydrological models they are equally plausible and we have no a priori expectations that one model should outperform the others ".**

P8L13: please be more explicit about how this 13

**Response: This has been further explained with the text now reading, "This observed error value was selected following previous research on quantifying discharge uncertainty at 500 UK gauging stations for high flows, and represents the average 95$^{th}$ percentile range of the discharge uncertainty bounds for high flows."**

P9L21: saying "snowmelt module" implies that accumulation is simulated but melt is not, use "snow module" instead.

**Response: We agree, and have changed "snowmelt module" to "snow module."**

**Response to reviewer 3 (Anonymous)**

This paper provides a detailed investigation into the performance of four lumped conceptual models over large number of catchments in the UK. It demonstrates some very interesting findings, such as the fact that all four models have very similar performance on a catchment-by-catchment basis, and that only one of the models is deemed suitable for catchments with very high BFI. This paper is generally well written, set out and easy to follow, and the graphics provided assist the reader well in the interpretation of the results, I particularly like Figures 5 and 7. The discussion section should be synthesised as it feels repetitive of the results section. Overall, I feel that the motivations of the research, and the implications of the results are not very well reasoned. The authors need to think a bit more carefully about how others may make use of these results, and in particular, should publish the model performance scores as supplementary information (see my comments below).

**Response: We would like to thank the reviewer for taking the time to read the paper in depth, and for their constructive comments.**

Comment 1. You've "benchmarked" performance, but you haven't provided these benchmarks. If I were to now go and simulate a UK catchment, I still cannot easily compare my results with yours to see if I have a better model. For you to have achieved your aims, I would expect a supplementary table of the best scores the models achieved in each catchment, and the parameter values that produced them.

**Response: We completely agree with this, and the results can now be accessed through a DOI.**

Comment 2. Section 3.2 – why NSE?

**Response: We originally selected NSE as it is a widely used and easy to interpret measure of performance. However, as noted by the other reviewers, in order to better understand model failures we will consider additional metrics. Therefore, we have also presented correlation and mean bias (see figures 5,9 and the associated text).**

Comment 3. Section 3.2 – "results are stored for a number of additional metrics not reported here". Stored where? Why would I care about this if you haven't made them available to me? I suggest you summarise these additional metrics in supplementary information. This may also address the issue of only reporting on NSE here.

**Response: We have removed this sentence from the methods and included additional metrics in the results which will also be provided thorough the DOI.**

Comment 4. Your statement in the abstract L23 that NSE scores of 0.72-0.78 were achieved for all catchments is misleading. How useful a measure is the "median maximum NSE for all the catchments"? It's pretty cryptic. There are catchments in E Scotland, and Anglian region that are showing pink/red for all 4 models, so NSE must be <0.5. Having got to page 9 I now see

what you meant, but it isn't clearly stated. The sentences on P12 L16-17 are a better summary of the performances across catchments. Same issue on P16 L 32.

**Response: Thank you for pointing out that this is not clear. We have replaced the statement in abstract L23 with "Our results show that simple, lumped hydrological models were able to produce adequate simulations across most of Great Britain, with each model producing simulations exceeding 0.5 Nash Sutcliffe efficiency over at least 80% of catchments."**

Comment 5. Catchment characteristics and climate – do all FUSE models maintain the water balance? Can you comment on the existence of models that don't (e.g. GR4J), and how those may overcome such problems? What are the implications of maintaining vs not maintaining water balance in conceptual lumped models? Are the four models you've chosen actually quite similar to each other? I think you need to make more of this somehow.

**Response: Yes, all the FUSE models used in this study maintain the water balance, and we have clarified this in the methods section. To address these questions we have added a paragraph in the discussion on how models that do not maintain the water balance have been used to improve modelling in groundwater dominated regions. In response to reviewer 1, this includes discussion of papers by Le Moine at al. (2007, 2008) about groundwater flows and water balance closure.**

Comment 6. P8 L23 – only a 1 year warm up period? This is not sufficient for many GW dominated catchments in the SE.

**Response: Thank you for this advice. We initially selected 1 year, as it is often considered sufficient for simple, lumped models such as the FUSE models. However, following this comment we carried out additional analysis of the simulated flows and found that whilst 1 year is a long enough warmup period for many catchments, it did not appear sufficient for some of the catchments in the SE as suggested. We will have increased the warmup period to 5 years, re-analysed the data and re-made all the figures to reflect this.**

Comment 7. P6 L6 – 2 years of data was your criteria for catchment selection, this doesn't seem sufficient to me

**Response: We originally aimed to keep as many catchments as possible for the analysis. However, you are correct that 2 years of data is not long for model evaluation. We have now added a tougher criterion for catchment selection, of more than 10 years of available discharge data during the model evaluation period. The figures have been re-made to reflect this.**

Comment 8. Reading through your discussion seems very repetitive of the results chapter. Can these be better synthesised, to reduce the discussion section?

16. Your discussion is longer than the rest of the paper put together!

**Response: We have reduced the length of the discussion section, and re-structured the old sections 5.1-5.3 to reduce repetition.**

Comment 9. P2 L32 "a national scale model" – you're talking about applying a catchment model nationally. Can this be classified a national scale model?

**Response: In this section we were aiming to discuss the importance of national scale modelling more generally, suggesting that our work could be informative for evaluation of a national scale model. We were not saying that our application of a catchment model across GB was a national scale model.**

Comment 10. P3 L16 - "Secondly, evaluating more complex hydrological models relative to benchmark performance of simple models ensures that the relative difficulty of simulating different catchments is implicitly considered (Seibert et al., 2018)." I don't think I understand what you're saying here.

**Response: This has been re-phrased and further explained to make the meaning clearer. It now reads "Secondly, lumped hydrological models provide a good benchmark for evaluating more complex models, as they give an indication of what it is possible to achieve for a specific catchment and the available data (Seibert et al., 2018). This can help us identify whether a model is performing well in a catchment relative to how it should be expected to perform for the particulars of that catchment. For example, if a modeller gains an efficiency score of 0.7 for their model in a specific catchment, it is subjective whether this is a good or poor performance. However, if lumped, conceptual models tend to have efficiency scores of around 0.9 for that catchment then the modeller knows that their model is performing poorly relative to what is possible."**

Comment 11. P10 L22-24 – "For very low values of the ARNO-VIC 'b' exponent (AXV_BEXP) as seen for high BFI vales in Fig. 6 for behavioural model distributions means that only at very high, near full upper storage levels is any larger extent of saturated areas predicted" – I don't follow this sentence either.

**Response: This has been re-phrased.**

Comment 12. P8 L3 - Can you explain conditional probabilities in more detail?

**Response: We have extended this paragraph, now saying "Conditional probabilities were assigned to each behavioural parameter set based on their behavioural Efficiency score, and these were normalised to sum to 1. This meant that the simulations which scored the highest efficiency value had larger conditional probabilities, and simulations which had efficiency values just above 0.5 would have very low conditional probabilities. For each daily timestep, a 5$^{th}$, 50$^{th}$ and 95$^{th}$ simulated discharge bound was produced from these conditional probabilities, for each catchment and model structure individually as described in Beven and Freer (2001). This meant that simulations with a higher efficiency score were given a higher weighting when producing the discharge bounds." Simply the behavioural weights**

**(probabilities) assigned to each model are conditional on the choices made in the modelling exercise, here dependent on the sample design, the choice of parameter ranges, the model performance metric, and hence conditional.**

Comment 13. P11 L 23 – "the top row of plots" – there is only one plot in Fig 8!

**Response: Thank you for noticing that! We had originally displayed figures 8 and 9 as a single plot. This has been corrected.**

Comment 14. P12 L 7-8 "However, variations between years are less apparent when looking at 25th and 75th percentiles in Fig. 8." We can't distinguish variation between years from Fig 8?

**Response: Again, thank you for noticing this, it has been corrected to point to the right figure.**

Comment 15. Please provide more sensible y axis labels for fig 8 and 9, e.g. "AMAX discharge score", and "AMAX percentage overlap" respectively. Multiply Fig 9 y axis by 100 to make it an actual percentage value, as you have referred to it as such in the text.

**Response: We agree with this comment and have changed the figure.**

Comment 17. P13 L 3 – you've made no reference to anthropogenic influences in Scotland. This statements seems a bit throwaway.

**Response: We have removed this sentence.**

Comment 18. P13 L9 – it is not just the Thames basin that is affected by abstractions! A lot of Anglian region is VERY heavily influenced.

**Response: we have changed this sentence to "a considerable proportion of river discharges throughout the Anglian region are abstracted."**

Comment 19. P13 L12 "we found that the ensemble of model structures produced better results overall than any single model" – can you validate that statement from your figures?

**Response: This can not be directly validated in a specific figure, but it can be seen across the figures, especially looking at Figure 7, where we see that no single model produces good results for all catchments.**

Comment 20. P13 L15 – "The ensemble of model structures was able to take advantage of this" - this seems to be a contradictory argument to the previous statement that the models all have similar performance to each other on a catchment by catchment basis. I think you need to tease these two arguments out better somehow. E.g. in some situations the choice of a

different model can yield better results (e.g. high baseflow), but in other situations, none of the models can do well (e.g. abstractions). What are the implications of this?

**Response: We have clarified these arguments in the discussion, see page 15, lines 19-33 in the revised manuscript.**

Comment 21. P17 L 11-14 "We also evaluated model predictive capability for high flows, as good model performance in replicating the hydrograph, assessed using Nash-Sutcliffe efficiency, does not necessarily mean models are performing well for other hydrological signatures. We found that the FUSE models tended to underestimate peak flows, and there were variations in model ability between years with models performing particularly poorly for extremely wet years." – so what? What are the potential implications?

**Response: We have added discussion about the implications for flood modelling and forecasting, see section 5.3, page 16 lines 15-16 in the revised manuscript.**

**Typos and grammar:**

1. P2 L27 – CAMELS and MOPEX datasets (what are they datasets of?)

**Response: This sentence has been clarified - "the CAMELS or MOPEX hydrometeorological and catchment attribute datasets."**

3. P5 L22 - remove "Environment Agency", a catchment is a catchment, the EA don't own the catchments, even if they do own the gauges!

**Response: "Environment Agency" has been removed.**

4. Amend "Rainfall is highest in the West and North of GB and lowest in the East and South varying from a minimum of 500mm to a maximum of 4496mm per year (see Fig. 1)" to "On average, rainfall is highest in the north and west of GB, and lowest in the south and east, with GB totals varying from a minimum of 500mm to a maximum of 4496mm per year (see Fig. 1).

**Response: Thank you for the suggestion, this sentence has been amended.**

2. P5 L14 - "these" should be "those"

5. P6 L1 - remove the "of" after "South-East"

6. P6 L12 – they are the "UK Met Office" not the "UK Meteorological Office".

7. P6 L14 and L20 – replace "laid" with "lay"

8. L6 L21 and elsewhere – "data" is plural, and should be followed by "were" instead of "was"

9. P8 L15 – "observational uncertainty certainty bounds" huh?? Can you not just remove the word certainty here?

10. P9 L15 – you haven't introduced the abbreviation "SAC"

11. P10 L5 – I'd call that northeast Scotland, not central Scotland

12. P11 L29 – "behavioural model" should be "behavioural models"

13. P13 L7 – do you mean model "structures"?

14. P16 L17 – "we also shown how"

15. P16 L19 – refer to Fig 6

**Response: Thank you for spotting these typos and grammatical errors, we have corrected these in the manuscript.**

16. P16/17 – "The performance of the four models was similar, and all models showed similar spatial patterns of performance, and there was no single model that outperformed the others across all catchment characteristics and for both daily flows and peak flows." – and, and, and

**Response: This sentence has been improved to "The performance of the four models was similar, with all models showing similar spatial patterns of performance, and no single model outperforming the others across all catchment characteristics for both daily flows and peak flows."**

17. P17 L8 – "we found models performed poorly for catchments for catchments with unaccounted losses"

**Response: we have removed the repetition.**

HESS REVIEW CHECKLIST

1. Does the paper address relevant scientific questions within the scope of HESS? Yes

2. Does the paper present novel concepts, ideas, tools, or data? Yes

3. Are substantial conclusions reached? Nearly, the wider implications, and utility of the research need to be better considered

4. Are the scientific methods and assumptions valid and clearly outlined? Yes

5. Are the results sufficient to support the interpretations and conclusions? Yes

6. Is the description of experiments and calculations sufficiently complete and precise to allow their reproduction by fellow scientists (traceability of results)? Yes

7. Do the authors give proper credit to related work and clearly indicate their own new/original contribution? Yes

8. Does the title clearly reflect the contents of the paper? Yes

9. Does the abstract provide a concise and complete summary? Yes

10. Is the overall presentation well-structured and clear? Yes

11. Is the language fluent and precise? Yes

12. Are mathematical formulae, symbols, abbreviations, and units correctly defined and used? Yes

13. Should any parts of the paper (text, formulae, figures, tables) be clarified, reduced,

combined, or eliminated? Yes, the discussion should be reduced

14. Are the number and quality of references appropriate? Yes

15. Is the amount and quality of supplementary material appropriate? No

5 **Response: Thank you for this largely positive summary checklist. We have addressed points 3 and 13 through our changes to the discussion section, and point 15 by making our output data available through a DOI.**

**Benchmarking the predictive capability of hydrological models for river flow and flood peak predictions across over 1000 catchments in Great Britain**

Rosanna A. Lane[1], Gemma Coxon[1], Jim E. Freer[1,3], Thorsten Wagener[2,3], Penny J. Johnes[1,3], John P. Bloomfield[4], Sheila Greene[5], Christopher J. A. Macleod[6], Sim M. Reaney[7]

[1]School of Geographical Sciences, University of Bristol, Bristol, BS8 2NQ, United Kingdom
[2]Faculty of Engineering, University of Bristol, Bristol, BS8 2NQ, United Kingdom
[3]Cabot Institute, University of Bristol, Bristol, BS8 2NQ, United Kingdom
[4]British Geological Survey, Maclean Building, Wallingford, OX10 8BB, United Kingdom
[5]Trinity College Dublin, Dublin, Ireland
[6]The James Hutton Institute, Craigiebuckler, Aberdeen, AB15 8QH, United Kingdom
[7]Department of Geography, Durham University, Durham, DH1 3LE, United Kingdom

*Correspondence to*: Rosanna A. Lane (R.A.Lane@bristol.ac.uk)

**Abstract.** Benchmarking model performance across large samples of catchments is useful to guide model selection and future model development. Given uncertainties in the observational data we use to drive and evaluate hydrological models, and uncertainties in the structure and parameterisation of models we use to produce hydrological simulations and predictions, it is essential that model evaluation is undertaken within an uncertainty analysis framework. Here, we benchmark the capability of several lumped hydrological models across Great Britain, by focusing on daily flow and peak flow simulation. Four hydrological model structures from the Framework for Understanding Structural Errors (FUSE) were applied to over 1000 catchments in England, Wales and Scotland. Model performance was then evaluated using standard performance metrics for daily flows, and novel performance metrics for peak flows considering parameter uncertainty.

Our results show that lumped hydrological models were able to produce adequate simulations across most of Great Britain, with each model producing simulations exceeding 0.5 Nash Sutcliffe efficiency for at least 80% of catchments. All four models showed a similar spatial pattern of performance, producing better simulations in the wetter catchments to the west, and poor model performance in Scotland and southeast England. Poor model performance was often linked to the catchment water balance, with models unable to capture the catchment hydrology where the water balance did not close. Overall, performance was similar between model structures, but different models performed better for different catchment characteristics and metrics, as well as for assessing daily or peak flows, leading to the ensemble of model structures outperforming any single structure thus demonstrating the value of using multi-model structures across a large sample of different catchment behaviours. This research evaluates what conceptual lumped models can achieve as a performance benchmark, as well as providing interesting insights into where and why these simple models may fail. The large number of river catchments included in this study makes it an appropriate benchmark for any future developments of a national model of Great Britain.

**Commented [RI1]:** R2C4 Page 7 Line 15: "Comment 4: Relevance for the broad hydrological modelling community: A challenge here is to provide guidance for model selection, which is also relevant for modellers not using FUSE."

**Commented [RI2]:** R3C4 Page 12 Line 30 "Your statement in the abstract L23 that NSE scores of 0.72-0.78 were achieved for all catchments is misleading. How useful a measure is the "median maximum NSE for all the catchments"? It's pretty cryptic. There are catchments in E Scotland, and Anglian region that are showing pink/red for all 4 models, so NSE must be <0.5. Having got to page 9 I now see what you meant, but it isn't clearly stated. The sentences on P12 L16-17 are a better summary of the performances across catchments. Same issue on P16 L 32."

**1 Introduction**

Lumped and semi-distributed hydrological models, applied singularly or within nested sub-catchment networks, are used for a wide range of applications. These include water resources planning, flood/drought impact assessment, comparative analyses of catchment and model behaviour, regionalisation studies, simulations at ungauged locations, process based analyses, and
5 climate or land-use change impact studies (see for example Coxon et al., 2014; Formetta et al., 2017; Melsen et al., 2018; Parajka et al., 2007; Perrin et al., 2008; Poncelet et al., 2017; Rojas-Serna et al., 2016; Salavati et al., 2015; van Werkhoven et al., 2008). However, model skill varies between catchments due to differing catchment characteristics such as climate, land use and topography. Evaluating where models perform well/poorly and the reasons for these variations in model performance, can provide a benchmark of model performance to help us better interpret modelling results across large samples of catchments
10 (Newman et al., 2017) and lead to more targeted model improvements through synthesising those interpretations.

**1.1 Large sample hydrology**

Large-sample hydrological studies, also known as comparative hydrology, test hydrological models on many catchments of varying characteristics (Gupta et al., 2014; Sivapalan, 2009; Wagener et al., 2010). Evaluating model performance across a large sample of catchments can lead to improved understanding of hydrological processes and teach us a lot about hydrological
15 models, for example, the appropriateness of model structures for different types of catchment characteristics (i.e. Van Esse et al., 2013; Kollat et al. 2012), emergent properties and spatial patterns, key processes that we should be improving and identification of areas where models are unable to produce satisfactory results (e.g. Newman et al., 2015; Pechlivanidis and Arheimer, 2015). This can guide model selection, and also teach us about appropriate model parameter values for different catchment characteristics, with the production of parameter libraries which can be used for parameter calibration in ungauged
20 basins, and increase robustness of calibration in poorly gauged basins (Perrin et al., 2008; Rojas-Serna et al., 2016).

At the same time, regional-continental scale hydrological modelling studies are increasingly needed, to address large-scale challenges such as managing water supply, water scarcity and flood risk under climate change, and to inform large-scale policy decisions such as the European Union's Water Framework Directive (European Parliament, 2000). National-scale hydrological modelling studies using a consistent methodology across large areas are increasingly applied (Coxon et al., 2018; Van Esse et
25 al., 2013b; Højberg et al., 2013a, 2013b; McMillan et al., 2016; Veijalainen et al., 2010; Velazquez et al., 2010) facilitated by increasing computing power and the availability of open source large datasets such as the CAMELS or MOPEX hydrometeorological and catchment attribute datasets in the USA (Addor et al., 2017; Duan et al., 2006). These have great benefits, as applying a consistent methodology across a large area enables comparison between places and identification of areas that may be at most risk of future hydrological hazards. However, the range of catchment characteristics and hydrological
30 processes across national scales pose a great challenge to the implementation and evaluation of a national-scale model (Lee et al., 2006), and we therefore need large-scale evaluations of model capability to identify which processes are important and which model structure(s) are most appropriate.

**Commented [RI3]:** R2: Page 10 Line 7 "P1L15: add "and support model selection""

**Commented [RI4]:** R1 Page 3 Line 28 "authors discuss the benefits of national scale hydrological modelling. Another benefits could be the production of parameter libraries, which could be used for regional studies or model calibration on poorly gauged to ungauged basins or engineering studies. Authors can make references to papers on this subject (Perrin et al., 2008 ; Rojas-Serna et al., 2016 ; or some other works by Seibert)."

**Commented [RI5]:** R3: Page 16 Line 10 "1. P2 L27 – CAMELS and MOPEX datasets (what are they datasets of?)"

**Commented [RI6]:** R2: Page 10 Line 12 "P2L29: impacted by what?"

**Commented [RI7]:** R2C10 Page 9 Line 10 "I find the introduction too long. It attempts to cover too much material, and hence ends up being too general and its different parts are not very well connected. I suggest that the authors focus on what is really necessary to introduce their study, transfer parts of the text to the rest of the paper (e.g. the methods), and delete the rest."

**1.2 Benchmarking hydrological models**

Model skill varies between places, and it is therefore important for a modeller to understand the relative model skill for their study region, and how that relates to their core objectives. A single model structure will vary in its ability to produce good flow time-series across different environments and time-periods (McMillan et al., 2016), expressed sometimes as model agility (Newman et al., 2017).  One way to evaluate this relative model skill is by comparing the model performance to a benchmark, which is an indicator of what it is possible to achieve in a catchment given the data available (Seibert, 2001). This helps a modeller make a more objective decision on whether their model is performing well. Examples of benchmarks that models can be evaluated against include climatology, mean observed discharge, or the performance of a simple, lumped hydrological model for the same conditions (Pappenberger et al., 2015; Schaefli and Gupta, 2007; Seibert, 2001; Seibert et al., 2018).

The creation of a national benchmark series of performance of simple, lumped models can therefore be useful for a variety of reasons. Firstly, a benchmark series of lumped model performance is a useful baseline upon which more complex or highly distributed modelling attempts can be evaluated (Newman et al., 2015). This would ensure that future model developments are improving upon our current capability therefore justifying additional model complexity. Secondly, lumped hydrological models provide a good benchmark for evaluating more complex models, as they give an indication of what it is possible to achieve for a specific catchment and the available data (Seibert et al., 2018). This can help us identify whether a model is performing well in a catchment relative to how it should be expected to perform for the particulars of that catchment. For example, if a modeller, using more complex modelling approaches, gains an efficiency score of 0.7 for their model in a specific catchment, there is some subjectivity whether this is a good or poor performance depending on the modelling objective. However, if lumped, conceptual models already applied at the same catchment tend to have efficiency scores of around 0.9 for that catchment then the modeller knows that their model is performing poorly relative to what is possible. Thirdly, national benchmarks are useful for users of models as they can highlight areas where models have more or less skill, and where model results should be treated with caution.

**1.3 Assessing Uncertainty**

Hydrological model output is always uncertain, due to uncertainties in the observational data used to drive and evaluate the models, boundary conditions, uncertainties in selection of model parameters and in the choice of a model structure (Beven and Freer, 2001). There is a large and rapidly growing body of literature on uncertainty estimation in hydrological modelling, with many techniques emerging to assess the impact of different sources of uncertainty on model output, as summarised in Beven (2009). Despite this, uncertainty estimation is not yet routine practice in comparative or large-sample hydrology and few nationwide hydrological modelling studies have included uncertainty estimation, tending to look more at regionalization of

**Commented [RI8]:** R2: Page 6 Line 15 "Model intercomparison vs. benchmarking: Since the authors use the term "benchmarking" in the title and throughout the manuscript, I encourage them to clarify in the introduction what differentiates model benchmarking from model intercomparison. As the authors compare FUSE structures with each other, isn't their study rather a model intercomparison? Do the authors mean that their runs can be used as benchmark by future studies, as suggested on P12L12? Please clarify." – I have clarified why benchmarks may be used here.

**Commented [RI9]:** R3C10 Page 14 Line 10 "10. P3 L16 - "Secondly, evaluating more complex hydrological models relative to benchmark performance of simple models ensures that the relative difficulty of simulating different catchments is implicitly considered (Seibert et al., 2018)." I don't think I understand what you're saying here."

parameters, multi-objective calibration techniques, or the use of flow signatures in model evaluation (i.e. Donnelly et al., 2016; Kollat et al., 2012; Oudin et al., 2008; Parajka et al., 2007b).

Parameter uncertainty is often evaluated through calibrating models within an uncertainty evaluation framework (e.g. GLUE, (Beven and Binley, 1992) or ParaSol (van Griensven and Meixner, 2006)). Whilst many studies have explored parameter uncertainty, it is less common to evaluate the additional impact of model structural uncertainty on hydrological model output (Butts et al., 2004). Model structures can differ in their choice of processes to include, process parameterisations, model spatial and temporal resolution and model complexity. Studies attempting to address model structural uncertainty often apply multiple hydrological model structures and compare the differences in output (Ambroise et al., 1996; Perrin et al., 2001; Vansteenkiste et al., 2014; Velázquez et al., 2013), and in climate impact studies (i.e. Bosshard et al., 2013; Karlsson et al., 2016; Samuel et al., 2012). These studies have found that the choice of hydrological model structure can strongly affect the model output, and therefore hydrological model structural uncertainty is an important component of the overall uncertainty in hydrological modelling and cannot be ignored.

Flexible model frameworks is a useful tool for exploring the impact of model structural uncertainty in a controlled way, and for identifying the different aspects of a model structure which are most influential to the model output. These flexible modelling frameworks allow a modeller to build many different model structures using combinations of generic model components (Fenicia et al., 2011). For example, the Modular Modelling System (MMS) of Leavesley et al., (1996) allows the modeller to combine different sub-models and the Framework for Understanding Structural Errors (FUSE), developed by Clark et al., (2008),  combines process representations from four commonly used hydrological models to create over 1000 unique model structures.

**1.4 Study Scope and Objectives**

The main objective of this study is to comprehensively benchmark performance of an ensemble of lumped hydrological model structures across Great Britain, focusing on daily flow and peak flow simulation. This is the first evaluation of hydrological model ability across a large sample of British catchments whilst considering model structural and parameter uncertainty. This will be useful both as a benchmark of model performance against which other models can be evaluated and improved upon in Great Britain, and as a large-sample study which can provide general insights into the influence of catchment characteristics and selected model structure and parameterisation on model performance.

The specific research questions we investigate are:
1. How well do simple, lumped hydrological model structures perform across Great Britain, when assessed over annual and seasonal time scales via standard performance metrics?
2. Are there advantages in using an ensemble of model structures over any single model, and so are there any emergent patterns/characteristics in which a given structure and/or behavioural parameter set outperforms others?
3. What is the influence of certain catchment characteristics on model performance?

**Commented [Rl10]:** R2 Page 10 Line 19 "P4L20: I suggest removing "(i.e. the number of storage components)" as it an arbitrary measure of complexity."

**Commented [Rl11]:** R1: Line 34 Page 4 "I would also make a reference to Perrin et al. 2001 here"

**Commented [Rl12]:** R2: Page 6 Line 15 "Do the authors mean that their runs can be used as benchmark by future studies, as suggested on P12L12? Please clarify."

4. What is the predictive capability of those identified as behavioural models for then predicting annual maximum flows when applied in a parameter uncertainty framework?

To address these questions, we have applied the four core conceptual hydrological models from the FUSE hydrological framework to 1013 British catchments, within an uncertainty analysis framework. Model performance and predictive capability
5  have been evaluated at each catchment, providing a national overview of hydrological modelling capability for simpler lumped conceptualisations over Great Britain.

**2 Data and Catchment Selection**

**2.1 Catchment Data**

This study was national in scope, using a large data set of 1013 catchments distributed across Great Britain (GB). The
10  catchments cover all regions and include a wide variety of catchment characteristics including topography, geology and climate (see Table 1), and include both natural and human impacted catchments (see Figure 1). On average, rainfall is highest in the north and west of GB, and lowest in the south and east, with GB totals varying from a minimum of 500mm to a maximum of 4496mm per year (see Figure 2). There is also seasonal variation with the highest monthly rainfall totals generally occurring during the winter months and the lowest totals occurring in the summer months.
15  This pattern is enhanced by seasonal variations in temperature with evaporation losses concentrated in the summer months from April – September. Besides climatic conditions, river flow patterns are also heavily influenced by groundwater contributions. Figure 1 shows the major aquifers in GB. In catchments overlying the Chalk outcrop in the South-East, flow is groundwater-dominated with a predominantly seasonal hydrograph that responds less quickly to rainfall events. Land use and human modifications to river flows also significantly impact river flows, with river flows heavily modified in the South-East
20  and Midland regions of England due to high population densities (Figure 1). Most catchments have very little or no snowfall in an average year, but there are some upland catchments in northern England and northeast Scotland where up to 15% of the annual precipitation falls as snow (Figure 2).

Catchments were selected from the National River Flow Archive (Centre for Ecology and Hydrology, 2016) based upon the quality and availability of rainfall, potential evapotranspiration (PET) and river discharge data over the period 1988-2008. The
25  full NRFA dataset contains records for 1463 catchments across GB. Of these, 1013 had sufficient information (defined as more than 10 years of available discharge data during the model evaluation period of 1993-2008) available to include in this analysis.

**2.2 Observational Data**

Twenty-one years of daily rainfall and PET data covering the period 01/01/1988 to 31/12/2008 were used as hydrological model input. Rainfall timeseries were derived from the Centre for Ecology and Hydrology Gridded Estimates of Areal Rainfall,
30  CEH-GEAR (Tanguy et al., 2014). This is a 1km$^2$ gridded product giving daily estimates of rainfall for Great Britain (Keller

Commented [RI13]: R3: Page 16 Line 24 "P5 L14 - "these" should be "those""

Commented [RI14]: Tougher selection criteria – still more than 1000 catchments.

Commented [RI15]: R3: Page 16 Line 14 "3. P5 L22 - remove "Environment Agency", a catchment is a catchment, the EA don't own the catchments, even if they do own the gauges!"

Commented [RI16]: R1: Page 5 Line 17 "Table 1 is not cited within §2"

Commented [RI17]: R3: Page 16 Line 18 "4. Amend "Rainfall is highest in the West and North of GB and lowest in the East and South varying from a minimum of 500mm to a maximum of 4496mm per year (see Fig. 1)" to "On average, rainfall is highest in the north and west of GB, and lowest in the south and east, with GB totals varying from a minimum of 500mm to a maximum of 4496mm per year (see Fig. 1)."

Commented [RI18]: R3: Page 16 Line 25 "P6 L1 - remove the "of" after "South-East""

Commented [RI19]: R1: Page 5 Line 12 "In §2, I would give an estimation of the proportion of watersheds where snowmelt processes are observable (solid precipitation >20% of total precipitation ?)"

Commented [RI20]: R2: Page 10 Line 27 "P6L5: please define "sufficient""

R3C7 : Page 13 Line 26 "P6 L6 – 2 years of data was your criteria for catchment selection, this doesn't seem sufficient to me"

[revised manuscript text omitted]

**Commented [RI27]:** R3C2: Page 12 Line 19 "Section 3.2 – why NSE?"

**Commented [RI28]:** R3C6: Page 13 Line 18" P8 L23 – only a 1 year warm up period? This is not sufficient for many GW dominated catchments in the SE."

**Commented [RI29]:** R1: Page 5 Line 3 "mistake with O (mean of observed discharge)"

**Commented [RI30]:** Additional metrics were suggested by all reviewers

$$\triangle \mu = \frac{\mu_s - \mu_o}{\mu_o} \qquad (2)$$

where $\mu_s$ and $\mu_o$ refer to the mean of the simulated and observed annual cycle. Using this equation, an unbiased model would score 0 (a perfect score), and a model that underestimated or overestimated the mean annual flow would score a negative or positive value respectively. A value of +/- 1 would indicate an overestimation/underestimation of flow by 100%.

5 The relative difference in standard deviation was calculated using:

$$\triangle \sigma = \frac{\sigma_s - \sigma_o}{\sigma_o} \qquad (3)$$

where $\sigma_s$ and $\sigma_o$ represent the standard deviation of the simulated and observed mean annual cycle. Again, a value of zero indicates a perfect score with no error, and positive/negative values indicate an overestimation/underestimation of the amplitude of the mean annual cycle respectively.

10 The correlation was calculated using Pearson's correlation coefficient. A value of 1 indicates a perfect correlation between the observed and simulated flows, whilst a value of 0 indicates no correlation. This indicates model skill in capturing both timing and shape of the hydrograph.

**3.3 Evaluation of Model Predictive Capability**

15 In order to evaluate model predictive capability, the widely applied Generalised Likelihood Uncertainty Estimation (GLUE) framework was used (Beven and Freer, 2001; Romanowicz and Beven, 2006). The GLUE framework is based on the equifinality concept, that there are many different model structures and parameter sets for a given model structure which result in acceptable model simulations of observed river flow (Beven and Freer, 2001). This methodology has been widely applied to explore parameter uncertainty within hydrological modelling (Freer et al., 1996; Gao et al., 2015; Jin et al., 2010; Shen et

20 al., 2012) and includes approaches to directly deal with observational uncertainties in the quantification of model performance (Coxon et al., 2014; Freer et al., 2004; Krueger et al., 2010; Liu et al., 2009). For every catchment and model structure, an Efficiency score was calculated for each of the 10,000 Monte Carlo (MC) sampled parameter sets. Parameter sets with an efficiency score exceeding 0.5 were regarded as behavioural, therefore all other sampled parameter sets were rejected and so given a score of zero. Conditional probabilities were assigned to each behavioural parameter set based on their behavioural

25 Efficiency score, and these were normalised to sum to 1. This meant that the simulations which scored the highest efficiency value had larger conditional probabilities, and simulations which had efficiency values just above 0.5 would have lower conditional probabilities. For each daily timestep, a $5^{th}$, $50^{th}$ and $95^{th}$ simulated discharge bound was produced from these conditional probabilities, for each catchment and model structure individually as described in Beven and Freer (2001). This meant that simulations with a higher efficiency score were given a higher weighting when producing the discharge bounds.

30 Predictive capability for an additional performance metric regarding annual maximum flows was then calculated from these behavioural simulations to test the model's ability to predict peak flood flows over the 21 year period. Annual maximum flows

**Commented [RI31]:** R1 Page 2 Line 18: "Authors decided to use the classical Nash-Sutcliffe efficiency (NSE) index to evaluate model performances (and select behavioural models, NSE > 0.5). NSE index is famous and widely used in Rainfall-Runoff modeling. Even if the perfect efficiency index do not exists, this index is also known to have some drawbacks (Schaefli and Gupta, 2007, among many references). Gupta et al. (2009) introduced the Kling-Gupta efficiency index that allows to explicitly account for bias (mean and variability) and correlation, in the evaluation of model performances. Given the ambition of this paper, I would recommand the authors to consider in their analyses the Kling-Gupta efficiency index, or at least to decompose their results in terms of correlation and mean bias."

R2: Page 6 Line 30 "Since the authors aim to better understand "where and why these simple models may fail" the choice of NSE is somewhat suprising, since NSE is a measure of overall performance, which provides limited insights into the reasons for high or low performance. Although an evaluation based on hydrological signatures would have enabled a more process-based diagnostic of model failures, I am not requiring this, since it would imply significant additional analyses. However, if the authors stick to NSE (or use KGE), I suggest that they use benchmarks (as suggested by Seibert et al., 2018) to account for the fact that high NSE/KGE values can be relatively easy to reach depending on the catchment and the season. I believe this would enable a more fair and enlightening assessment of the hydrological models across the catchments."

**Commented [RI32]:** R2 Page 10 Line 17 "P4L12-17: this belongs to Data and Methods"

**Commented [RI33]:** R3C12: Page 14 Line 27 "12. P8 L3 - Can you explain conditional probabilities in more detail?"

[revised manuscript text omitted]

**Commented [RI40]:** R1 Page 5 Line 6 "values instead of vales"

**Commented [RL41]:** R2C7: Page 8 Line 22 "Comment 7: Just like hydrological behaviour, model performance is not determined by a single catchment characteristic, but rather, by the interaction of multiple catchment characteristics. So, firstly, would it be possible to consider a wider range of catchment attributes? So far, the authors employ the BFI, annual rainfall, the wetness index and the runoff coefficient, but many more attributes could be used to describe each catchment, but many more attributes could be used to describe each catchment (e.g., Beck et al., 2015). I encourage the authors to add other attributes, which they might have computed for other studies or retrieved from the UK hydrometric register, which they mention in Table 1, in order to describe the landscape in a more complete fashion (indicators of human interventions would also be useful, see below).
Comment 8: And secondly, I think it would be beneficial to better account for the interactions between these attributes. The authors combine several attributes in Figure 7 to explain model performance, which I find particularly interesting. Maybe that the analyses they will perform when revising this study will lead to more figures of this type, and enable a more systematic analysis of the interactions between these predictors (perhaps using regression trees, see Poncelet et al., 2017). This is critical to go from describing where models fail and to explaining why they fail."

so we have an even larger water deficit which the model structures cannot simulate. For the driest catchments, models have higher error in predicting the standard deviation and correlation.

**4.5 Benchmarking Predictive Capability for Annual Maximum Peak Flows**

Model predictive capability for simulating annual maximum (AMAX) flows from behavioural models defined from the NSE measure is shown in Figure 11 and Figure 12. Figure 11 assesses the ability of models to produce AMAX discharge estimates which are as close as possible to observations. Here, a value of 0 means simulated AMAX discharge is equal to observed, up to 1 means simulated AMAX discharge is within the bounds of the observational uncertainties applied and larger values such as 2 indicate that simulated discharge is double the limit of observational uncertainties away from the observed discharge (negative values mean that the model simulations are lower than the observed). Median $E_{amax}$ values from Eq. (2) are around -2.4 to -3.2 across all four models, with PRMS producing slightly better predictions in general than the other models. This shows that the models are underestimating peak annual discharges across the majority of GB catchments even though behavioural models have been selected using NSE which favours models that perform well at higher flows.

Figure 12 shows the percentage overlap between the simulated 5th and 95th AMAX bounds and the observed AMAX uncertainty bounds. Here, the boxplot on the left shows the variation of results across all catchments and models for each year, whilst the boxplot on the right summarizes results across all catchments and years for each model. The median value across all catchments is 0.16, meaning that there is a 16% overlap between the observed and simulated AMAX bounds averaged across all 20 years.

There are large variations in model ability to simulate observed annual maximum flows between years, when looking at median predictions. For example, 1990 and 2008, which were wetter than average years across most of GB, model ability to represent annual maximum discharge is poor. However, in 1996, which was a particularly dry year following the 1995 drought (Marsh et al., 2007), the models do a much better job of representing the annual maximum discharge. This may be in part due to the model tendency to underestimate discharge as seen in Figure 11. However, variations between years are less apparent when looking at 25th and 75th percentiles in Figure 122. This could suggest that there are some catchments where predictions are more consistent between years, or that the large climatic variation across GB may conceal some of the effects of inter-year differences.

**5 Discussion**

This study provides a useful benchmark of the performance and associated uncertainties of four commonly used lumped model structures across GB, for future model developments and model types to be compared against. The large number of catchments included makes this assessment a fair benchmark for any future national modelling studies, as well as smaller scale modelling efforts. A full list of models scores can be found at https://doi.org/10.5523/bris.3ma509dlakcf720aw8x82aq4tm

**Commented [RI42]:** R3C13: Page 15 Line 1 "13. P11 L 23 – "the top row of plots" – there is only one plot in Fig 8!"

**Commented [RI43]:** R3C14: Page 15 Line 6 "14. P12 L 7-8 "However, variations between years are less apparent when looking at 25th and 75th percentiles in Fig. 8." We can't distinguish variation between years from Fig 8?"

**Commented [RI44]:** Removed as repetition of results

**Commented [RI45]:** R3C8: Page 13 Line 32 "Reading through your discussion seems very repetitive of the results chapter. Can these be better synthesised, to reduce the discussion section?
16. Your discussion is longer than the rest of the paper put together!" – we have removed this paragraph to reduce repetition of the results.

**5.1 Identifying missing process parameterisations**

There were some clusters of catchments, notably catchments in northern and northeast Scotland and those on permeable bedrock in southeast England, where all models failed to produce good simulations. The Scottish catchments are mountainous catchments, at a considerably higher elevation than the rest of GB, and experience colder temperatures with daily maximum
5    temperatures in January consistently below zero (Met Office, 2014). Many catchments in northeast Scotland are classed as natural, but there are a group of catchments in central northern Scotland which are impacted by hydro-electric power (HEP) generation and subsequent diversions out of the catchment as well as storage influences on the regime (Marsh and Hannaford, 2008b). As model failures in northeast Scotland were particularly pronounced during winter and spring, this suggests that models were unable to capture the different seasonal climatic conditions of these catchments, such as snow accumulation and
10    melt or the impact of frozen ground. This is supported by the low correlations between simulated and observed flows in northeast Scotland, suggesting that the models are unable to represent the overall shape and timing of flows. Many catchments in central/northern Scotland had particularly low NSE values which were worst in summer/autumn. Modifications to the flow regime resulting from HEP can explain poor model performance for these catchments, supported by the models failing to reflect model bias and correlation. The FUSE models in this study do not incorporate snow processes and indicates that future
15    modelling efforts for GB may need to include a snowmelt regime, and the anthropogenic impacts resulting from hydroelectric power generationHEP, to produce good simulations in these catchments.

The catchments in southeast England receive relatively little rainfall compared to the rest of GB and are overlaying a chalk aquifer as can be seen in Figure. 2. Previous studies have found that hydrological models tend to perform better in wetter catchments (Liden and Harlin, 2000; McMillan et al., 2016), which could be part of the reason model performance is so poor
20    for these catchments. The presence of the chalk aquifer could also stress the models, as there is nothing in the model structures to account for groundwater and particularly groundwater flows between catchment boundaries. Equally, the South-East has some of the highest population densities in the UK and human influences can significantly impact flows in this region, particularly for a considerable proportion of river discharges throughout the Anglian region are abstracted, this clearly impacts lower flow conditions in the drier seasonal periods.

25    For catchments where groundwater is the reason for model failure, a possible solution could be to use a conceptual model that allows for groundwater exchange (as opposed to the models used here which all maintain the water balance). Hydrological models such as GR4J and SMAR have been developed with functions that allow models to gain or lose water, to represent inter-catchment groundwater flows (Le Moine et al., 2007). The use of these models where there is evidence of groundwater flows can help to improve model performance and reduce discrepancies between observed and simulated flows, but must be
30    used with caution to avoid overfitting of the water balance where there is no physical reasoning for a catchment to be gaining or losing water. Whilst it has been noted that there is a general pattern of poor performance for catchments in southeast England, it is hard to disentangle the reasons that this may be the case. Both the underlying chalk geology causing water transfer between catchments, and heavily human modified flow regimes could explain model failures which are greatest during the summer.

**Commented [RI46]:** R3: Page 15 Line 15 "Comment 17. P13 L 3 – you've made no reference to anthropogenic influences in Scotland. This statements seems a bit throwaway."

**Commented [RI47]:** R3C17 Page 15 Line 15 "17. P13 L 3 – you've made no reference to anthropogenic influences in Scotland. This statements seems a bit throwaway."

**Commented [RI48]:** R3C18 Page 15 Line 19 "P13 L9 – it is not just the Thames basin that is affected by abstractions! A lot of Anglian region is VERY heavily influenced."

Interestingly, McMillan et al., (2016) found that whilst aquifer fraction was expected to have a strong link to model performance, no relationship was found for the TOPNET model applied in New Zealand.

**5.2 Influence of Catchment Characteristics and Climate on Model Performance**

One of the key advantages of large-sample studies is that by applying models to many catchments, we can see general trends and identify important catchment characteristics or climates that are not represented well by our choice of model structures. We found that looking at the catchment water balance, considering the relationship between catchment precipitation, evaporation and observed flows, helped to identify common features of catchments where all models were failing (Figs. 4,95,10). All model structures produced poor simulations in catchments where either total runoff exceeded total rainfall or where observed runoff was very low compared to total rainfall, and this runoff deficit could not be accounted for by evapotranspiration losses alone. These differences in water balance are likely due to human modifications to the natural flow regime such as dams, effluent returns or inter-catchment water transfers, groundwater flow between catchments or it is also possible that there are systematic errors in the observational data and this information is dis-informative (Beven and Westerberg, 2011; Kauffeldt et al., 2013). Most of these catchments were located within chalk aquifers in southeast England, and therefore are in a heavily urbanised area where groundwater abstractions and flows between catchments could be expected. The simple, lumped models used here were only given inputs of observed precipitation and PET, therefore they are unable to account for the additional observed runoff and so are 'stressed' even in terms of simulating mean annual runoff, irrespective of more detailed hydrograph behaviour.

We also found that catchment characteristics were important in determining which model structure was most appropriate. For catchments with a high baseflow index, only the ARNO/VIC model was able to produce behavioural simulations. This could be explained by the strong non-linear relationship in the upper storage zone of the ARNO/VIC model, which separates it from the other model structures. This enables the ARNO/VIC model to constrain the fast rainfall-runoff processes, which would only occur for extreme events in these groundwater dominated catchments and so allow for a complex mixture of highly non-linear saturated fast responses coupled with more general baseflow dynamics to be captured effectively. The catchment annual rainfall total also influenced which model structure was most appropriate. We found that for catchments with average annual rainfall values of around 2000mm/year or lower, the SACRAMENTO model structure is more dominant. As we move towards catchments with higher annual rainfall, the relative importance of the different structures shift until all structures are approximately equal for the catchments with the highest annual rainfalls. This shows that for very wet catchments, the model structure is less important as all models can produce behavioural simulations through some part of the parameter space, as seen by the relatively high number of behavioural simulations for wetter catchments (Fig. 85b). This agrees with previous studies, where models have been found to perform better for wetter catchments, which are likely to have more connected saturated areas, as there is a more direct link between rainfall and runoff (McMillan et al., 2016).

**Commented [RI49]:** R1: Page 4 Line 19 "authors discuss about groundwater flows between catchments, with losses or gain of waters. This problem is not new and some conceptual modelisation could be found in the literature since one or two decades. In a natural context, authors could make a reference to Le Moine at al. (2007, 2008) papers about groundwater flows and water balance closure. The existence of such groundwater flows in permeable geological context (chalk, limestones and/or karstic systems, etc.) was one of the reasons of the development of a groundwater exchange function within the GR model family. The use of this function should be motivated by (hydrogeologic) evidences of such groundwater flows (in order to avoid "overfitting" of the water balance, i.e. fudge factor), but might be useful in catchments where water balance is difficult to close, such as the one influenced by chalk aquifers in southeast england."

R2C5: Page 13 Line 8 "Catchment characteristics and climate – do all FUSE models maintain the water balance? Can you comment on the existence of models that don't (e.g. GR4J), and how those may overcome such problems? What are the implications of maintaining vs not maintaining water balance in conceptual lumped models? Are the four models you've chosen actually quite similar to each other? I think you need to make more of this somehow."

**Commented [RI50]:** R3C8: Page 13 Line 32 "Reading through your discussion seems very repetitive of the results chapter. Can these be better synthesised, to reduce the discussion section? 16. Your discussion is longer than the rest of the paper put together!"

**Commented [RI51]:** R2C6 Page 8 Line 4 "How critical is the selection of model structure? There are cases of great equifinality (i.e., high NSE for all structures, mostly for humid catchments). As mentioned above, a high NSE is not a guarantee that the model structure is adapted, but as long as this is recognised (and this could be clearer throughout the manuscript), I think it is fine for this study. But in other (more interesting) catchments, some model structures clearly outperform other structures, and there, model choice is critical. I think this should be stressed more prominently, since these are cases in which the inadequacy of the model structure cannot be overcome by parameter tuning. Given the general tendency of using the same model structure across very diverse environments (as discussed e.g. by Addor and Melsen, 2019), I think this is an important result, which could be underscored more. A related question is: which catchment characteristics explain these large NSE differences between model structures?"

R3C20 Page 20 Line 29 "20. P13 L15 – "The ensemble of model structures was able to take advantage of this" - this seems to be a contradictory argument to the previous statement that the models all have similar performance to each other on a catchment by catchment basis. I think you need to tease these two arguments out better somehow. E.g. in some situations the choice of a different model can yield better results (e.g. high baseflow), but in other situations, none of the models can do well (e.g. abstractions). What are the implications of this? "

[revised manuscript text omitted]

**Commented [RI53]:** R1 Page 3 Line 3: "One of the main drawback of the NSE (and linear regression as well) is that the standard deviation of the simulated time-series is biased and underestimated, i.e. flood underestimated and drought overestimated. Among other arguments, this drawback partly explain why flood values are underestimated. I would add at least a comment on the fact that this statement is dependent on the behavioural model selection metrics in §5.4."

R2C5: Page 7 Line 27 "Can they formulate hypotheses on why annual maximum flows are underestimated, which could be tested by future studies?"

high baseflow catchments yet the PRMS model was the best at capturing AMAX peak flows. Furthermore, it was found that for some catchments only a selection of the model structures were able to produce good simulations, such as the baseflow dominated catchments which only ARNO could simulate well. For these catchments, selection of the appropriate model structure is important to produce good simulations and unsuitability of the model structure cannot be corrected for through parameter calibration. This supports previous research highlighting the importance of considering alternative model structures and using model structure ensembles or flexible frameworks such as FUSE (Butts et al., 2004; Clark et al., 2008; Perrin et al., 2001). Consequently, future hydrological modelling over a national scale and/or over a large sample of catchments need to ensure appropriate model structures are selected for these catchments and consider the possibility of using multiple model structures to represent hydrological processes in varied catchments.

The results also highlighted the importance of considering parameter uncertainty. It was shown that there were often many different parameter sets which could produce good simulation results for the same model structure. For some catchments, particularly the wetter catchments in the west, all model structures were able to produce good simulations through sampling the parameter space. We also show how behavioural parameter distributions change with regards to BFI (Figure 9Figure 8), which shows expected shifts in some of the common behavioural parameters/concepts for different conditions, showing the model behaviour and parameter formulations are in general making rationale sense (i.e. Higher BFI equals higher time delays). While this study incorporated uncertainties in model structures and parameters, future work will also focus on incorporating uncertainties in the data used to drive hydrological models and more sophisticated representation of discharge uncertainties. This is important because errors in observational data will introduce errors to runoff predictions when fed through rainfall-runoff models (Andréassian et al., 2001; Fekete et al., 2004; Yatheendradas et al., 2008), and in conjunction with uncertainties in the observational data used to evaluate hydrological models will also affect our ability to calibrate and evaluate hydrological models (Blazkova and Beven, 2009; Coxon et al., 2014; McMillan et al., 2010; Westerberg and Birkel, 2015).

**6 Summary and Conclusions**

In this study, we have benchmarked the performance of an ensemble of lumped, conceptual models across over 1000 catchments in Great Britain.

Overall, we found the four models performed well over most of Great Britain with each model producing simulations exceeding 0.5 Nash Sutcliffe efficiency over at least 80% of catchments. The performance of the four models was similar, with all models showing similar spatial patterns of performance, and no single model model outperforming the others across all catchment characteristics for both daily flows and peak flows. However, decomposing NSE into model performance for bias, standard deviation error and correlation, clear differences emerged between the best simulation produced by each of the model structures. The ensemble did better than each individual model, demonstrating the value of model structure ensembles when exploring national-scale hydrology.

> **Commented [RI54]:** R3: Page 17 Line 1 "P16 L19 – refer to Fig 6"

> **Commented [RI55]:** R3C4 Page 12 Line 30 "Your statement in the abstract L23 that NSE scores of 0.72-0.78 were achieved for all catchments is misleading. How useful a measure is the "median maximum NSE for all the catchments"? It's pretty cryptic. There are catchments in E Scotland, and Anglian region that are showing pink/red for all 4 models, so NSE must be <0.5. Having got to page 9 I now see what you meant, but it isn't clearly stated. The sentences on P12 L16-17 are a better summary of the performances across catchments. Same issue on P16 L 32."

We found that all models showed higher skill in simulating the wet catchments to the west, and all models failed in areas of Scotland and southeast England. Seasonal performance and analysis of the water balance suggested that these model failures could be at least in part attributed to missing snowmelt or frozen ground processes in Scotland and chalk geology in southeast England where water was able to move between catchment boundaries. In general, we found models performed poorly for catchments with unaccounted losses or gains of water, which could be due to measurement errors, water transfer between catchments due to groundwater aquifers and human modifications to the water system. Therefore, these factors would need to be considered in a national model of Great Britain.

We also evaluated model predictive capability for high flows, as good model performance in replicating the hydrograph, assessed using Nash-Sutcliffe efficiency, does not necessarily mean models are performing well for other hydrological signatures. We found that the FUSE models tended to underestimate peak flows, and there were variations in model ability between years with models performing particularly poorly for extremely wet years.

This benchmark series provides a useful baseline for assessing more complex modelling strategies. From this we can resolve how or where we can and need to improve models, to understand the value of different conceptualisations, linkages to human impacts, and levels of spatial complexity our model frameworks could deploy in the future. Therefore, the results of this study are made available at https://doi.org/10.5523/bris.3ma509dlakcf720aw8x82aq4tm.

**Commented [RI56]:** R1 Page 5 Line 9 "for catchments, repeated 2 times"

[revised manuscript text omitted]

Commented [RI57]: R2C9: Page 9 Line 1 "Anthropogenic activities are repeatedly mentioned to explain poor model performance (e.g., P12L29, P14L16, P15L3). This is indeed plausible, but if qualitative or maybe quantitative indicators of the extent of human interventions could be included, so that their impacts on streamflow and model performance could be demonstrated or maybe even quantified, it would strengthen the study."

[Figure]

**Figure 2: A) Major aquifers across Great Britain, based upon BSS Geology 625k, with the permission of the British Geological Survey B) Mean annual rainfall for 10km² rainfall grid cells across Great Britain. C) Fraction of rainfall falling as snow for catchments across Great Britain, where a value of 0.15 indicates that 15% of the catchment precipitation falls on days when the temperature is below zero.**

**Commented [RI58]:** R1: Page 5 Line 12 "In §2, I would give an estimation of the proportion of watersheds where snowmelt processes are observable (solid precipitation >20% of total precipitation ?)"

[Figure]

**Figure 3: FUSE wiring diagram, showing the model structure decisions. TOPMODEL and ARNO/VIC have 10 parameters, PRMS has 11 parameters and SACRAMENTO has 12 parameters.** Adapted from Clark et al., (2008).

Commented [RI59]: R1: Page 5 Line 30 "In Figure 2, I would put the number of free parameters to calibrate."

[Figure]

**Figure 4. Distribution of model performance across all catchments, for all 4 individual model structures and the model structure ensemble. Each plot shows model performance assessed using a different metric. Top left shows model performance assessed using Nash-Sutcliffe Efficiency, top right shows model relative bias or relative error in simulated mean runoff (%), bottom left shows relative error in the standard deviation of runoff (%), and bottom right gives correlation between observed and simulated streamflow.**

[Figure]

**Figure 54: GB maps of model performance for each structure. Each point is a gauge location which is coloured based upon the best Nash Sutcliffe score attained by the model for that catchment.**

[Figure]

**Figure 65. GB maps of model performance for each structure for 3 different metrics. Top row shows model relative bias or relative error in simulated mean runoff (%), middle row shows relative error in the standard deviation of runoff (%), and the third row shows correlation between observed and simulated streamflow. Each point is a gauge location which is coloured based upon the best score for that metric.**

**Commented [RI60]:** R1 Page 2 Line 18: "Authors decided to use the classical Nash-Sutcliffe efficiency (NSE) index to evaluate model performances (and select behavioural models, NSE > 0.5). NSE index is famous and widely used in Rainfall-Runoff modeling. Even if the perfect efficiency index do not exists, this index is also known to have some drawbacks (Schaefli and Gupta, 2007, among many references). Gupta et al. (2009) introduced the Kling-Gupta efficiency index that allows to explicitly account for bias (mean and variability) and correlation, in the evaluation of model performances. Given the ambition of this paper, I would recommand the authors to consider in their analyses the Kling-Gupta efficiency index, or at least to decompose their results in terms of correlation and mean bias."

R2: Page 6 Line 30 "Since the authors aim to better understand "where and why these simple models may fail" the choice of NSE is somewhat suprising, since NSE is a measure of overall performance, which provides limited insights into the reasons for high or low performance. Although an evaluation based on hydrological signatures would have enabled a more process-based diagnostic of model failures, I am not requiring this, since it would imply significant additional analyses. However, if the authors stick to NSE (or use KGE), I suggest that they use benchmarks (as suggested by Seibert et al., 2018) to account for the fact that high NSE/KGE values can be relatively easy to reach depending on the catchment and the season. I believe this would enable a more fair and enlightening assessment of the hydrological models across the catchments."

[revised manuscript text omitted]

**Supplementary Information 1 – Plots looking at the relationship between catchment characteristics and model performance**

In the main body of the paper, weThe main body of the paper looked at the relationship between the catchment wetness index and runoff coefficient and model performance. Thᶜese attributesis wᵉereas selected as theyse variables were strongly were related

5  to model performance and explained differences between catchments. Here, we providegive additional plots looking at the relationship between model performance and many different catchment characteristics to demonstrate how other catchment attributes impact model performance. These characteristics were either taken from the hydrometric register or calculated from the model input data timeseries (Centre for Ecology and Hydrology, 2016; Marsh and Hannaford, 2008a; Robinson et al., 2015a).

Figure S1 summarises the overall performance of the four models, and was used to help identify overall trends in model structure performance across all catchments. Figures S2 S1 – S5 S4 are scatter plots looking at the relationship between model performance (assessed using NSE, bias, error in standard deviation and correlation respectively) and different catchment attributes. Figures S6 S5 onwards are plots looking at interactions between different catchment attributes and model

15  performance.

Some links between catchment attributes and model performance can be seen from Figures S1-S4. Firstly, Ssmall catchments (<200km$^2$) tend to have more variable NSE scores (both high and low), whilst large catchments (>3000km$^2$) always do fairly wellare easier to model. This is seen with all the decomposed metrics — with small catchments more likely to have errors in

20  bias and standard deviation of flows, and having more variable correlation scores. A possible explanation for this could be that daily data is less able to capture the variation in these catchments. potentially indicating that daily data is not able to capture flow variation in small catchments. Secondly, Bbaseflow dominated catchments (BFI > 0.7) are more likely to gain really very low NSE values (although some high BFI catchments can be simulated well). Interestingly, BFI seems to have a relationship with error in the standard deviation, with baseflow dominated catchments the only catchments where the best simulations tend

25  to overpredict variation. This can be explained, as groundwater would dampecould be due to groundwater dampeningn variation in flows. Thirdly, Ggauge elevation seems to cap overall model performance—. with higher elevation gauges unable to achieve performance scores as high as low elevation gauges. Or this could potentially be a pattern because there are fewer high elevation gauges. This could also be a problem with using NSE, as lower scores are naturally given to catchments with more seasonal variation. We see that for these high elevation gauges, the best simulations always underpredict the flow

30  standard deviation. Finally, it is Ssurprisingly, that urbanisation does not seem to decrease model performance.

From figures S6 S5 onwards we can see that the worst NSE performaning catchments in terms of Nash-Sutcliffe efficiency are grouped being small catchments less than 120km$^2$, with elevations below 125m, mid to high BFIs (>0.5), low annual rain less than 1000mm and annual runoff values which differ from other catchments with similar annual rainfall totals. Poor NSE

~0.5 is achieved for wetter catchments (annual rain > 1200mm), which have relatively low annual runoff generally less than 900mm. Many have flow attenuation from reservoirs and lakes, and for these catchments correlation is poor.

The largest problems with standard deviations seem to be small catchments, where standard deviation is generally
5   underpredicted except for catchments with a high BFI where it is sometimes overpredicted. This can be explained as baseflow dominated catchments may have less variation in flow so it is most likely that variation is overpredicted for these catchments. For small catchments relative errors are also most likely to be high.
Bias is very clearly linked to the climatic variables. There is a clear linear relationship with rainfall and runoff, and catchments deviating from this underpredict mean flows and have poorer correlations. Wetter catchments have reduced relative biases,
10   which is likely to be due to biases being normalised by mean flow.

[Figure]

Figure S1. Cumulative distribution functions showing performance of the 4 model structures across all catchments, when
15   assessed using NSE and decomposed metrics for the simulation with the highest NSE.

[Figure]

Figure S12: Relationship between NSE and a selection of 15 catchment descriptor variables. Column 1 is moregives general catchment attributes from the hydrometric register (Marsh and Hannaford, 2008). These are catchment area (km$^2$), Baseflow index (BFI), Gauge elevation (m above sea level), mean drainage path slope (DPSBAR) which indicates overall catchment steepness in metres per kilometre, and flood attenuation by reservoirs and lakes (FARL) where values close to 1 indicate the absence of flow attenuation and values below 0.8 indicate a substantial influence. CColumn 2 gives hydroclimatic attributes calculated from our data, and proportion catchment is wet (PROPWET) from the hydrometric register. Annual Loss is Rainfall-

Runoff, whilst Annual flood is the Median Annual maximum flood peak, and all are reported in mm. Column 3 gives land-use and bedrock permeability descriptors (%), also from the UK hydrometric register.

[Figure]

5    Figure S3S2: Relationship between bias and a selection of 15 catchment descriptor variables, as in Figure S2S1.

[Figure]

Figure S4S3: Relationship between error in standard deviation and a selection of 15 catchment descriptor variables, as in Figure S2S1.

[Figure]

Figure S5S4: Relationship between correlation and a selection of 15 catchment descriptor variables, as in Figure S2S1.

[Figure]

Figure S5: Relationship between general catchment characteristics, coloured by model ensemble NSE score for that catchment. Column 1 gives general catchment attributes from the hydrometric register (Marsh and Hannaford, 2008). These are catchment area (km$^2$), Baseflow index (BFI), Gauge elevation (m above sea level), mean drainage path slope (DPSBAR) which indicates overall catchment steepness in metres per kilometre, and flood attenuation by reservoirs and lakes (FARL) where values close to 1 indicate the absence of flow attenuation and values below 0.8 indicate a substantial influence. Column 2 gives hydroclimatic attributes calculated from our data, and proportion catchment is wet (PROPWET) from the hydrometric

register. Annual Loss is Rainfall-Runoff, whilst Annual flood is the Median Annual maximum flood peak, and all are reported in mm. Column 3 gives land-use and bedrock permeability descriptors (%), also from the UK hydrometric register.

[Figure]

Figure S6b: Relationship between general catchment characteristics, coloured by model ensemble bias score for that catchment.

[Figure]

Figure S6c: Relationship between general catchment characteristics, coloured by model ensemble error in standard deviation for that catchment.

[Figure]

Figure S6d: Relationship between general catchment characteristics, coloured by model ensemble correlation for that catchment.

[Figure]

Figure 7aS6. Same as figure S6S5, but this time looking at hydroclimatic catchment descriptors. Annual rainfall (mm), annual runoff (mm), annual loss (mm) and mean annual maximum flood (mm), were all calculated from the model input data used in

this study. PROPWET is a measure of the percentage of time soils are wet, as calculated by the UK hydrometric register (Marsh and Hannaford, 2008).

[Figure]

Figure 7b. Same as figure S6, but this time looking at hydroclimatic catchment descriptors.

[Figure]

Figure 7c. Same as figure S6, but this time looking at hydroclimatic catchment descriptors.

[Figure]

Figure 7d. Same as figure S6, but this time looking at hydroclimatic catchment descriptors.